
# Measurements of spectral irradiance during the solar eclipse of 21 August 2017: reassessment of the effect of solar limb darkening and of changes in total ozone

Germar Bernhard[1], Boyan Petkov[2]

[1]Biospherical Instruments Inc., San Diego, CA 92110, USA
[2]Institute of Atmospheric Sciences and Climate (ISAC) of the Italian National Research Council (CNR), I-40129 Bologna, Italy

*Correspondence to*: Germar Bernhard (bernhard@biospherical.com)

**Abstract.** Measurements of spectral irradiance between 306 and 1020 nm were performed with a GUVis-3511 multi-channel filter radiometer at Smith Rock State Park, Oregon, during the total solar eclipse of 21 August 2017. The radiometer was equipped with a shadowband, allowing to separate the global (sun and sky) and direct components of solar radiation. Data were used to study the wavelength-dependent changes of solar irradiance at Earth's surface. Results were compared with theoretical predictions using three different parameterizations of the solar limb darkening (LD) effect, which describes the change of the solar spectrum from the Sun's center to its limb. Results indicate that the LD parameterization that has been most widely used during the last 15 years underestimates the LD effect, in particular at UV wavelengths. The two alternative parameterizations are based on two independent sets of observations from the McMath-Pierce Solar Telescope. When these parameterizations are used, the observed and theoretical LD effects agree to within 4 % for wavelengths larger than 400 nm and occultation of the solar disk of up to 97.8 %. Maximum deviations for wavelengths between 315 and 340 nm are 7 %. These somewhat larger differences compared to the visible range may be explained with varying aerosol conditions during the period of observations. Aerosol optical depth (AOD) and its wavelength dependence was calculated from measurements of direct irradiance. When corrected for the LD effect, AOD monotonically decreases over the period of the eclipse: from 0.41 to 0.32 at 319 nm and from 0.05 to 0.04 at 1018 nm. These results show that AODs can be accurately calculated during an eclipse if the LD effect is corrected. The total ozone column (TOC) was derived from measurements of global irradiance at 306 and 340 nm. Without correction for the LD effect, the retrieved TOC increases by 20 DU between the 1st and 2nd contact of the eclipse. With LD correction, the TOC remains constant to within natural variability (±2.6 DU or ±0.9 % between 1st and 2nd contact and ±1.0 DU or ±0.3 % between 3rd and 4th contact). In contrast to results of observations from earlier solar eclipses, no fluctuations in TOC were observed that could be attributed to gravity waves, which can be triggered by the supersonic speed of the Moon's shadow across the atmosphere. Furthermore, systematic changes in the ratio of direct and global irradiance that could be attributed to the solar eclipse were not observed. This finding agrees with results of three-dimensional radiative transfer models but contradicts reports from earlier observations, which indicate that the diffuse-to-direct ratio may change by 30 %. Our results advance the understanding of the effects of solar LD on the spectral irradiance





at Earth's surface, the variations of ozone during an eclipse, and the partitioning of solar radiation in direct and diffuse components.

## 1 Introduction

A total solar eclipse could be observed on 21 August 2017 across the United States, from Oregon in the West to South Carolina in the East (Pasachoff, 2017). We performed measurements of global (direct Sun plus sky) and diffuse (sky only) spectral irradiance during this event with a GUVis-3511 multi-channel radiometer, which was equipped with a shadowband to separate the two components. Data collected during this campaign can be used for studying processes initiated in the atmosphere by the passing of the Moon's shadow and for validating three-dimensional radiative transfer (3DRT) calculations that simulate the irradiance at Earth's surface during the period of totality (Emde and Mayer, 2007). Similar observations have been used during previous solar eclipses to study wavelength-dependent changes in spectral irradiance (Blumthaler et al., 2006; Kazadzis et al., 2007); the change in the contribution of the diffuse irradiance to the global irradiance over the period of the eclipse (Zerefos et al., 2000, 2001); short-term and longer-lasting fluctuations in the total ozone column (TOC) (Antón et al., 2010; Kazadzis et al, 2007; Mateos et al., 2014; Mims and Mims, 1993; Zerefos et al., 2000; 2001; 2006); variations in the $NO_2$ column (Adams et al., 2010); and comparison of measurements during totality with 3DRT model results (Kazantzidis et al., 2007). We revisit some of these issues by focusing on the wavelength dependence of solar limb darkening (LD), ozone observations, and the changing direct and diffuse radiation. For example, some observations of changes in the TOC during an eclipse are contradictory. Increasing TOCs were reported for measurements from Dobson spectrophotometers (Bojkov, 1968) and NILU-UV multifilter instruments (Antón et al., 2010) while measurements with Brewer photometers generally decrease as a eclipse progresses (Kazadzis et al, 2007). These differences cannot be explained with a real change in TOC but must be an artifact from either measurement or data processing.

Our measurements were also used to assess the quality of data of the GUVis-3511 radiometer, which is a relatively new instrument type. For this purpose, a new method to calibrate the instrument was developed and is described in detail in the Supplement to this publication. Data from the campaign are available for download and can, for example, be used to study variations of radiation during totality with 3DRT simulations.

## 2 Location and local conditions

The total eclipse was observed at Smith Rock State Park, located in Oregon between Bend and Madras. The partial eclipse started at 16:06:27 UT (1st contact, local time of 9:06:27 Pacific Daylight Time (PDT)) and ended at 18:41:02 UT (4th contact, 11:41:02 PDT). Totality occurred for one minute and 25 seconds, between 17:19:42 (2nd contact, 10:19:42 PDT) and 17:21:07 UT (3rd contact, 10:21:07 PDT). All times from here onward refer to UT. The GUVis-3511 radiometer was set up





at 44.362950° N and 121.139667° W, 867 m above sea level. The site was approximately 37 km south of the centerline of the Moon's shadow (e.g., the line where the duration of totality is the longest). The ground surrounding the instrument was covered by dry grass. The park's prominent mountain peaks were mostly in the west, north, and northwest of the instrument (Fig. 1), and extended up to 12.5° above the horizon. The average extension was 5.75°. By assuming isotropic sky radiance

distribution, we determined that objects above the horizon reduced the (cosine-weighted) diffuse irradiance by about 1.1 %. Data were not corrected for this effect.

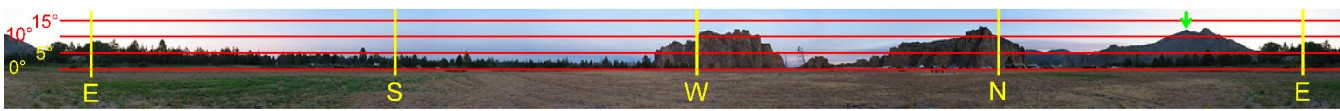

Fig. 1. Panorama as seen from instrument location. Mountains are located in the west (W), north (N), and northeast, and

extend up to 12.5° above the horizon. Trees restrict the horizon in eastern (E) direction by up to 9°. There is little (<5°) obstruction towards the south (S). The green arrow marks the mountain used for calibrating the elevation scale (red lines).

The state of Oregon experienced a total of 1069 reported wildfires in 2017, burning a total area of 451,863 acres or 1829 km$^2$ (https://en.wikipedia.org/wiki/2017_Oregon_wildfires). There were several active wildfires in the vicinity of the observation

site and the air was filled with smoke and aerosols during the days preceding the eclipse. However, a few hours before the start of the eclipse, the wind direction changed and the aerosol loading decreased substantially. Aerosol optical depths (AOD) measured shortly before and after the eclipse period are discussed in Sect. 7.1. The sky was free of clouds in the direction of the Sun with small clouds lingering only near the horizon.

### 3 Instrumentation

Measurements were performed with a GUVis-3511 multi-channel filter radiometer (Seckmeyer et al., 2010) designed and built by Biospherical Instruments Inc (BSI). The system was set up on a sturdy tripod (Fig. 2) and powered by a 12 V dry-cell car battery and a sine-wave inverter.

The instrument was equipped with 18 channels with spectral bandwidths of approximately 10 nm and the following nominal

wavelengths (nm): 305, 305, 313, 320, 340, 340, 380, 395, 412, 443, 490, 532, 555, 555, 665, 875, 940, and 1020. Spectral response functions of these channels are shown in Fig. 3 and were used for the calibration of the instrument (Sect. 4 and Supplement). Data from a 19[th] channel measuring photosynthetic active radiation (PAR) were not used in this study. Each channel uses a hard-coated, ion-assisted deposition interference filter plus bandpass filters for additional out-of-band rejection that are coupled to a "microradiometer" (Morrow et al., 2010). Microradiometers consists of a silicon photodiode,

three-stage preamplifier, 24 bit analogue-to-digital converter, microprocessor, and an addressable digital port. Data streams





from the 19 microradiometers are combined and transmitted via a USB interface to a laptop computer. Measurements at 305, 340, and 555 nm were performed with two channels, respectively, using either a standard production photodiode (model S1336 from Hamamatsu) or an alternative photodiode (model S12698 from Hamamatsu) for evaluation purposes. The instrument's internal temperature was stabilized to 40±0.5 °C.

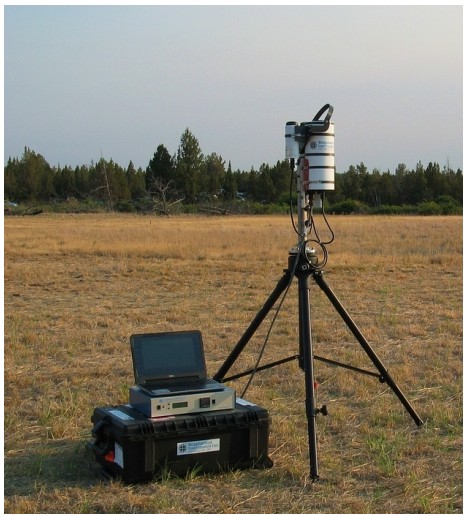

Fig. 2. Setup of the GUVis-3511 radiometer at Smith Rock State Park. The radiometer (white cylinder), BioSHADE assembly and BioGPS are mounted on a tripod. Shown next to the instrument is the laptop for recording the data, the instrument's control unit, and a black case containing the power source.

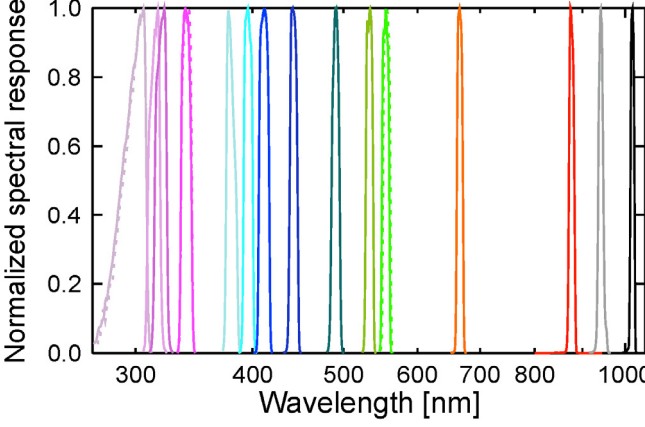

Fig. 3. Normalized spectral response functions of the GUVis-3511 radiometer discussed in this study. Functions were measured with BSI's spectral tester (Bernhard et al., 2005a) and corrected for the finite bandpass of the tester using a simple deconvolution routine also described by Bernhard et al. (2005a). Color coding is identical to that in Fig. 10. Note that the x-
15  axis is logarithmic to emphasize the UV range.



The filtered microradiometers point at the center of an irradiance collector, which features a composite diffuser made of layers of generic and porous polytetrafluoroethylene (PTFE) sheets (Hooker et al., 2012). This design leads to relatively small cosine errors also in the infrared, where the scattering properties of traditional PTFE diffusers are typically degraded. The directional response of the collector is virtually independent of wavelength for wavelengths smaller than 800 nm; for

larger wavelengths, the response is slightly lower (Fig. 4). Cosine errors (Seckmeyer and Bernhard, 1992) of all channels are smaller than ±5 % for incidence angles smaller than 78°. The error in measuring isotropic radiation is smaller than 0.3 % for wavelengths smaller than 800 nm, and –1.8 % (worst case) for the 1020 nm channel.

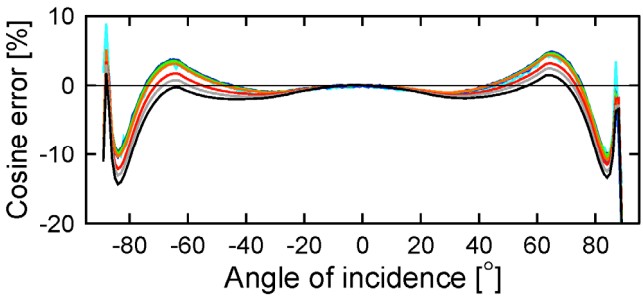

Fig. 4. Cosine error of the GUVis-3511 radiometer in one azimuthal plane at wavelengths between 395 and 1020 nm. Color coding is identical to that of Fig. 10. Measurements performed at the orthogonal plane are similar. Measurements at wavelengths below 395 nm are affected by noise due the low UV output of the incandescent lamp used for the characterization, and are not shown for clarity.

The radiometer was equipped with a computer-controlled shadowband, called BioSHADE (Morrow et al., 2010). The band can either be "stowed" below the instrument's diffuser for measuring global spectral irradiance or move slowly over the instrument to collect data for calculating the direct spectral irradiance from the solar beam with an algorithm developed for the BioSHADE (Hooker et al., 2012; Witthuhn et al., 2017). In brief, the algorithm first determines the spectral irradiance when the centers of diffuser, shadowband, and Sun are collinear. This is the spectral irradiance from the sky minus the

portion of the sky that is shaded by the band. Next, the spectral irradiance from the direct Sun plus the unshaded portion of the sky is obtained by analyzing several seconds of measurements when the shade from the band is close to the diffuser but not in contact with it. These measurements are then combined with the directional test data (Fig. 3) and measurements of global spectral irradiance to calculated cosine-error-corrected global and direct spectral irradiances.

For observing the solar eclipse, the band was programmed to perform one shadowband sweep every two minutes, between 14:36 and 19:30. Within each two-minute window, this configuration resulted in 45 seconds of global spectral irradiance measurements (with the shadowband stowed and applying a sampling rate of 1 Hz) and 75 seconds of data during which the band rotated by 180°. The sampling rate was set to 15 Hz during these "sweeps" to adequately resolve the short period when



the band's shadow moves over the diffuser. The system is also equipped with a GPS receiver, termed BioGPS, which was used to determined the geoposition specified in Sect. 2.

## 4 Calibration

The GUVis-3511 radiometer was calibrated using a new method described in the Supplement of this publication. Calibrated measurements report the solar spectral irradiance at the instrument's collector at a spectral resolution of 1 nm. More specifically, calibrated measurement resemble measurements of a hypothetical spectroradiometer with a slit function $s(\lambda)$, where $s(\lambda)$ is a triangular function with a bandwidth of 1 nm full width at half maximum (FWHM). For the channels with nominal wavelengths of 305 and 313 nm, the instrument was vicariously calibrated against measurements of a SUV-100 spectrometer, which has a spectral resolution of 1 nm. All other channels were calibrated using a calibration lamp traceable to a standard issued by the U.S. National Institute of Standard and Technology. The calibration methods takes into account that the bandwidth of GUVis-3511 channels is about 10 nm FWHM. The calibration was further optimized for the conditions (solar zenith angle, TOC, AOD, etc.) at the measurement site. Results indicated that it is advantageous to calibrate the channels with nominal wavelengths of 305 and 313 nm for spectral irradiance at 306 and 315 nm, respectively, and measurements reported in the following are therefore referenced to these two wavelengths. Finally, an uncertainty budget was established. Expanded ($k = 2$, equivalent to a confidence interval of 95 %) uncertainties for the 305 and 313 nm channels are 7.5 and 7.3 %, respectively. Expanded uncertainties of all other channels, with the exception of the 940 nm channel are 2.7 %. An uncertainty budget for the 940 nm channel is not provided here because of the large effect of water vapor on measurements at this wavelength. Measurements of this channel should therefore be interpreted with caution.

## 5 Secondary data products

Secondary data products derived from the measurements of the GUVis-3511 include AOD and TOC.

### 5.1 Aerosol optical depth

Aerosol optical depth $\tau_a$ was calculated from calibrated direct measurements and Beer-Lambert's law:

$$\tau_a = \frac{-1}{\mu_a}\left[\ln\left(\frac{\rho \hat{E}_{S_d}(\lambda_i)}{\hat{E}_0(\lambda_i)}\right) + \tau_r\mu_r + \tau_o\mu_o\right],\tag{1}$$

where

$\hat{E}_{S_d}(\lambda_i)$    is the response-weighted direct irradiance measured by the instrument (see Supplement for definition),



$\hat{E}_0(\lambda_i)$  is the extraterrestrial solar spectrum as defined in Sect. S1.1.1 of the Supplement at 1 astronomical unit (AU), weighted with the response functions of the GUVis-3511,

$\rho$  scales the extraterrestrial solar spectrum from 1 AU to the Sun-Earth distance applicable to 21 August 2018,

$\tau_r$  is the Rayleigh optical depth calculated by Eq. (30) of Bodhaine et al (1999),

$\mu_r$  is the Rayleigh airmass factor, calculated according to Kasten and Young (1989),

$\tau_o$  is the ozone optical depth derived from the ozone absorption cross section by Bass. and Paur (1985) and assuming a TOC of 298 DU,

$\mu_o$  is the ozone airmass factor calculated by Eq. (11) of Bernhard et al. (2005b) assuming an ozone layer height of 22 km, and

$\mu_a$  is the aerosol airmass factor calculated by Eq. (11) of Bernhard et al. (2005b) assuming an aerosol layer 2 km above the surface.

Aerosol optical depth $\tau_a$ is reported at the centroid wavelength (Seckmeyer et al., 2010) of each channel.

### 5.2 Total ozone column

The TOC during the eclipse was derived from measurements of response-weighted global irradiance $\hat{E}_{S_g}(\lambda_i)$ measured by the GUVis-3511 (see Supplement for definition of $\hat{E}_{S_g}(\lambda_i)$). Specifically, ratios of $\hat{E}_{S_g}(340)/\hat{E}_{S_g}(305)$ were compared with similar ratios in a look-up table that was calculated with a RT model (also described in the Supplement) as a function of SZA and TOC. This method was first proposed by Stamnes et al. (1991) and was validated for GUV instruments by Bernhard et al. (2005a).

### 6 Celestial calculations

The interpretation of measurements performed during the solar eclipse requires very accurate calculations of the relative distance between Sun and Moon and angular diameter of both celestial bodies. Such calculations are provided by the HORIZONS on-line solar system data and ephemeris computation service of the Jet Propulsion Laboratory (https://ssd.jpl.nasa.gov/?horizons). From the many parameters made available through this system in time intervals of one minute, we used the elevation, azimuth, and angular radius of Sun ($\theta_S$, $\varphi_S$, $\delta_S$) and Moon ($\theta_M$, $\varphi_M$, $\delta_M$), calculated the angular distance $\xi$ between Sun and Moon via

$$\xi = \arccos[\sin(\theta_S)\sin(\theta_M) + \cos(\theta_S)\cos(\theta_M)\cos(\varphi_S - \varphi_M)], \qquad (2)$$

and interpolated these measurements to the times of our measurements (Fig. 5). The eclipse begins and ends when $\xi = \delta_S + \delta_M$, and the maximum eclipse is observed when $\xi$ is at its minimum. The start (1[st] contact), "maximum eclipse",





and end (4th contact) times calculated with this method were compared with solar eclipse data calculated with an online calculator provided by the Astronomical Applications Department of the U.S. Naval Observatory (USNO) at http://aa.usno.navy.mil/data/docs/Eclipse2017.php. Results of our calibration and those by the USNO are compared in Table 1 and agree to within ±3 seconds, giving confidence in our calculations. Note that the period between 1st contact and maximum is about 6.5 minutes shorter than the period between maximum and 4th contact.

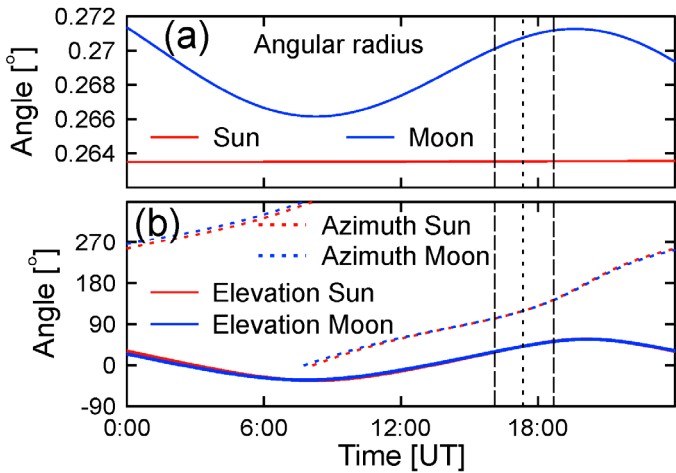

Fig. 5. Panel (a): Apparent angular radii of Sun and Moon on 21 August 2018. The apparent radius of the Sun is virtually constant over time while the radius of the Moon varies considerably during the day due to the rotation of the Earth, which changes the distance between two hypothetical observers standing on the Earth and the Moon. Panel (b): Corresponding solar and lunar elevation and azimuth angles. Note the conversion of the curves at the time of maximum eclipse (short-dashed vertical line). Long-dashed lines indicate the times of the 1st and 4th contact.

Table 1. Comparison of Eclipse calculations by us and USNO.

|  | Time (UT) | | Difference | Solar zenith | Solar Azimuth |
|---|---|---|---|---|---|
|  | Our calculation | USNO | (seconds) | (°) | (°) |
| Start partial eclipse (1st contact) | 16:06:24 | 16:06:27 | –3 | 60.6 | 102.7 |
| Start total eclipse (2nd contact) |  | 17:19:42 |  | 48.4 | 118.9 |
| Maximum eclipse | 17:20:27 | 17:20:24 | 3 | 48.3 | 119.1 |
| End total eclipse (3rd contact) |  | 17:21:07 |  | 48.2 | 119.3 |
| End partial eclipse (4th contact) | 18:41:05 | 18:41:02 | 3 | 37.4 | 143.6 |

The fraction of the solar disk seen by the GUVis-3511 and the change of the extraterrestrial spectrum as function of time during the progression of the eclipse were calculated with the algorithm by Koepke et al. (2001). Koepke et al. (2001) uses a




simple analytical formula to describe the wavelength dependence of solar LD, which only considers the temperature at the Sun's surface (Waldmeier, 1941). We also use two other parameterizations of the LD functions that are based on data collected by the McMath-Pierce Solar Telescope of the National Solar Observatory on Kitt Peak. Specifically, the function $\Gamma_\lambda(r)$ defined in Eq. (2.1) of Koepke et al. (2001) was replaced with the polynomial

$$P(\lambda, r) = \sum_{k=0}^{5} a_k(\lambda) \cos^k(\psi(r)),$$

(3)

5    where the angle of incidence $\psi$ is defined in Fig. 6 and the coefficients $a_k(\lambda)$ were derived from measurements at Kitt Peak. The quantity $\Gamma_\lambda(r)$ used by Koepke et al. (2001) depends on the distance $r$ between a point on the solar disk and the center of Sun, relative to the radius of the Sun, $R_S$ (Fig. 6). The angle $\psi$ was calculated from $r$ using basic trigonometry.

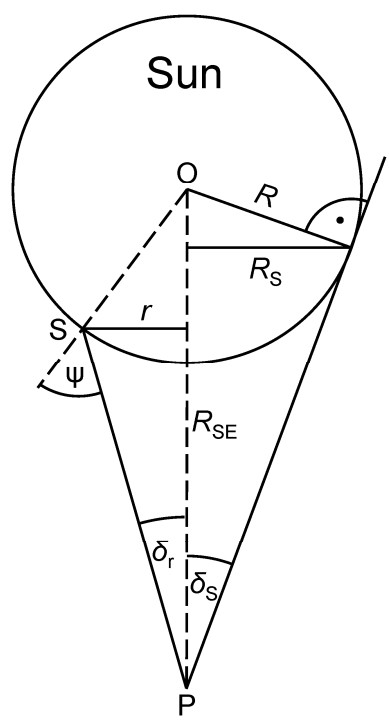

Fig. 6. Limb darkening geometry. The Sun is centered at point O and has the radius $R$. The observer is at point P at a distance $R_{SE}$ from the center of the Sun, and is looking at point S on the surface of the Sun. From the point of view of the observer, S is at an angle $\delta_r$ relative to the center of the Sun, and the Sun's limb is at angle $\delta_S$. The angular radius of the solar disk as viewed by the observer is $R_S$ and the angular radius associated with S is $r$. $\psi$ is the angle between the Sun's normal at S, and the line between S and P.

Two sets of coefficients $a_k(\lambda)$ were used in this study. The first set was developed by Pierce and Slaughter (1977) ('PS') and Pierce et al. (1977) ('PSW') using measurements collected between 1974 and 1976 near a minimum of solar activity. These coefficients are tabulated in Table IV of PS (wavelength range 303.327–729.675 nm) and Table IV of PSW (740.46–



1046.7 nm). The second set was published by Neckel (2005), using measurements performed in 1986 and 1987 near the following solar minimum and these coefficients are listed in Table I of Neckel (2005) (300–1100 nm). Both sets of data were collected with the observatory's large vertical spectrograph, which consists of a double-monochromator with prism pre-disperser. Measurements across the solar disk were taken by stopping the telescope drive to let the Sun pass across the

entrance port of the radiometer.

We note that different parameterizations are used by Neckel (2005) for the wavelength ranges 300.00–372.98 nm, 385.00–422.57 nm, and 422.57–1100 nm, but no parameterization is provided for the range of 372.98–385.00 nm because this range is affected by the Balmer "jump" in the absorption of the hydrogen atom resulting in a large uncertainty in the solar LD

function in this wavelength range. We compare our measurements at 380 nm with the parameterization developed for the 385.00–422.57 nm range.

Fig. 7 compares results of the three parameterizations, referred to in the following as the parameterizations by Waldmeier, Pierce, and Neckel. Fig. 7a is based on the parameterization by Pierce and shows ratios of the extraterrestrial solar spectral

irradiance during the eclipse to the spectral irradiance that would be expected if the Sun were a uniformly bright star. Results are plotted versus the fraction of the Sun's area that is visible (0 = Sun completely occluded by Moon, 1 = Sun completely visible). The LD effect is strongly dependent on wavelength: when the Sun is almost completely eclipsed, the spectral irradiance at 306 nm is reduced to about 22 % of the intensity that would be expected from purely geometrical considerations, whereas the spectral irradiance at 1020 nm is only reduced to about 67 %. Figure 9a also indicates that the

LD effect is slightly positive when 66 % or more of the Sun's area are visible. In these cases, the Moon occludes the darker, outer parts of the Sun, and the average intensity of the visible area is therefore larger than that of a hypothetical, uniformly bright star.

Fig. 7b shows similar results but is based on the parameterization by Waldmeier. For wavelengths larger than 500 nm, the

LD effect is similar to that calculated with the parameterization by Pierce. For smaller wavelengths, results derived with the parameterization by Waldmeier indicate a much weaker LD effect.

Fig. 7c shows the ratio of the LD effect calculated with the parameterization by Neckel and Pierce. Results for the two parameterizations agree to within ±2.3 % when 5 % or more of the Sun is visible. The largest difference of 7.6 % is observed

at 395 nm when only 0.6 % of the Sun are still visible. On the other hand, results between the Waldmeier and Pierce parameterizations differ by as much as a factor of 1.78 (Fig. 7d).



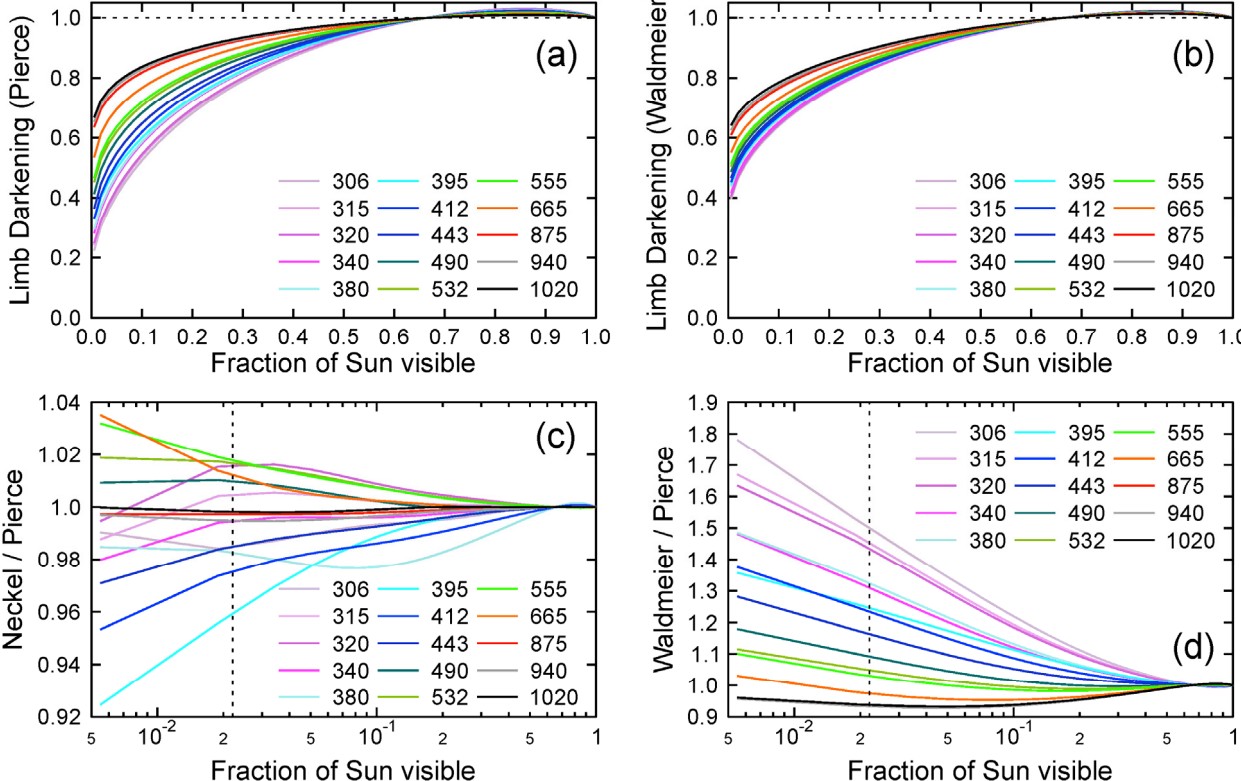

Fig. 7. Comparison of the solar LD effect calculated with the parameterizations by Pierce, Neckel and Waldmeier. Top row: Ratios of the extraterrestrial solar spectral irradiance during the eclipse to the spectral irradiance from a hypothetical Sun without LD, plotted versus the fraction of the Sun that is visible. Panels (a) and (b) are based on the parameterizations by Pierce and Waldmeier, respectively. Bottom row: Ratios of LD effect calculated by the parameterizations by Neckel and Pierce (Panel c) and by Waldmeier and Pierce (Panel d). The vertical broken line indicates the minimum fraction of 0.022 (2.2 % of the Sun's disk) that could be assessed with our measurements.

## 7 Results

In this section, we show the AOD and TOC derived from our measurements before, during and after the eclipse, present the ratio of direct-to-global irradiance, validate the solar LD parameterizations by the three methods discussed in Sect. 6, and compare measurements near and during totality with reconstructed measurements for the same period but without the Moon occluding the Sun.





## 7.1 Aerosol optical depth

AOD was calculated with Eq. (1) from direct spectral irradiance, which in turn was derived from data collected during shadowbanding. The standard algorithm computing AODs does not take into account that a solar eclipse is occurring and (incorrectly) attributes the reduced direct irradiance during the eclipse to increased extinction of radiation by aerosols. The

5      result (thin lines in Fig. 8a) is the large spike in AOD, peaking at the time of totality. Calculations were repeated by first scaling the direct spectral irradiance with the LD effect using the parameterization by Pierce (Sect. 6) and then feeding the scaled irradiances in the AOD algorithm. Results corrected for the LD effect (thick lines in Fig. 8a) indicate that the AOD was monotonically decreasing over the period of the eclipse (from 0.41 to 0.32 at 319 nm and from 0.05 to 0.04 at 1018 nm) without a spurious spike near the time of totality.

AOD measurements before and after the eclipse were used to characterize the aerosol loading of the atmosphere. Before 15:45, AODs were somewhat variable due to aerosols from nearby wildfires. Fig. 8b shows AOD as a function of wavelength shortly before (16:02:58) and after the eclipse (19:00:30). AODs were somewhat lower after the eclipse compared to the start. An Ångström function of the type $\tau_a = \beta \lambda^{-\alpha}$ , where the wavelength $\lambda$ is provided in µm was fitted

15      to the data using channels between 340 and 1020 nm, excluding the channel at 940 nm. Values of the Ångström coefficient were $\alpha = 1.96$ and $\beta = 0.057$ for the measurement at 16:02:58, and $\alpha = 2.1$ and $\beta = 0.0394$ for the measurement at 19:00:30. The second set of coefficients was used for the RT calculations discussed in Sect. S1.1.1 of the Supplement. Ångström exponents $\alpha$ in the order of 2.0 may seem large, however, O'Neil et al. (2002) have demonstrated that values of $\alpha = 2$ are typical for aerosols originating from forest fires. Deviations of the measured AODs from the Ångström parameterizations

20      shown in Fig. 8b are smaller than 0.02, with the exception of the AODs at 314.3, 319.4, and 442.4 nm. Measured AODs are up to 0.15 smaller than the parameterization at these wavelengths. The reason for these discrepancies is unknown, however, differences between AODs at the three wavelengths relative to AODs at adjacent wavelengths are larger at 19:00:30 than 16:02:58, suggesting that the cause is an unknown absorber and not due to a measurement artifact.



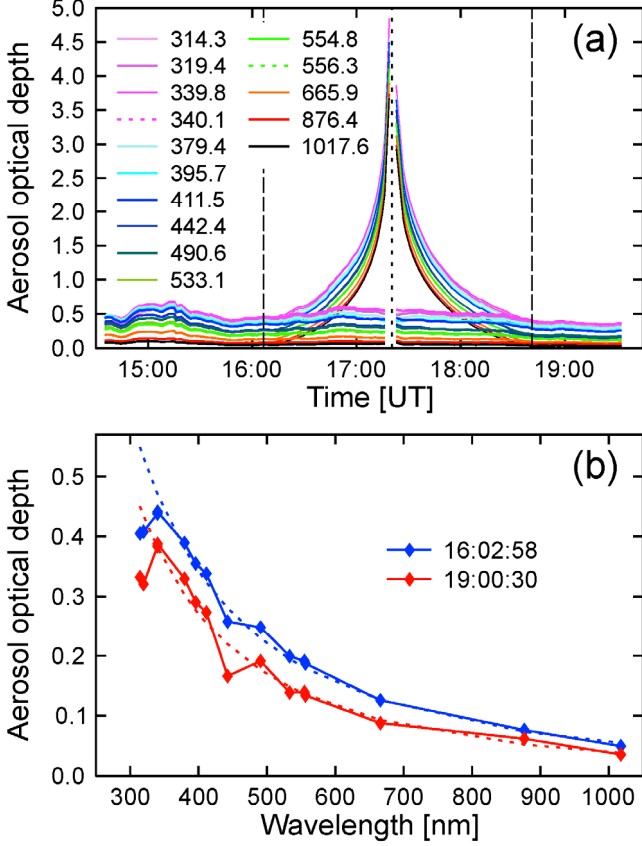

Fig. 8. Aerosol optical depth. Panel (a): AOD measured by the GUVis-3511 radiometer at Smith Rock as a function of time. Thin lines indicate AOD measured without LD correction while thick lines indicate AOD corrected using the LD parameterization by Pierce. Long-dashed lines indicate the start and end times of the eclipse and the short-dashed line indicates the time at totality. Without LD correction, the reduced irradiance during the eclipse period is incorrectly attributed to aerosols, leading to a spurious spike at the time of totality. With LD correction, the effect of the eclipse is barely noticeable in the AOD data. The legend indicates the centroid wavelengths of the GUV's channels. Panel (b): Aerosol optical depth as a function of wavelength shortly before (16:02:58) and after the eclipse (19:00:30). Dashed lines indicate Ångström function fits to the data based on measurements of the AODs between 339.8 and 1017.6 nm.

## 7.2 Total ozone column

Total ozone column was calculated from GUVis-3511 measurements of the 305 and 340 nm channels as described in Sect. 5.2. Data of the two channels were first processed without taking into account that the solar LD effects reduces the extraterrestrial spectrum more strongly in the UV-B than at 340 nm. Processing was repeated by scaling the raw data with the three parameterizations of the LD effect described in Sect. 6. Fig. 9 compares these results with measurements by OMI. (Of note, OMI data for 21 August 2018 are not available for the pixel that contains our measurement site. However, the





average of OMI measurement within ±1° in latitude and ±5° in longitude is 297.9 DU with a standard deviation of 2.1 DU.)
The TOC measured by the GUVis-3511 was 293 DU shortly before the start of the eclipse and 294 DU shortly after the end,
suggesting that the eclipse had no lasting effect on atmospheric ozone concentrations. Measured TOCs without LD
correction spiked during the eclipse, resulting in 316 DU shortly before totality. When correcting the data using the LD

parameterization by Waldmeier, the height of the peak is reduced to 312 nm (a 1.3 % reduction). When using either the
parameterizations by Neckel or Pierce, the effect of the eclipse on measured TOCs almost disappears, in particular for the
period between the 3$^{rd}$ and 4$^{th}$ contact. (We have no explanation for the increase in TOC of about 5 DU between 16:00 and
17:00 other than natural variability). The results shown in Fig. 9 can be explained with the wavelength dependence of LD.
The stronger reduction of the extraterrestrial irradiance at 306 nm compared to 340 nm caused by the LD effect is interpreted

by the algorithm as a larger TOC. Once the LD effect is removed, the TOC remains constant to within ±2.6 DU or ±0.9 %.

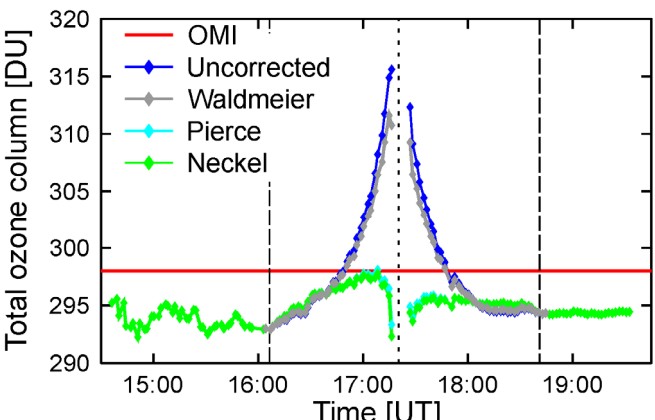

Fig. 9. Total ozone column measured by GUVis-3511 and OMI. For "uncorrected" data, the spike in TOC during the period
of the eclipse is an artifact that can be attributed to solar LD. The effect is only marginally reduced when using the LD

correction based on the parameterization by Waldmeier. By using either the parameterizations by Pierce or Neckel,
variability in TOC during the eclipse is reduced to ±2.6 DU (±0.9 %).

**7.3 Direct-to-global ratio**

Fig. 10 shows the ratio of direct spectral irradiance $\overline{E}_{S_d}(\lambda_i)$ (derived from shadowband data) and global spectral irradiance
$\overline{E}_{Sg}(\lambda_i)$ (see Supplement for definition of symbols). The general upward slope is due to the change in SZA between

morning and noon, and the steep slope between 15:15 and 17:45 is due to a decrease in aerosols as indicated by Fig. 8. Thin
lines in Fig. 10 connect measurements at the start of end of the eclipse and are drawn to guide the eye. Measured ratios tend
to be below these lines by ratio values of up to 0.04, and these low ratio values could either be caused by processes initiated
by the occlusion of the Sun or by variability from aerosols. (A change by 0.04 is well within the range of aerosol effects



occurring before 15:30). Aerosols effects are more likely because any eclipse effects should conspicuously peak at the time of totality as it was the case for the spurious peaks in AOD (Fig. 8) and TOC (Fig. 9) measurements. Note that direct measurements are not available between 200 s before the 2nd contact and 105 seconds after the 3rd contact.

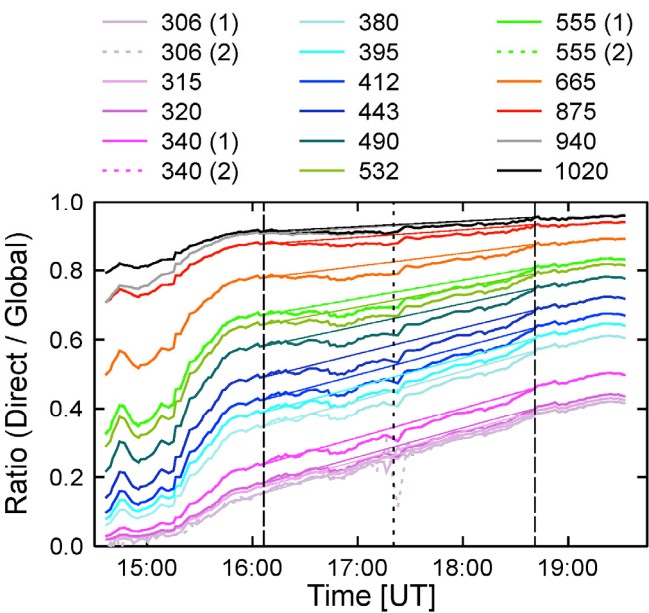

Fig. 10. Ratio of direct-to-global spectral irradiance. Measurements are shown as heavy lines. Thin lines connect measurements at the start of end of the eclipse and are drawn to guide the eye. (1) and (2) indicate the channel number for channels equipped with identical wavelengths.

**7.4 Effect of solar limb darkening**

Theoretical predictions of the solar LD effect described in Sect. 5 were compared with the LD effect derived from our measurements (Fig. 11). To account for the change in SZA, data were first compared with results of the RT model described in the Supplement. Model input parameters (e.g., AOD) are based on the measurement at 19:00:30 (shortly after the end of the eclipse) as described in Sect. 7.1. The only model parameter changing as a function of time is the SZA. Fig. 11a shows a comparison of measurements of global spectral irradiance (symbols) and these model calculations (thin lines). The

agreement between measurement and model for times outside the period of the eclipse is reasonable, but not perfect. The ratio of measurement and model is shown in Fig. 11b. At 19:00:30 (i.e., the time relevant for the input parameters of the RT calculations) the ratio varies between 0.958 and 1.027, except for the channels at 313 and 320 nm (where the ratio is 0.932), and at 940 nm (ratio of 1.139) due to the uncertainty of the water vapor column affecting this wavelength. The discrepancy between measurement and model at the start of the eclipse is generally larger, likely due to the change in AOD, which was

not considered by the model. To correct for the difference between measurement and model, linear functions were





constructed to match the ratio of measurement and model at the times of the 1st and 4th contact (thin lines in Fig. 11b). This correction assumes that changes in the atmospheric constituents (aerosols, ozone, etc.) led to a linear change in the atmospheric transmission over the period of the eclipse. Next, ratios of measurement and model were normalized by dividing with these linear functions. The resulting bias-corrected ratios represent the LD effect derived from our measurements

(symbols in Fig. 11c). Thin lines in that figure show the theoretical LD effect calculated with the parameterization by Pierce. The insert of Fig. 11c shows both datasets for the time between 17:01:00 and 17:04:30, indicating that there is almost perfect agreement for wavelengths larger than 400 nm, but measurements at UV wavelengths are below the theoretical prediction.

Lastly, the measured LD effect was divided by the theoretical data and the resulting ratios are shown in Fig. 11d. For
wavelengths larger than 400 nm, measurement and theory agree better than ±4.0 %. When excluding the period between 16:44 and 17:00, which was likely affected by aerosols, the agreement is to within ±2.5 %. For times outside of ±7 minutes from totality, measurement and theory for 380 and 395 nm also agree within this range. The larger discrepancy closer to totality is likely caused by the uncertainty of the theoretical prediction due to the Balmer "jump" affecting these wavelengths. The discrepancy between measurement and theory for wavelengths below 380 nm (306–340 nm) becomes
larger than 3 % as totality is approached, with the smallest wavelengths showing the largest change (e.g., the ratio is 0.90 for 306 nm at 17:01). For these wavelengths, the discrepancy is generally larger before totality than after totality. We do not know the reason for this asymmetry but note that the measured LD effect is lower than the theoretical prediction both for wavelengths affected by ozone absorption (306, 315, and 320 nm) and wavelengths not affected by ozone absorption (340 nm). Hence, the larger discrepancy at these UV wavelengths (and the asymmetry relative to totality) cannot solely be caused
by ozone absorption in the atmosphere. Results for wavelengths that were measured independently by two channels (306, 340, and 555 nm) are virtually indistinguishable, suggesting that calibrations were applied consistently and that both types of photodiodes used for these channels led to practically identical results.

Fig. 12 compares the measured LD effect with the theoretical prediction based on the parameterization by Neckel instead of
Pierce. Results shown in Fig. 11d and Fig. 12 are very similar and differences are within the uncertainties of the measurements and the change of the atmospheric extinction due to the variability of the aerosol load. Despite the similarity of the two figures, there is one difference worth noting. For 380 and 395 nm, discrepancies calculated for the parameterization by Neckel are larger than those derived from the parameterizations by Pierce. This is not surprising considering that Neckel does not provide coefficients for the wavelength range of 372.98–385.00 nm because of the
uncertainty from the Balmer jump. As described in Sect. 5, our parameterization for the LD effect at 380 nm is based on Neckel's coefficients for the 385.00–422.57 nm range and therefore subject to uncertainties.





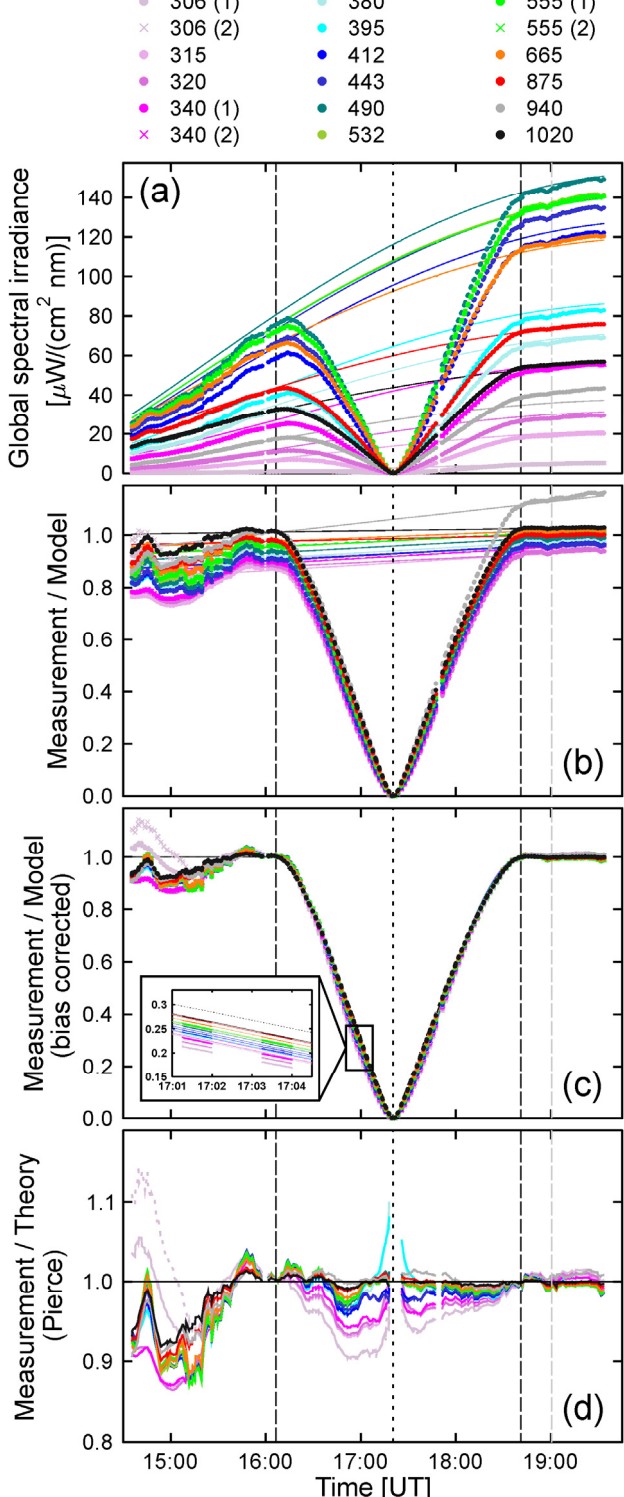

Fig. 11. Comparison of the solar LD effect derived from measurements of global spectral with theoretical predictions based on the parameterizations by Pierce. Panel (a): measurements of global spectral irradiance (symbols) and results of the RT calculations described in Sect. S1.1.1 of the Supplement (lines). Only every 20[th] data point is plotted for clarity. Panel (b): ratio of measurement and model (symbols). Lines connect the ratios at the times of the 1[st] and 4[th] contact, which are indicated by the vertical long-dashed lines. Panel (c): Ratio of measurement and model, corrected for the bias between measurement and model (symbols). The insert shows this ratio for times between 17:01:00 and 17:04:30. The theoretical LD effect calculated with the parameterizations by Pierce is also shown and indicated with lines. Panel (d): ratio of biased-corrected measurements (symbols in panel (c)) and theoretical prediction (lines in panel (c)). Ratios for the alternative channels at 306, 340, and 555 nm are indicated with broken lines and are virtually indistinguishable from those using the standard photodiode, which are indicated by solid lines. The time of maximum eclipse is indicated by the short-dashed black line. The dashed gray line specifies the time of 19:00:30 for which the AOD used by the model calculations was derived. The legend on top of the figure indicates spectral irradiance in nm.



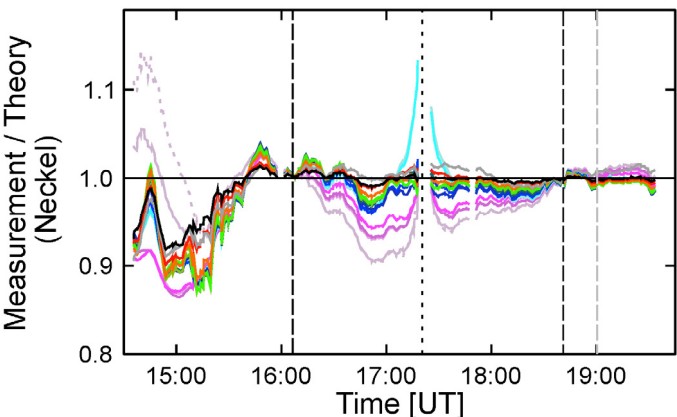

Fig. 12. Similar to Fig. 11d, but theoretical LD effect parameterized based on Neckel (2005) instead of Pierce.

Fig. 13 shows the ratio of the LD effect calculated with the parameterizations by Neckel and Pierce. The figure is based on the same data as those used for Fig. 7c, but are plotted versus time instead of the fraction of the Sun not obstructed by the Moon. Results of the parameterization are virtually identical if 60 % or more of the Sun is visible. Discrepancies increase as the Sun becomes more and more occluded and are largest at 380 and 395 nm, with a maximum difference of about 3 %.

10 There is generally no systematic dependence on wavelength.

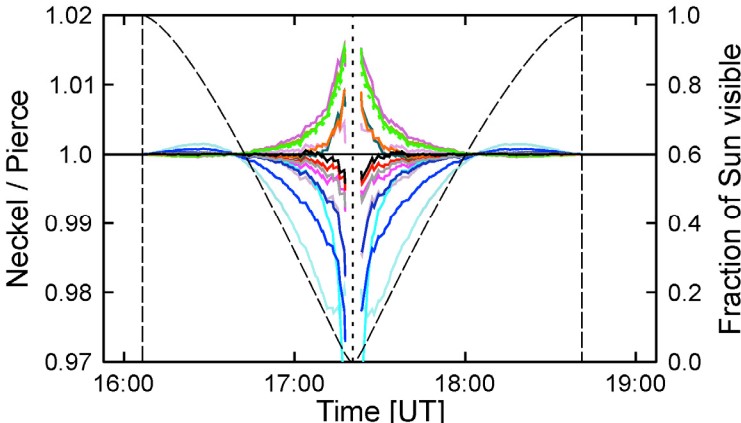

Fig. 13. Ratio of the LD effect calculated with the parameterizations by Neckel and Pierce. The color coding is identical with that used in Fig. 11. The dashed line indicates the fraction of the Sun's disk that is visible and is referenced to the right axis.

15 Vertical dashed lines indicate the 1st and 4th contact.





Lastly, when we compared our observations with theoretical prediction based on the parameterization by Waldmeier, observation were lower than theory by up to 37 % at 306 nm, 24 % at 340nm, and 19 % at 412 nm. For longer wavelengths, differences were in the ±8 % range. These differences are well outside the uncertainty of our measurements, confirming that the parameterization by Waldmeier is too simple, in particular in the UV range.

**7.5 Measurements near and during totality**

Fig. 14 shows measurements between about 1.5 minutes before the 2nd contact and about 2 minutes after the 3rd contact. The measurement protocol was not changed during this period and the radiometer was therefore shadowbanding during a large part of totality (between 17:20:05 and 17:21:11). Periods when shadowbanding was active are indicated by grey shading in Fig. 14. At 17:18:53 (before 2nd contact) and 17:22:53 (after 3rd contact) the shadowband blocked the direct Sun, leading to a

strong decreases of the measured spectral irradiance. In contrast, the effect of the shadowband is barely noticeable during totality, although the small dip in the spectral irradiance between 17:20:10 and 17:20:40 can be attributed to the blocking of sky light by the band. Shadowband data are considerably noisier than global irradiance measurements because of the difference in integration times (1 second for global irradiance and 1/15 second for shadowband measurements). To compensate for this effect, shadowband data shown in Fig. 14 are a running average of 15 measurements. In addition,

shadowband measurements were scaled with a cosine error correction factor calculated for isotropic incident radiation, but were not corrected for the shading of skylight by the band.

Measurements of global spectral irradiance are available between 17:19:15, through the 2nd contact, and until 17:20:20. These measurements were cosine-error corrected without taking into account the change in the direct-to-global ratio at the

beginning of totality. Despite this approximation, there is no obvious discontinuity at 17:20:20 when global irradiance measurements transition into shadowband data. During the last half-minute before the 2nd contact, spectral irradiances in the UV and visible decrease by about one order for magnitude. The decrease in the IR is considerably larger with a change of about 2 and 2.5 orders of magnitude at 1020 nm and 940 nm, respectively. The wavelength dependence of these changes can be explained with the wavelength-dependence of Rayleigh scattering, which makes it less likely for photons entering the

atmosphere outside the Moon's shadow to travel to the area of the umbra. The long pathlength also amplifies absorption by gases (ozone, $NO_2$, $H_2O$) and aerosols. Spectral irradiance at 940 nm, which is affected by water vapor absorption, decreases therefore more rapidly than spectral irradiance at 1020 nm, which is absorbed much less. Measurements at 306 nm are below the detection limit during totality.



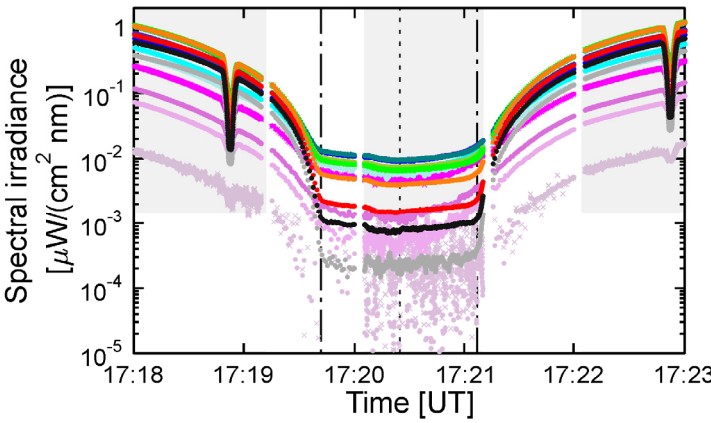

Fig. 14. Spectral irradiance measured between about 1.5 minutes before the 2nd contact and about 2 minutes after the 3rd contact. The color coding is identical with that used in Fig. 11. Grey shading indicates times of shadowbanding. The times of

the 2nd and 3rd contact are indicated by dot-dashed lines and the short-dashed line marks the time of maximum totality.

The difference of the global spectral irradiance observed during totality relative to the unoccluded Sun was quantified with the data shown in Fig. 11c. Specifically, the bias-corrected ratio of measurement and model averaged over the period of 17:19:50 and 17:20:00 (i.e., the last 10 seconds of global spectral irradiance measurements during totality before the start of

the shadowbanding sequence) were compared to the respective ratio at the 4th contact, which is one by design. Fig. 15 shows the so-calculated reduction factors. Factors decrease strongly between 306 and 340 nm, followed by a gradual increase from 340 nm to 1020 nm. Spectral irradiance is reduced by a factors of 18,700 at 315 nm, 7,000 at 340 nm, 18,340 at 665 nm, 136,650 at 940 nm and 46,150 at 1020 nm. The relatively high factors in the UV-B and at 940 nm are caused by ozone and water vapor absorption, respectively. The average reduction for the channels in the visible (412–665 nm) is 11,460. For

comparison, the irradiance from the full Moon is about a factor of 400,000 smaller than that of the Sun (Macdonald, 2012). Hence, the visible irradiance during totality is still a factor of 35 larger than the irradiance from the full Moon. This estimate is in agreement with empirical observations. For example, the light during totality is still bright enough to perceive colors while this is barely possible with moonlight.





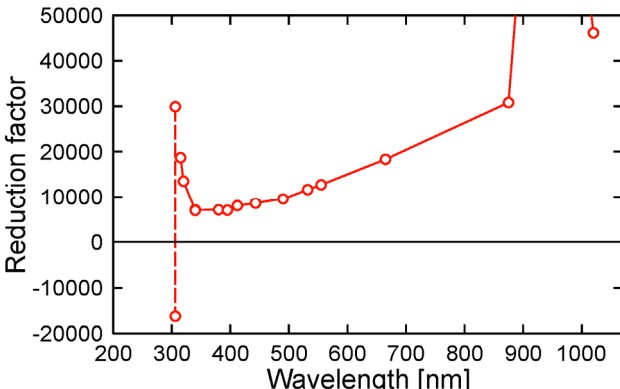

Fig. 15. Factor quantify the reduction of spectral irradiance during totality relative to spectral irradiance from the unoccluded Sun. Measurements of the two channels at 306 nm are below the detection limit during totality and the factors calculated for two channels (connected by a dashed line) are therefore greatly different and meaningless. Factors for the other wavelengths with two channels (340 and 555 nm) are virtually identical. The reduction factor for 940 nm is 136,650.

## 8 Discussion

Some of the findings of this paper are in conflict with results from similar publications discussing solar eclipses that have occurred in the past. These pertain to changes of the diffuse-to-direct ratio observed during an eclipse, the magnitude of LD correction, and the TOC variations during an eclipse. The three issues are discussed below, followed by a discussion of the value of eclipse data for validating the performance of radiometers.

### 8.1 Variations of direct-to-global ratio

The direct-to-global ratios shown in Fig. 10 increase with time as expected from the decrease in SZA. Deviations from a straight line between the 1st and 4th contact are less than 0.04. It is difficult to determine whether this deviation is caused by processes initiated by the occlusion of the Sun, by variability from aerosols, or artifacts of the algorithm to calculate the direct irradiance from shadowband data. We therefore consider the deviation of 0.04 as an upper limit for the eclipse effect. According to theoretical calculations by Emde and Mayer (2007), eclipse effects that affect direct and diffuse radiation differently are smaller than 1 % for times 10 minutes away from totality and smaller than 4 % for times 105 seconds away from totality (i.e., the shortest time to totality available from our shadowband measurements). Our results as well as these theoretical calculations also agree with the conclusion by Sharp et al., (1971) that "sky light may be considered as attenuated sunlight up to at least 99.8 % obscuration." However, these results disagree with the results by Zerefos et al. (2000; 2001), who suggest that the diffuse irradiance in the UV is declining at slower rate than the direct irradiance. When 90 % of the Sun are occluded (i.e., the maximum coverage for the eclipse discussed by Zerefos et al. (2000)), diffuse irradiance is reduced 30



% less than direct irradiance. The available evidence from our observations suggests that the large change of the diffuse-to-direct ratio reported by Zerefos et al. (2000; 2001) was a measurement artifact.

## 8.2 Magnitude of solar limb darkening

The excellent agreement (better than 4.0 % overall and better than 2.5 % excluding the period affected by aerosols) between
the solar LD effect derived from our measurements and theoretical predictions for wavelengths above 400 nm suggests that the parameterizations of LD by Pierce and Neckel are correct to within the measurement uncertainty for this wavelength range. The larger discrepancies below 400 nm could be due to systematic errors in these parameterization. Specifically, Neckel uses different coefficients for the wavelengths ranges 300–372.98 nm, 385.00–422.57 nm and 422.57–1100 nm because of the Balmer jump. The electronic transitions from the 2$^{nd}$ atomic shell to shells higher than 6 occur at wavelengths
between 410.2 (transition 2 > 6) and 364.6 nm (Balmer break). The hypothesis of possible systematic errors in the LD data by Neckel is supported by the relatively large noise in data at UV wavelengths presented by Neckel (2005), which formed the basis of LD parameterizations used here. On the other hand, the ratio of the measured and theoretical LD effect (Fig. 11) shows some asymmetry for wavelengths below 380 nm and these could be caused by measurement artifacts of unknown origin.

Our results also indicate that the LD parameterization by Waldmeier is too simple to accurately describe the change of the radiance across the solar disk, in particular in the UV range. Several recent papers (e.g., Blumthaler et al., 2006; Emde and Mayer, 2007; Kazadzis et al., 2007) have used the method by Koepke et al. (2001) combined with the LD darkening parameterization by Waldmeier. Some conclusions in those papers should therefore be reconsidered. For example, Kazadzis
et al. (2007) calculated the TOC during the total eclipse of 29 March 2006 at Kastelorizo, Greece, from direct irradiance measurements of a Brewer spectrophotometer. By combining the method by Koepke et al. (2001) with the parameterization by Waldmeier (1941), they calculated negligible LD corrections of less than 0.01 % for their ozone retrievals. (Of note, the maximum correction using the parameterization by Waldmeier for our data (grey line in Fig. 9) is 1.3 %, which is much larger than the correction of 0.01 % quoted by Kazadzis et al. (2007). The reason for this discrepancy is unknown, however,
the wavelengths used in our TOC retrieval (306 and 340 nm) are different from the Brewer wavelengths, which range between 306.2 and 320.1 nm.)

In contrast to the *increase* that is apparent in our uncorrected measurements shown in Fig. 9, the ozone retrievals by Kazadzis et al. (2007) show a *decrease* in TOC during the eclipse. Kazadzis et al. (2007) attribute this apparent reduction in
TOC to contamination of the Brewer's direct measurements by diffuse radiation in the instrument's field of view, quoting a previous conclusion by Zerefos et al. (2000; 2001). We find this conclusion surprising considering that the direct-to-diffuse ratio does virtually not change during the eclipse (both according to our measurements and the theoretical calculations by Emde and Mayer (2007)), with the exception of a short (< 5 min) period before and after the period of totality. Using a LD



correction method based on the NLTE Spectral Synthesis code (Tagirov et al. 2016), Gröbner et al. (2017) provide additional evidence that the systematic bias in Brewer TOC measurements is primarily due to LD effects. Of note, the LD correction applied to Dobson spectrophotometer data from the solar eclipse of 20 May 1966 discussed of Bojkov (1968) reduces the TOC by up to 6 %, in good agreement with our calculations.

## 8.3 Total ozone column variations during an eclipse

Chimonas (1970) hypothesized that the Moon's shadow, as it travels at supersonic speeds across the atmosphere during a solar eclipse, induces "bow waves" (a type of gravity waves) due to sudden cooling of the atmosphere at altitudes between 10 and 60 km, where the ozone layer normally converts solar UV radiation to heat. Bow waves with wavelengths between of 300 and 400 km and a period of about 25 minutes have indeed been observed during the solar eclipse of 21 August 2017 (Zhang et al., 2017).

Using data measured during the solar eclipse of 11 August 1999, Zerefos et al. (2000) observed an oscillation with a period of 20 minutes in erythemal (sunburning) irradiance before and after the time of the eclipse with a peak-to-peak variation of about 1 %. Zerefos et al. (2000) attributed these oscillation to fluctuations in total ozone caused by eclipse-induced gravity waves. A variation of ±0.5 % in erythemal irradiance would require a variation in total ozone of similar magnitude (McKenzie et al., 2011). In a similar analysis, using data from the 29 March 2006 solar eclipse, Zerefos et al. (2007) reports oscillations in TOC with a peak-to-peak amplitude of 2.0–3.5 % and periods ranging between 30 and 40 minutes. However, these measurements also indicate an unrealistic drop in the TOC from 295 to 225 DU between the beginning and the maximum of the eclipse, followed by an increase to 300 DU. This artifact is attributed to the contamination of direct irradiance measurements (which are used for the ozone retrieval) by diffuse radiation. It is concerning that the magnitude of this artifact (75 DU or 25 %) exceeds the magnitude attributed to the bow wave by a factor of 10. Finally, Mims and Mims (1993) report that TOC measurements taken during the total solar eclipse of 11 July 1991 show fluctuations in ozone with a peak-to-peak amplitude of up to 5 DU (1.7 %), which began 700 seconds after the third contact. However also these observations were affected by incomplete LD correction, resulting in TOC changes of 26 DU (about 9 %) over the course of the eclipse.

Our data do not support the observation by Zerefos (2000; 2007) and Mims and Mims (1993) that bow waves from the Moon's shadow lead to oscillations in TOC. For example, between 17:36 and 18:36, LD-corrected TOC data decreased by 1 DU (0.33 %) with no obvious oscillations. Between 18:38 and the end of the measurements at 19:32, TOC remained constant to within 0.3 DU (0.10 %). These small variations are well within the natural variability of the TOC. The lack of an effect is not surprising considering that diurnal variations in ozone are small. For example, at the Mauna Loa observatory (19.5° S), the day/night difference of the ozone profile is smaller than ±2 % for altitudes up to 45 km where 99 % of the ozone column is located (Parrish et al., 2014). Since day/night changes below and above 30 km have a different sign and partly cancel each



other, the effect on TOC is negligible. It would be surprising if TOC variations over the relatively short period of a solar eclipse driven by bow waves were larger than variations driven by the diurnal cycle. Despite the results by Zerefos (2000; 2007) and Mims and Mims (1993), we feel that the question of whether or not bow waves from the Moon's shadow can lead to variations in TOC is still up for debate. This debate could be settled by performing LD-corrected measurements of TOC

with different instrument types during one of the upcoming solar eclipses. If such measurements were to show fluctuations in TOC with the same magnitude and timing, the effect of bow waves on TOC could be convincingly demonstrated.

## 8.4 Validation of GUVis-3511 measurements

The GUVis-3511 is a relatively new instrument and the measurements discussed here present a unique opportunity to examine some aspects of the instrument's performance. For example, as radiation levels decrease and increase by more than

four orders of magnitude over the course of the eclipse (Fig. 15), the linearity of the measurements can be assessed. The GUVis-3511 uses a 24-bit amplifier with three gains, providing a dynamic range of over 11 orders of magnitude. Measurements of channels with wavelength larger than 400 nm change by 8.5 orders of magnitude between dark measurements (collector of the instrument capped) and full sunlight, and at some point, have to switch between "high" and "medium" gain to extend the ADC's dynamic range of about 7 orders of magnitude. Non-linearities have been noted for

legacy instruments when the gain changes and may manifest itself as steps in the ratio of the instrument's measurements and a smoothly varying reference. Because every channel changes the gain at a different time, such step-changes would also occur at different times. While ratios of measurement and theory shown in Fig. 11d and Fig. 12 do exhibit some step changes of up to 0.5 % (e.g., when global measurements resume after shadowbanding), such steps occur simultaneously for all channels. We estimate from this analysis that the non-linearity caused by gain changes are smaller than 0.1 % and there is

also no evidence of non-linearity at other times.

Measurements during totality (Fig. 14) suggest that the detection limit of the GUVis-3511 radiometer is about $10^{-4}$ μW cm$^{-2}$ nm$^{-1}$. This detection limit is comparable with that of scanning spectroradiometers with PMT detector, which are considered the most accurate instruments for observing the solar irradiance at Earth's surface (Seckmeyer et al., 2001). While multi-

channel filter radiometer such as the GUVis-3511 do not have the wavelength resolution of these spectroradiometers, they have a superior sampling rate than these instruments (e.g., 1 Hz versus five minutes for one spectrum), and are therefore better suited to study fast-changing phenomena such as the rapid change in irradiance close to the 2$^{nd}$ or 3$^{rd}$ contact.

## 9 Conclusions and outlook

Measurements of spectral irradiance during the total solar eclipse of 21 August 2017 were used to validate parameterizations

of solar LD and to assess the effects of the eclipse on AOD, TOC, and the ratio of direct-to-global irradiance. For wavelengths longer than 400 nm, the change of global spectral irradiance over the period of the eclipse agrees to within ±4.0



% with theoretical predictions that were derived from two independent parameterizations of solar LD by Pierce and Neckel. When excluding a 16-minute period, which was likely affected by aerosols, the agreement is to within ±2.5 %. Between 315 and 400 nm, differences between observation and theory are smaller than 5.5 % and increase to about 9 % at 306 nm. In the visible and infrared ranges, these differences are similar in magnitude before and after totality. For wavelengths between 306 and 340 nm, the difference is larger between the 1st and 2nd contact than between the 3rd and 4th contact. A part of this asymmetry could be caused by changing aerosol conditions.

Several recent papers (e.g., Blumthaler et al., 2006; Emde and Mayer, 2007; Kazadzis et al., 2007) have used the LD parameterization by Waldmeier, which does not consider the wavelength dependence of hydrogen absorption in the Sun's photosphere. When we compared our observations with data derived from this parameterization, observation were lower than theory by up to 37 % at 306 nm and 19 % at 412 nm. For longer wavelengths, differences are in the ±8 % range. These differences are well beyond the uncertainty of our observations, suggesting that the parameterization by Waldmeier is too simple and should not be used.

Results shown in Fig. 9 illustrate that accurate TOC measurements during a solar eclipse can only be achieved if the wavelength-dependence of LD is appropriately corrected. Without correction, TOC values increase with increasing solar occultation and are too high by 23 DU (or 8.2 %) shortly before the 2nd contact. Using the LD correction by Waldmeier, this bias is reduced to 19 DU. When data are corrected using the parameterization by either Pierce or Neckel, TOC measurements stay constant to within ±2.6 DU during the duration of the eclipse. Our conclusion that the LD correction is important contradicts findings of several recent papers, which determined that the LD effect is either less than 0.01 % (Blumthaler et al., 2006; Kazadzis et al., 2007), 1 % (Zerefos et al., 2000), or less than 1.6 % (Kazantzidis et al., 2007). As a result, TOC measurements reported in these papers show spurious changes over the course of an eclipse.

Our data do not support the conclusions by Zerefos (2000; 2007) that gravity or bow waves, which are set off by the Moon's shadow moving at supersonic over the atmosphere, lead to variations the TOC. For example, variations in TOC are smaller than ±1.0 DU (±0.33 %) between the 3rd and 4th contact and smaller than ±0.15 DU (±0.05 %) between the 4th contact and the end of our measurements. These variations are well within the natural variability and the uncertainty of our measurements.

There is no clear evidence that the solar eclipse drives changes in the ratio of direct-to-global irradiance for the period accessible from our measurements (i.e., up to 200 s before the 2nd contact and 105 seconds after the 3rd contact). The observed small changes in this ratio can be explained with variability from aerosols stemming from wildfires. The invariance of this ratio is expected from theory (Emde and Mayer 2007), but contradicts results by Zerefos et al. (2000), who

determined that the ratio of diffuse-to-direct radiation increases gradually by 30 % during the eclipse studied by Zerefos et al. (2000).

During totality, the global spectral irradiance was decreased by factors of 18,700 at 315 nm, 7,000 at 340 nm, 18,340 at 665 nm, 136,650 at 940 nm and 46,150 at 1020 nm. The average reduction for channels in the visible (412–665 nm) is 11,460, which is about a factor of 35 smaller than the ratio of about 400,000 between the irradiance from the Sun and the full Moon.

Results also confirmed that the GUVis-3511 radiometer is an excellent instrument for observing the fast-changing radiation levels during an eclipse with a detection limit of $10^{-4}$ $\mu$W cm$^{-2}$ nm$^{-1}$. By implementing a new calibration method developed
for the specific conditions during the eclipse, measurements with low uncertainty (e.g., 2.7 % for wavelengths between 320 and 875 nm) can be achieved. Comparisons between measurement and theory did not indicate any signs of non-linearity over the dynamic range of about four orders of magnitude between full sunlight and totality.

Lastly, data from the campaign are available as Supplement and can be used to study the rapid change in irradiance during
the transition from a partial to a total eclipse with the help of 3DRT models like that developed by Emde and Mayer (2007). During this transition, the radiation field changes fundamentally as the Sun's direct component is removed and the illumination becomes entirely diffuse. During totality, the irradiance at the surface will become also more sensitive to the topography (e.g., the mountains surrounding the measurement sites (Fig. 1)), surface albedo (and its spectral dependence), and the distribution of ozone in the atmosphere (the ozone profile). These aspects will be discussed in a follow-on
publication.

**Data availability**

Calibrated data and data used to draw Fig. 11 and Fig. 14 are available as a Supplement.

**Author contribution**

GB designed and executed the measurements, performed the data analysis, and wrote the manuscript.
BP contributed to the implementation of the solar limb darkening parameterizations by Pierce and Neckel and contributed to writing the manuscript.

**Competing interests**

GB is employed by Biospherical Instruments Inc, which is also the manufacturer of the GUVis-3511 radiometer described in this paper.




*Acknowledgements.* GB thanks Biospherical Instrument Inc. for supporting this activity and Anne Hoppe for her encouragement to travel to Oregon and her help in setting up the experiment. We thank Bjørn Johnsen from the Norwegian Radiation Protection Authority for performing spectral response measurements of the GUVis-3511 radiometer used in this campaign. We also thank Bernhard Mayer and Paul Ockenfuß from the Ludwig-Maximilians-Universität München, Munich, Germany, for discussing our results and for their valuable suggestions.

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
