# Peer review of "Measurements of spectral irradiance during the solar eclipse of 21 August 2017: reassessment of the effect of solar limb darkening and of changes in total ozone"

_Atmospheric Chemistry and Physics, 2018_

## Short Comment (SC1) · 24 Nov 2018

While the results in this paper are certainly intriguing, there are significant differences between the instrument employed by the authors and the TOPS instrument employed by Mims and Mims. The author's instrument uses filters having a FWHM bandpass of 10 nm, while TOPS has filters with a 5-nm bandpass FWHM. TOPS also measured column ozone at 300nm and 305 nm, which is much more sensitive to ozone variations than the wavelengths used by the authors. TOPS is also a direct sun instrument that can provide measurements in a few seconds, while the author's instrument is a full-sky

device with an exceptionally long 2-minute scan time. As we have learned from comparisons with an EPA Brewer placed at our site, the much faster scan time provided by TOPS provides higher resolution results and avoids errors caused by aerosol changes that can occur during minute-duration scans. Moreover, TOPS often detects subtle changes in the ozone column missed by Dobson and Brewer instruments, which both require considerably more time for an ozone measurement. Before our findings are ruled out by this paper, we feel that the authors should point out the very significant instrumental differences, especially the filter wavelengths, the filter bandpasses and the time required per scan. In each of these cases, TOPS offers superior performance when compared with their instrument. Thus, the findings of subtle waves in the ozone layer by TOPS cannot be so quickly discounted by this paper. I close by observing that TOPS uncovered a calibration drift in the Nimbus-7 Total Ozone Mapping Spectrometer (TOMS) (Satellite Monitoring Error, Nature 361, 1993). TOPS evolved into Microtops and then Microtops II. All these instruments provide results in close agreement with Brewers and Dobsons at the Mauna Loa Observatory. Thank you for considering the points made herein. Forrest M. Mims III fmimsiii@yahoo.com

---

## Referee Comment (RC1) · Anonymous Referee #2 · 27 Nov 2018

The paper presents in a detailed way spectral solar measurements during an eclipse and provides improved insights on RT modeling and LD effect approaches.

One important aspect is the comparison of the results of this study with previous papers Zerefos (2000, 2007), Kazadzis, Blumthaler, Grobner concluding major or minor differences in the methodological approaches and results.

Most of these studies have been performed based on the Brewer spectroradiometer. The major differences among this instrument and the instrument used in this study are:

- The methodology of deriving the total column ozone (Brewer uses pairs of different wavelength ratios and GUV one fixed ratio)

- The use of direct instead of global solar irradiance

- The spectral response of each wavelength measured (Brewer wavelength are fixed exactly at absorbing / non absorbing wavelengths with 0.5nm slits)

- And mostly: the FOV of the instruments.

In the current study the instrument (effective) FOV is defined by the dimensions of the shadowband and the corresponding dimensions of the shadow in combination with the sun dimensions.

A further examination on the effect of the above (mainly the FOV) issues have to be considered when comparing results of this and older papers using different principles of measurements. For example the definition of the diffuse (or direct) irradiance calculated using a shadow (band) that has spectral, solar zenith angle, (and in this case) also sun- dimensions dependent, apparent shadow dimensions compared with the instrument diffuser, could impact the presented results.

What do you mean by: P6, L10 The calibration was further optimized for the conditions (solar zenith angle, TOC, AOD, etc.) at the measurement site.
* * *

---

## Author Comment (AC1) · 20 Dec 2018

**Response to comments by Forrest M. Mims III.**

We thank Mr. Mims for his good comments, which have helped to improve our manuscript.

**Comment by reviewer**

While the results in this paper are certainly intriguing, there are significant differences between the instrument employed by the authors and the TOPS instrument employed by Mims and Mims. The author's instrument uses filters having a FWHM bandpass of 10 nm, while TOPS has filters with a 5-nm bandpass FWHM. TOPS also measured column ozone at 300nm and 305 nm, which is much more sensitive to ozone variations than the wavelengths used by the authors. TOPS is also a direct sun instrument that can provide measurements in a few seconds, while the author's instrument is a full-sky C1 device with an exceptionally long 2-minute scan time. As we have learned from comparisons with an EPA Brewer placed at our site, the much faster scan time provided by TOPS provides higher resolution results and avoids errors caused by aerosol changes that can occur during minute-duration scans. Moreover, TOPS often detects subtle changes in the ozone column missed by Dobson and Brewer instruments, which both require considerably more time for an ozone measurement. Before our findings are ruled out by this paper, we feel that the authors should point out the very significant instrumental differences, especially the filter wavelengths, the filter bandpasses and the time required per scan. In each of these cases, TOPS offers superior performance when compared with their instrument. Thus, the findings of subtle waves in the ozone layer by TOPS cannot be so quickly discounted by this paper. I close by observing that TOPS uncovered a calibration drift in the Nimbus-7 Total Ozone Mapping Spectrometer (TOMS) (Satellite Monitoring Error, Nature 361, 1993). TOPS evolved into Microtops and then Microtops II. All these instruments provide results in close agreement with Brewers and Dobsons at the Mauna Loa Observatory. Thank you for considering the points made herein. Forrest M. Mims III fmimsiii@yahoo.com Interactive comment on Atmos. Chem. Phys. Discuss., https://doi.org/10.5194/acp-2018-1048, 2018.

**Authors' Response**

As described in Section 5.2. of the manuscript, the total ozone column (TOC) was derived from measurements of response-weighted global irradiance measured by the

GUVis-3511. Specifically, ratios of measurement at 340 and 305 nm were compared with similar ratios in a look-up table that was calculated with a radiative transfer model as a function of SZA and TOC. The look-up table was calculated by taking the response functions of the instrument (Fig. 3 of manuscript) and observing conditions (e.g., aerosol optical depth, (AOD)) into account. The method of calculating TOC from measurements of global irradiance (instead of direct irradiance as it is typically done for Dobson,

Brewer, TOPS, and Microtops instruments) was first proposed by Stamnes et al. (1991).

We found that the accuracy of TOCs derived from global irradiance is similar to that of data from Dobson instruments or satellite (TOMS, OMI) observations if the look-up table takes local conditions into account (ozone profile, albedo, elevation, etc.) (Bernhard et al., 2005b). For example, this study uncovered systematic errors in Dobson measurements associated with approximations in the standard Dobson retrieval method, which subsequently helped to better understand the limitation of Dobson measurements.

We have further validated the method for GUV instruments (Bernhard et al., 2005a).

Based on this work we believe that our TOC measurements are not of inferior quality compared to TOPS measurements and provide further evidence below.

Ability to detect small changes in TOC

As described in the manuscript (P23, L26ff.), our data do not indicate oscillations in TOC

that may have been triggered by bow waves from the Moon's shadow. In contrast,

Zerefos et al. (2000) reported peak-to-peak variation in TOC of about 1 %. while Zerefos et al. (2007) reported a peak-to-peak amplitude of 2.0–3.5 %. Finally Mims and Mims (1993) describe a peak-to-peak amplitude of up to 5 DU (1.7 %). We show in the following that our method is capable of detecting fluctuations of this magnitude.

By analyzing the values of our ozone look-up table, we determined that a 1 % change in the ratio of the response-weighted global irradiance at 340 and 305 nm results in a TOC

change of 1.1 DU at the start of the eclipse. At the end of the eclipse (when the SZA is smaller) a 1 % change in the ratio leads to a change of 1.6 DU. As described in the manuscript, ozone calculations are based on the average of 45 seconds (not 2 minutes as stated by the reviewer) of global spectral irradiance measurements that are sampled at 1 Hz. By calculating the standard deviation of these samples and using standard error propagation, we calculated an uncertainty (confidence interval of 95 %) for the ratio of measurements at 340 and 305 nm of 0.13 % for the start and 0.06 % for the end of the eclipse. By combining these uncertainty estimates with the sensitivity of the TOC to changes in this ratio, we determined that our measurements are able to detect changes in ozone of 0.14 DU at the start and 0.10 DU at the end of the eclipse. Since the average TOC on 21 August 2017 was about 298 DU, these absolute changes translate to relative changes of 0.05 % and 0.03 %, respectively. Our method is therefore well capable to detect changes of the magnitude of 1 to 3.5 % reported by Zerefos et al. (2000, 2007) and Mims and Mims (1993).

We agree with the reviewer that measurements at 300 nm are more sensitive to changes in ozone than measurements at 305 nm. However, measurements at 300 nm are also noisier than measurements at 305 nm because the irradiance at 300 nm is smaller than that at 305 nm by factors of 10 (end of eclipse) to 30 (start of eclipse). Moreover, small errors in the characterization (e.g., center wavelength) of the filters functions have a larger effect at shorter wavelength. Since we do not have access to a TOPS or Microtops, we cannot determine whether these instruments are really superior to the GUVis-3511 in determining the TOC as the reviewer asserts. (Of note, the shortest wavelength of a Dobson is 305.5 nm and if measurements at 300 nm would be of great advantage, these instruments would likely use a shorter wavelength.) In any case, our analysis above illustrates that our instrument is sensitive enough for detecting bow-waved induced TOC variations of the proposed magnitude.

Effect of aerosols on ozone retrieval

The look-up table for TOC retrievals was calculated with the same model parameters that were used to convert response weighted irradiance measurements to spectral irradiance as described in the Supplement to this publication. Specifically, aerosol extinction was parameterized with Ångström's turbidity formula by setting the Ångström coefficients $\alpha$

and β to 2.10 and 0.0394, respectively. These values were derived from the instrument's measurement of direct solar irradiance at 19:00:30 after the end of the eclipse. At the start (16:02:58; see Fig. 8b of manuscript), the Ångström coefficients were 1.96 and 0.0570, respectively. To quantify the effect of changing aerosol conditions on TOC retrievals, we modeled the global spectral irradiance for 16:02:58 using either the Ångström coefficients used for the TOC look-up table or the coefficients applicable to this time.

The difference in AOD changed the ratio of global spectral irradiance at 340 and 305 nm by only 0.12 %. Using the same sensitivity factor discussed above, we determined that the resulting bias in TOC is 0.13 DU or 0.04 %.

Limb-darkening corrected AODs (Fig. 8a), suggest that the largest AOD during the eclipse occurred at 16:52 when the Ångström coefficients were 1.95 and 0.0788, respectively. (This estimate is somewhat uncertain due to the uncertainty of the limb- darkening correction.) We calculated the aerosol effect in a similar way as before and conclude that the elevated AOD at 16:52 increases the retrieved TOC values by 1.5 DU

or 0.5 %. Again, this value is considerably smaller than the fluctuations reported by

Zerefos et al. (2000, 2007). Yet, aerosols can explain 1.5 DU of the 5.0 DU increase in limb-darkening corrected TOC measurements that can be seen in Fig 8a of the manuscript between 16:00 and 17:00.

We also like to point out that TOC retrievals from global irradiance is less sensitive to changes in AOD than retrievals using the direct beam (such as those utilized by the TOPS

instruments) because photos that are removed from the direct beam by aerosols contribute to the global irradiance. So direct measurements are not necessarily better.

Sampling frequency

TOC measurements discussed in the manuscript are available at a rate of one value every

2 minutes. It is therefore not possible to determine whether the moon's shadows resulted in fluctuations in TOC on a shorter time scale. However, oscillations reported by Zerefos et al. (2000) had a period of 20 minutes and those reported by Zerefos et al. (2007) had periods ranging between 30 and 40 minutes. Zhang et al. (2017) reported ionospheric bow waves, which manifested themselves as electron content disturbances, with a period of about 25 minutes during the solar eclipse of 21 August 2017. If these bow waves had also affected the ozone layer, these oscillations should have been detectable with our sampling frequency. Finally, Mims and Mims (1993) report that TOC measurements taken during the total solar eclipse of 11 July 1991 show three fluctuations, which began

700 seconds after the third contact with durations of 378, 270 and 432 seconds, respectively. It is curious that these oscillations had a much shorter duration than those reported by Zerefos et al., (2000, 2007) and Zhang et al. (2017). In any case, oscillations of this duration should still shown up in our 2-minute data considering the low noise in our data discussed above and illustrated in Fig. 9, in particular between the 3$^{rd}$ and 4$^{th}$

contact.

Concluding remarks

We did not "rule out" the findings of the paper by Mims and Mims (1993) as stated by the reviewer. Instead, we simply stated that our data do not support the observation by

Zerefos (2000; 2007) and Mims and Mims (1993), and concluded that the question of whether or not bow waves from the Moon's shadow can lead to variations in TOC is still up for debate. We suggested that this debate could be settled by performing limb- darkening-corrected measurements of TOC with different instrument types during one of the upcoming solar eclipses. If such measurements were to show fluctuations in TOC

with the same magnitude and timing, the effect of bow waves on TOC could be convincingly demonstrated. We still believe that this is a reasonable path forward and might stimulate future research.

**Proposed changes to manuscript**

The following text will be added to Sect. 5.2 of the manuscript:

[revised manuscript text omitted]

---

## Author Comment (AC2) · 20 Dec 2018

**Response to comments by Anonymous Referee #2**

We thank the referee for his or her comments, which we have addressed as follows:

**Comment by Referee**

A further examination on the effect of the above (mainly the FOV) issues have to be considered when comparing results of this and older papers using different principles of measurements. For example the definition of the diffuse (or direct) irradiance calculated using a shadow (band) that has spectral, solar zenith angle, (and in this case) also sun-dimensions dependent, apparent shadow dimensions compared with the instrument diffuser, could impact the presented results.

**Authors' Response**

First we like to point out that total ozone column (TOC) was calculated from measurements of global spectral irradiance, not direct solar irradiance as indicated by the referee's introductory statements. The FOV of the shadowband is therefore irrelevant for TOC retrievals. Please see our response to the comments by Forrest M. Mims III regarding TOC calculations.

Shadowband measurements were only used to determined aerosol optical depth and the direct-to-diffuse ratio. The shadowband obstructs a wedge of the sky with a width of 15° when the band is vertical and 13° when it is horizontal. These relative large angles may suggest that the instrument is not able to adequately remove the contribution of circumsolar radiation when calculating the direct irradiance, which would also lead to systematic errors in aerosol optical depth (AOD) retrievals. However, this notion is not correct due to the unique way of shadowband operation and data analysis. Other shadowband radiometer typically use two measurements on either side of the Sun to correct for the signal lost from the portion of the sky that is obscured by the shadowband. For example, side-band measurements of Multifilter Rotating Shadowband Radiometer (MFRSR) are typically performed at 9° from the center of the Sun (Krotkov et al., 2005). Hence, the increase of diffuse radiation towards the Sun is often not corrected adequately.

In contrast, direct measurements and AOD retrievals of GUVis-3511 measurements are based on measurements at high sampling rate (15 Hz) whereby the band moves slowly and continuously over the instrument. The processing algorithm determines the irradiance from direct Sun plus the unshaded portion of the sky by analyzing several seconds of measurements when the shade from the band is close to the diffuser but not in contact with it. The algorithm is described in detail by Hooker et al. (2012) and Witthuhn et al. (2017). The algorithm compensates for the effect of the relatively large FOV, resulting in AOD data that have similar accuracy than that of traditional shadowband radiometers.

**Proposed change to manuscript**

The following text will be added to Sect. 3 of the manuscript:

"The band obstructs a wedge of the sky with a width of 15° when the band is vertical and 13° when it is horizontal. […] The algorithm compensates for the effect of the relatively large width of the shadowband, resulting in AOD data that have similar accuracy than those of traditional shadowband radiometers, which use measurements on either side of the Sun to correct for the portion of the sky that is obscured by the band. For example, side-band measurements of Multifilter Rotating Shadowband Radiometer (MFRSR) are typically performed at 9° from the center of the Sun (Krotkov et al., 2005) and therefore may not adequately measure circumsolar radiation. Our algorithm alleviates this problem."

**Comment by Comment by Referee**

What do you mean by: P6, L10 The calibration was further optimized for the conditions (solar zenith angle, TOC, AOD, etc.) at the measurement site.

**Authors' Response**

As described in the Supplement of the manuscript, the instrument measures "response-weighted irradiance". The conversion to spectral irradiance requires knowledge of the solar zenith angle, TOC, AOD, etc., and uncertainties in these parameters results in

uncertainty in the conversion factor. These issues are discussed at length in the Supplement.

**Proposed change to manuscript**

The following will be added after the sentence in question:

"Details of this optimization are provided in the Supplement."

**References**

Hooker, S. B., Bernhard, G., Morrow, J. H., Booth, C. R., Comer, T., Lind, R. N., and Quang, V.: Optical Sensors for Planetary Radiant Energy (OSPREy): calibration and Validation of Current and Next-Generation NASA Missions., NASA Goddard Space Flight Center, NASA/TM–2011–215872, 2012.

Krotkov, Nickolay A., Pawan K. Bhartia, Jay R. Herman, James R. Slusser, Gordon J. Labow, Gwendolyn R. Scott, George T. Janson, Tom Eck, and Brent N. Holben. "Aerosol ultraviolet absorption experiment (2002 to 2004), part 1: ultraviolet multifilter rotating shadowband radiometer calibration and intercomparison with CIMEL sunphotometers." *Optical Engineering* 44, no. 4 (2005): 041004

Witthuhn, J., Deneke, H., Macke, A., and Bernhard, G.: Algorithms and uncertainties for the determination of multispectral irradiance components and aerosol optical depth from a shipborne rotating shadowband radiometer, Atmos. Chem. Phys., 10(2), 709–730, https://doi.org/10.5194/amt-10-709-2017, 2017.

---

## Short Comment (SC2) · 21 Dec 2018

TWO KEY POINTS: The authors have made several important revisions to their paper, but they simply must remove their erroneous assertions to the effect that: (1) the signal at 300 nm is noisy, which suggests poor measurements by TOPS (which used 300 nm and 305 nm) and (2) full-sky measurements of the ozone layer are "similar" to the direct sun measurements employed by Dobsons (and TOPS, Microtops, Brewers and Pandoras). These inappropriate claims and their 2-minute measurement time support their general assertions that raises doubts about the papers by me and others.

My recommendations are fully supported by these facts:

1. OPTIMUM WAVELENGTH SELECTION FOR MEASURING TOTAL COLUMN OZONE DURING A SOLAR ECLIPSE:

The authors claim the 300-nm minimum employed by TOPS provides a noisy signal. They erroneously observe that: "Of note, the shortest wavelength of a Dobson is 305.5 nm and if measurements at 300 nm would be of great advantage, these instruments would likely use a shorter wavelength."

The 305.5-nm Dobson minimum wavelength was selected due to the use of the instrument across a wide band of latitudes. However, there is ample signal at 300 nm at my site (29.9 N), as demonstrated by the instrument's detection of an error in NASA's Nimbus-7 TOMS ozone instrument (Mims, Nature, 1993). The 300 nm signal was especially strong at 22 N during the 1991 solar eclipse reported in my paper in GRL. The authors might be right about the 300-nm signal at their northerly location. But they cannot compare what might have been a noisy 300 nm signal at their northerly 44.36 N site to the much higher amplitude 300-nm signal at the 22 N site for the eclipse I measured.

Furthermore, DeLuisi and others have demonstrated that wavelengths below 305 nm provide more accurate ozone measurements than higher wavelengths. (This explains why TOPS found the satellite error.) Of special concern is this from the author's abstract: "The total ozone column (TOC) was derived from measurements of global irradiance at 306 and 340nm." While the 306 nm minimum is appropriate for 44.36 N, the 340 nm upper wavelength is far too high, for it allows for significant aerosol errors. The closely-spaced 300 nm and 305 nm wavelengths of TOPS nearly eliminated the aerosol error, which explains this instrument's excellent accuracy when compared with hundreds of satellite measurements and an EPA Brewer for 60 days at my site. Consider this abstract by Saunders et al in "High‐precision atmospheric ozone measurements using wavelengths between 290 and 305 nm" (JGR Atmospheres 1984

https://doi.org/10.1029/JD089iD04p05215 ) "Abstract "It is shown theoretically that many errors are significantly less when determining atmospheric ozone thicknesses from measurements of solar terrestrial spectral irradiance in the wavelength region between 290 and 305 nm as compared to the 305ÃŘ to 340ÃŘnm region employed by the Dobson spectrophotometer. In order to test this conclusion experimentally, an elaborate set of stateÃŘofÃŘtheÃŘart measurements have been made in the shorter wavelength region in Gainesville, Florida, between June 13 and June 18, 1980. Details of these measurements, including an extensive error analysis, are presented and indicate that such shortÃŘwavelength measurements, particularly between 295 and 305 nm, can be used to detect longÃŘterm changes of atmospheric ozone with an uncertainty not exceeding 1%. Observing conditions restricted the Gainesville measurements to zenith angles of less than 35°. Further investigations are required to determine the shortest wavelength that can be used at significantly greater zenith angles."

2. FULL-SKY VS. DIRECT SUN TOTAL OZONE MEASUREMENTS:

The authors state in their paper: "The method of calculating TOC from measurements of global irradiance (instead of direct irradiance as it is typically done for Dobson, Brewer, TOPS, and Microtops instruments) was first proposed by Stamnes et al. (1991). We found that the accuracy of TOCs derived from global irradiance is similar to that of data from Dobson instruments or satellite (TOMS, OMI) observations if the look-up table takes local conditions into account (ozone profile, albedo, elevation, etc.) (Bernhard et al., 2005b)."

The authors suggest that global irradiance provides TOC data ". . . similar to that of data from Dobson instruments or satellite. . . ." But the authors provide no data or citations to support this assertion, while leaving open the counter suggestion that Dobsons (and TOPS, Brewers and Pandoras) could be replaced by much simpler instruments that measure global irradiance and require no tracking and pointing. Of course, this is not the case. The authors have gone much, much too far in suggesting their global,

full-sky data is "similar" to direct sun measurements made by the recognized standard for nearly a century and all other ozone instruments.

I have always been intrigued by the prospect of accurate ozone retrievals by pairs of global UV measurements at closely-space wavelengths. I urge the authors to prepare a detailed paper about their claim that global TOC measurements are "similar" to those by traditional direct sun instruments. An ideal comparison would be with Brewers, which measure both direct sun and global TOC. Meanwhile, it is inappropriate to make an unsupported claim about "similar" results.
* * *

---

## Referee Comment (RC2) · Anonymous Referee #1 · 27 Jan 2019

The paper provides a in-depth analysis of spectral solar measurements during an eclipse focusing on a number of measuring and modelling issues.

Paragraph 7.4: I am referring firstly to this paragraph because the paper is becoming quite confusing at this point. If the authors conclude that the LD parameterization by Waldmeier is too simple, particularly in UV range, why this is being discussed previously in the estimation of total $O_3$? It is recommended to move this paragraph to 7.1 and then discuss the results based on those findings on LD parameterizations. In this concept, paragraph 7.3 should follow the new 7.1. In figure 10, the effect of the eclipse is masked by the dominant effect of changing solar zenith angle. The authors are asked to subtract the solar zenith angle effect by using the AOD and $O_3$ measurements just at the end (or start) of the eclipse (as inputs in model calculations) and, then, discuss the effect of the eclipse and the changes on aerosols and ozone (these paragraph should follow now).

Page 2, lines 15-20: Kazantzidis et al (2007) were measuring with NILU-UVs and calculated total ozone, too. They reported a slight increase when the visible part of the sun was more than 20% and decreased significantly as the eclipse progressed.

Figure 3 and relevant text: please provide a figure only with the UV wavelengths, no logarithmic axes. From the literature it seems that the bandwidth of 305nm channel is quite wide: it is more than 10nm even at full width at half maximum. In this case, please discuss the possible implications when measuring with this high bandwidth.

Page 5, lines 15-24: please provide/add some sentences about the performance of this method on estimating the direct and the shadowband corrected spectral irradiances.

Figure 4 and relevant text: What is the expected cosine error in UV wavelengths? Are there any measurements? Are the measurements in UV channels (used for this study) cosine corrected? How significant this effect is expected to be during the solar eclipse (when the direct component of solar irradiance is minimized)?

Figure 8: Aerosol Optical Depth (AOD) around 305nm is not depicted in figures 8a and 8b. Why? Do you think that the LD correction (8b) by Pierce is valid also for the lower UV wavelengths? If yes, how this decrease of AOD in these wavelengths can be explained? There is an assumption that this is not a measurement artifact but an unknown absorber. Which type of absorber could change the expected AOD values only at 314.2, 319.4nm (and 442.4nm!)? Moreover, what would be the AOD estimations if you apply the Neckel LD parameterization?

Paragraph 7.2: It is quite surprising that although AOD can be estimated by direct GUV measurements, this is not happening for the total $O_3$ amount as well, despite that direct measurements (divided by global ones) are presented in figure 10. Moreover, the 305/340 wavelength ratio methodology to derive total $O_3$ is based on model calculations as a function of $O_3$ and solar zenith angles but under cloud-free skies e.g. specifically defined direct and diffuse components of solar irradiance. This is not the solar eclipse case. And this is accounted by the authors. However, how valid are the LD parameterizations on CHANNEL irradiances when it is known that the direct/diffuse

ratio has significant spectral sensitivity? And how repeatable will be their results if the they use: a) direct irradiances, b) different channel ratios, e.g. 305/320, 312/340? The authors here should acknowledge a couple of very crucial facts: 1) the 305/340 wavelength ratio of channel irradiances is a well-known method that can be used for total $O_3$ estimations but it is accompanied by significant uncertainties (aerosol optical depth and scattering properties, cloudiness, ozone profile, direct/diffuse irradiance etc), 2) the comparison with ozone values or previous studies derived from instruments measuring the direct irradiance should be done under the acknowledgement that these measurements are correct not only because they are the standard ones but also because they are correct in terms of physics and the best in terms of overall uncertainty.

Figure 9 and relevant text: Kazantzidis et al (2007), when using the 305/320 wavelength ratio (in order to reduce the effect of spectral effect of the eclipse on direct and diffuse irradiances) reported very similar results to those derived from yours when using the 305/340 wavelength ratio AND applying the Neckel or Pierce parameterization. Surprisingly, this paper is referenced only for the comparison of measured and model irradiances. However, the results of this paper for ozone, irradiances and irradiance ratios vs eclipse percentage are not referenced, although it is based on results from 8 narrowband multi-channel NILU-UV6 radiometers.

Paragraph 8.1: the authors seem to have a point here. In order to better understand the similarities/differences with previous studies, a more detailed information is needed apart from direct/diffuse ratios: the measured wavelengths and the eclipse percentages should be provided. Moreover, the theoretical calculations from Emde and Mayer are quite capable to estimate the global irradiance (when normalized 5 minutes before totality) at 380 nm but maybe irradiance is significantly underestimated at 312 nm (Kazantzidis et al., 2007, figure 7 and relevant text). This affects directly the diffuse component. Of course, also this result is sensitive to factors like surface albedo, ozone profile and the dynamic range of the measuring system. All these factors should be mentioned.

Page 22-23: This comment refers to the ozone issue, described in detail by the authors. From my point of view, some (or much?) of these differences could be attributed on the measuring methods and the selected pairs of wavelengths. As mentioned before, a decrease of (uncorrected for LD effect) ozone retrievals has been reported by Kazantzidis et al (2007) for the same eclipse when using 8 NILU-UVs. As it was stated earlier in this review, the authors could strongly defend their findings if they will come up with the same results when using other wavelength pairs and the direct GUV irradiances. Unfortunately, this measuring campaign is not accompanied by more instruments.

---

## Author Comment (AC3) · 12 Mar 2019

**Response to comments by Anonymous Referee #1**

We thank the referee for his or her comments, which we have addressed as follows:

**Comment by Referee**

Paragraph 7.4: I am referring firstly to this paragraph because the paper is becoming quite confusing at this point. If the authors conclude that the LD parameterization by Waldmeier is too simple, particularly in UV range, why this is being discussed previously in the estimation of total O3? It is recommended to move this paragraph to 7.1 and then discuss the results based on those findings on LD parameterizations. In this concept, paragraph 7.3 should follow the new 7.1. In figure 10, the effect of the eclipse is masked by the dominant effect of changing solar zenith angle. The authors are asked to subtract the solar zenith angle effect by using the AOD and O3 measurements just at the end (or start) of the eclipse (as inputs in model calculations) and, then, discuss the effect of the eclipse and the changes on aerosols and ozone (these paragraph should follow now).

**Authors' Response**

We have changed the sequence of Sect. 7 as suggest by the referee. The sequence is now:

7.1 Effect of solar limb darkening

7.2 Direct-to-global ratio

7.3 Aerosol optical depth

7.4 Total ozone column

7.5 Measurements near and during totality

We have also changed the sequence of Sect. 8 (Discussion) to be consistent with Sect. 7. The new sequence of Sect. 8 is now:

8.1 Magnitude of solar limb darkening

8.2 Variations of direct-to-global ratio

8.3 Total ozone column variations during an eclipse

8.4 Validation of GUVis-3511 measurements

Following the suggestion by the referee, we have compared the measured direct-to-global ratio with the corresponding modeled ratio and have added a new panel to Figure 10 (now Figure 11) , which shows the ratio of measurement and model. The text was modified to describe the new panel, see below.

**Change to manuscript**

The following was added to Sect. "Direct-to-global ratio" (now Section 7.2)

"The measured direct-to-global ratio was compared with the modeled direct-to-global ratio using the same method as that employed in Sect. 7.1. In brief, the measured direct-to-global ratio was divided by the modeled direct-to-global ratio. To correct for the difference between measurement and model, linear functions were again constructed to match the ratios of measurement and model at the times of the 1$^{st}$ and 4$^{th}$ contact  Finally, the ratios were normalized by dividing with these linear functions. The resulting bias-corrected ratios of the measured and modeled direct-to-global ratio are shown in Fig. 11b and are denoted $R_{DG}(\lambda_i)$.

Before 15:00, the measured direct-to-global ratio is smaller than 0.03 at wavelengths in the UV-B

and values of $R_{DG}(\lambda_i)$ are subject to large uncertainties at these wavelengths. Before 15:20,

$R_{DG}(\lambda_i)$ varies between 0.55 and 0.87 for wavelengths in the visible range, indicating that the measured direct-to-global ratio is 13 to 45 % below the modeled one.  These low values can be attributed to increased aerosol loading prior to the start of the eclipse. After the start of the eclipse (1$^{st}$ contact), values of  $R_{DG}(\lambda_i)$ start to drop to reach a local minimum at 16:52, approximately

30 minutes before the eclipse maximum. The decline has a clear wavelength dependence with wavelengths in the UV-B decreasing the most (up to 20 %), followed by wavelengths in the visible (up to 15%) and IR (up to 3.5%). After this local minimum, $R_{DG}(\lambda_i)$ slowly increases during the remainder of the eclipse.

The change of $R_{DG}(\lambda_i)$ over the period of the eclipse could either be caused by processes initiated by the occlusion of the Sun or by variability from aerosols. Aerosols effects are more likely because any eclipse effects should conspicuously peak at the time of totality, not 30

minutes earlier. The wavelength dependence of $R_{DG}(\lambda_i)$ is also characteristic for aerosol effects (Sect. 7.3). In addition, the  minimum values of $R_{DG}(\lambda_i)$ during the eclipse are well within the range of values observed prior to the eclipse, which could be unambiguously attributed to aerosols."

[Figure]

Fig. 1. Panel (a): ratio of direct-to-global spectral irradiance. Measurements are shown as heavy lines. Thin
lines connect measurements at the start of end of the eclipse and are drawn to guide the eye. Panel (b):
Ratio of the measured and modeled direct-to-global ratios, corrected for the bias between measurement and
model. Numbers "(1)" and "(2)" in the legend indicate the channel number for channels equipped with
identical wavelengths.

In Section 8.1., the following paragraph:

"The direct-to-global ratios shown in Fig. 10 increase with time as expected from the
decrease in SZA. Deviations from a straight line between the 1[st] and 4[th] contact are less
than 0.04. It is difficult to determine whether this deviation is caused by processes
initiated by the occlusion of the Sun, by variability from aerosols, or artifacts of the
algorithm to calculate the direct irradiance from shadowband data. We therefore consider
the deviation of 0.04 as an upper limit for the eclipse effect."

was removed and replaced with:

"Relative to model calculations for the unoccluded Sun, direct-to-global ratios during the
eclipse decreased by up to 20 % in the UV-B, 15 % in the visible and 3.5 % in the IR
(Fig. 1b). The largest decrease was observed about 30 minutes before totality. The timing
of this decrease plus its spectral dependence suggest that changes in aerosol amounts are
the main driver for the observed drop in the ratio. However, aerosol effects are difficult to
decouple from processes initiated by the occlusion of the Sun or artifacts of the algorithm
used to calculate the direct irradiance from shadowband data."
Note that this paragraph is now part of Section 8.2.

**Comment by Referee**

Page 2, lines 15-20: Kazantzidis et al (2007) were measuring with NILU-UVs and calculated total
ozone, too. They reported a slight increase when the visible part of the sun was more than 20%
and decreased significantly as the eclipse progressed.

**Authors' Response and change to manuscript**

The reference to Kazantzidis et al. (2007) has been added to the introduction in support of the
following statements:
"study wavelength-dependent changes in spectral irradiance"
"short-term and longer-lasting fluctuations in the total ozone column (TOC)"

Furthermore, we have added the following to Sect. 8.3. of the Discussion:
"Finally, Kazantzidis et al (2007) discuss TOC measurements performed with NILU-UV
filter radiometers at several locations in Greece during the total solar eclipse of 29 March
2006. They did not observe any periodic fluctuations in TOC and only report a small
increase in TOC of about 5 DU as the visible fraction of the Sun decreases from ~60% to
~20%. This small change could be caused by incomplete correction of the LD effect."

**Comment by Referee**

Figure 3 and relevant text: please provide a figure only with the UV wavelengths, no logarithmic
axes. From the literature it seems that the bandwidth of 305nm channel is quite wide: it is more
than 10nm even at full width at half maximum. In this case, please discuss the possible
implications when measuring with this high bandwidth.

**Authors' Response and change to manuscript**

Figure 3 was plotted separately for the UV, visible and IR wavelength range on a non-logarithmic x-axes. The new figure plus raw data in text format will be made available as Supplements.

The following was added to Section 3:

"The spectral bandwidth of all channels is approximately 10 nm full width at half maximum (FWHM) with the exception of the two channels at 305 nm, which have a bandwidth of 18.5 nm."

The following was added to the Caption of Fig. 3:

"A version of the figure, plotted separately for UV, visible, and IR wavelengths, is part of the Supplement."

The following was added to Sect. 5.1:

"Aerosol optical depth was not calculated for the two 305 nm channels because of the large bandwidth of these channels and the strong interference with ozone absorption at this wavelength. Both factors lead to large uncertainties."

As described in Section 5.2., TOCs are calculated using look-up tables that are based on response-weighted global irradiance, i.e., the spectral irradiance weighted with the spectral response functions shown in Fig. 3. TOC calculations therefore take the large bandwidth of the

305 nm channels into account.

The impact of the relatively large bandwidth of the 305 nm channels on spectral irradiance calibrations is implicitly addressed by the vicarious calibration method described in Sect. S1.2 of the Supplement. Uncertainties to derive spectral irradiance at 1 nm resolution from response- weighted irradiance are discussed in Section S1.3 of the Supplement.

| 1 | **Comment by Referee** |
| --- | --- |
| 2 | Page 5, lines 15-24: please provide/add some sentences about the performance of this method on |
| 3 | estimating the direct and the shadowband corrected spectral irradiances. |
| 4 | |
| 5 | **Authors' response and change to manuscript** |
| 6 | The following was added to the paragraph in question: |
| 7 | "The uncertainty of our method was estimated by Witthuhn et al. (2017). AOD can be |
| 8 | retrieved with an uncertainty of 0.02 for all channels within a 95 % confidence interval." |
| 9 | |
| 10 | **Comment by Referee** |
| 11 | Figure 4 and relevant text: What is the expected cosine error in UV wavelengths? Are there any |
| 12 | measurements? Are the measurements in UV channels (used for this study) cosine corrected? |
| 13 | How significant this effect is expected to be during the solar eclipse (when the direct component |
| 14 | of solar irradiance is minimized)? |
| 15 | |
| 16 | **Authors' Response** |
| 17 | The cosine error in the UV has also been measured but results were omitted from Fig. 4 because |
| 18 | these measurements are very noisy due to the low output of the FEL used for the characterization |
| 19 | in the UV. In general, the cosine error of wavelengths between 305 and 380 nm is similar to that |
| 20 | at 395 nm because scattering properties of PTFE (Teflon) deteriorate only towards longer |
| 21 | wavelengths but not towards shorter wavelengths. We have now generated a composite figure |
| 22 | showing cosine errors in the UV in the upper panel and those for the visible and IR ranges in the |
| 23 | lower panel: |

[Figure]

Fig. 2: Cosine error of the GUVis-3511 radiometer in one azimuthal plane at wavelengths between 305 and

395 nm (Panel a) and 395 and 1020 nm (Panel b). Measurements performed at the orthogonal plane are similar. Measurements at wavelengths below 395 nm are affected by noise due the low UV output of the incandescent lamp used for the characterization.

The new figure will be included in the Supplement. Measurements of UV channels used in our study are cosine error corrected. Corrections of channels between 305 and 380 nm are based on the cosine error of the 395 nm channel to avoid spurious corrections caused by the noisy cosine response data. The cosine error correction takes the measured direct-to-global ratio into account.

The accuracy of the cosine error correction method is therefore not affected by the solar eclipse.

**Change to manuscript**

The following was added to Sect. 3:

"Measurements of the cosine error at wavelengths below 395 nm are affected by noise due the low UV output of the incandescent lamp used for the characterization.

Corrections for the cosine error of UV channels described further below are based on the measured cosine error at 395 nm."

The following was added to the caption of Fig. 4:

"(A figure with cosine errors in the UV is provided in the Supplement.)"

The following was added to Sect. 4:

"Both direct and global irradiance measurements were corrected for the cosine error of the instrument's collector as described by Morrow et al. (2010), based on the measured cosine error (Fig. 4) and the ratio of direct and global irradiance extracted from the shadowband measurements."

**Comment by Referee**

Figure 8: Aerosol Optical Depth (AOD) around 305nm is not depicted in figures 8a and 8b. Why?

Do you think that the LD correction (8b) by Pierce is valid also for the lower UV wavelengths? If yes, how this decrease of AOD in these wavelengths can be explained? There is an assumption that this is not a measurement artifact but an unknown absorber. Which type of absorber could change the expected AOD values only at 314.2, 319.4nm (and 442.4nm!)? Moreover, what would be the AOD estimations if you apply the Neckel LD parameterization?

**Authors' Response and change to manuscript:**

The uncertainty in calculating AOD at 305 nm is far too high to be useful. As already mentioned earlier, we have added the following to Sect. 5.1 to make this clear:

"Aerosol optical depth was not calculated for the two 305 nm channels because of the large bandwidth of these channels and the strong interference with ozone absorption at this wavelength. Both factors lead to large uncertainties."

Yes, we believe that the LD correction by Pierce is valid also for the lower UV wavelengths because the difference in global irradiance between measurement and theory is also reasonable at

305 nm (Fig. 11d of original manuscript). Furthermore, the LD corrections by Pierce and Neckel (which were derived from independent datasets) agree to within 1.8% at 305 nm.

We do not know with certainty why the AODs at 314.3, 319.4, and 442.4 nm are below the

AODs expected from the Ångström parameterization, which was derived using channels between

340 and 1020 nm, excluding the channel at 940 nm, as stated in the manuscript. These low values could be caused by absorbing aerosols released by the near-by fires or measurement errors, but this is speculation. If the low values were due to measurement errors, one would expect that the discrepancies for the three wavelengths observed at 16:02:58 and 19:00:30 are of similar magnitude because measurements at both times were processed in the same way. The fact that these discrepancies are larger at 19:00:30 suggests that they are caused by real changes in aerosol properties. While these drops could be caused by an unknown absorber, we removed this suggestion from the text because we don't have evidence that such an absorbing aerosol or gas was indeed present.

Considering the uncertainty with respect to the cause of the low AODs at the three wavelengths, we changed the last sentence of the paragraph to:

"As measurements at the two time were processed in the same way it seems unlikely that the discrepancies are caused by a measurement artifact."

With respect to the question "Moreover, what would be the AOD estimations if you apply the

Neckel LD parameterization?" we note that AODs shown in the figure were calculated for times before the start (16:02:58) and after the end (19:00:30) of the eclipse. Hence, no LD

parameterization was applied or is necessary.

* * *
**Comment by Referee**

Paragraph 7.2: It is quite surprising that although AOD can be estimated by direct GUV

measurements, this is not happening for the total O3 amount as well, despite that direct measurements (divided by global ones) are presented in figure 10. Moreover, the 305/340

wavelength ratio methodology to derive total O3 is based on model calculations as a function of

O3 and solar zenith angles but under cloud-free skies e.g. specifically defined direct and diffuse components of solar irradiance. This is not the solar eclipse case. And this is accounted by the authors. However, how valid are the LD parameterizations on CHANNEL irradiances when it is known that the direct/diffuse ratio has significant spectral sensitivity? And how repeatable will be their results if the they use: a) direct irradiances, b) different channel ratios, e.g. 305/320,

312/340? The authors here should acknowledge a couple of very crucial facts: 1) the 305/340

wavelength ratio of channel irradiances is a well-known method that can be used for total O3

estimations but it is accompanied by significant uncertainties (aerosol optical depth and scattering properties, cloudiness, ozone profile, direct/diffuse irradiance etc), 2) the comparison with ozone values or previous studies derived from instruments measuring the direct irradiance should be done under the acknowledgement that these measurements are correct not only because they are the standard ones but also because they are correct in terms of physics and the best in terms of overall uncertainty.

**Authors' Response**

First, we like to emphasize that this paper is not about the most accurate or appropriate method to calculate TOC. TOC measurements were mainly included in the manuscript (i) to illustrate that

TOC measurements during an eclipse are affected by the solar limb darkening (LD) effect and that errors arising from this effect can be greatly reduced with appropriate LD corrections, and (ii)

to determine whether gravity waves generated by the Moon's shadow can cause oscillations in

TOC as suggest by several authors in the past. In our opinion, both questions were sufficiently addressed with the results presented in the original manuscript. Please see also our response to the comments by Forrest M. Mims.

Motivated by the referee's comments, we have calculated total ozone now also from global irradiance of the wavelength pairs 305 / 313 and 313 / 340. Results are presented in Fig. 3. Fig. 3a shows results without LD correction. TOCs for the 305 / 340 pair (blue) are identical with the

"uncorrected" data shown in Fig. 9 of the original manuscript. Ozone values for the 313 / 340 are

*lower* than those of the 305 / 340 pair and exhibit a *negative* slope as a function of time. In contrast, ozone values for the 305 / 313 pair are *larger* than those of the 305 / 340 pair and exhibit a *positive* slope. These biases are likely caused by a small systematic error in measurements of the 313 nm channel. When measurements of this channels are scaled with a factor of 0.95 and ozone retrievals repeated, TOCs derived from the three wavelengths pairs are consistent to within

±2.5 DU outside the period of the eclipse (Fig. 3b). A systematic error of 5% is well within the expanded uncertainty of 7.3% of the 313 nm channel (see table S1 of Supplement.). Using this adjustment, measurements of all pairs agree with OMI data to within 5 DU (1.7%) (red line in

Fig. 3).

[Figure]

Fig. 3. Total ozone column derived from global irradiance measurements of the GUVis-3511 radiometer using different wavelength pairs. Panel (a): comparison of ozone retrievals with the 305 / 340 pair (blue), the 313 / 340 pair (orange), and the 305 / 313 pair (green). Panel (b): same as Panel (a) but measurements of the 313 nm channel were scaled with a factor of 0.95 before calculating ozone. Panel (c): TOC retrievals for the 313 / 340 pair using either no correction (blue), or the LD-corrections by Waldmeier (grey), Pierce (cyan), and Neckel (green). Panel (d): same as Panel (c) but for the 305 / 313 pair. Measurements of the 313 nm channel were scaled by 0.95 for all retrievals shown in Panels (c) and (d). Long-dashed lines indicate the start and end times of the eclipse and the short-dashed line indicates the time at totality. Note that y-scales of the various panels differ.

TOC data shown in Fig. 3c and 3d were corrected for the LD-effect similar to the TOC data derived from the 305 / 340 pair shown in Fig. 9 of the original manuscript. As in that figure, TOCs calculated for the 313 / 340 and 305 / 313 pairs (Fig. 1c and 1d, respectively) spike at the time of totality when no LD correction is applied. The LD correction by Waldmeier has only a marginal effect on this result. In contrast, LD corrections using the parameterizations by Pierce and Neckel greatly decrease this spurious spike.

Calculations of TOCs from global irradiance using the pairs 305 / 313 and 313 / 340 therefore confirm the conclusions reached from results of the 305 / 340 nm pair that were presented in the original manuscript:

- There are no periodic oscillations in the TOC, either before or after totality, that could be attributed to the effects of bow waves from the Moon's shadow.

- The LD corrections by Pierce and Neckel greatly reduce the spurious spike in TOCs during totality resulting from the LD effect. In contrast, the correction that is based on the parameterization by Waldmeier is too small.

- The spurious TOC spike is larger for the 313 / 340 pair than for the 305 / 340 pair even though the relative difference in the LD effect is smaller for the two wavelengths used in the former pair. This observation is due to the fact that small errors in measurements at 313 nm have a larger effect on TOC than at 305 nm because the absorption coefficient of ozone is much smaller at 313 than 305 nm. The effect of ozone absorption on the 305 / 340 ratio therefore outweighs the LD effect.

- There is no evidence that aerosol effects have an important effect on our TOC measurements. While we agree with the referee's assessment that the method of calculating TOCs from ratios of global irradiance at 305 and 340 nm "is accompanied by significant uncertainties (aerosol optical depth and scattering properties, cloudiness, ozone profile, direct/diffuse irradiance etc)," we believe that these uncertainties are of little relevance for the conclusions reached in our paper. Specifically:
  - AODs were characterized and used for the lookup tables for ozone retrievals. The good consistency of TOC retrievals for different wavelength pairs (after adjusting for the bias of measurements at 313 nm) confirms that aerosol effects were addressed adequately.
  - There were no clouds during the period of the measurements that could have skewed our TOC retrievals.
  - The ozone profile (through the Umkehr effect) becomes only important for SZAs > 75° (i.e., outside the range of SZAs occurring during the eclipse).
  - Changes in the direct-to-global irradiance agree to within 2.5% for the wavelength range 305 - 340 nm, except for a short period centered about totality (Fig. 1 above). So our look-up tables, which were calculated for no-eclipse conditions, are also suitable for calculating TOCs during the eclipse.

We did not calculate TOCs from measurements of direct irradiance because such retrievals have a comparatively large uncertain when direct irradiance in the UV-B (in particular 305 nm) is calculated from shadowband data. In contrast to wavelengths in the UV-A and visible, where direct irradiance under clear skies with moderate aerosol loading makes up a large fraction of the global irradiance, this is not true for wavelengths in the UV-B, in particular at large SZAs (see Fig. 1a). When using a shadowband, direct irradiance in the UV-B is calculated as the difference of two large numbers (global and diffuse irradiance), and the uncertainty of the result is therefore higher than those of TOCs derived from global irradiance.

Regarding the referee's comment:

> "the comparison with ozone values or previous studies derived from instruments measuring the direct irradiance should be done under the acknowledgement that these measurements are correct not only because they are the standard ones but also because they are correct in terms of physics and the best in terms of overall uncertainty."

While we agree that measuring TOC from direct irradiance is the more straightforward method (as only Beer-Lambert's law is involved), the referee's assertion that "these measurements are correct" is not appropriate. There are many error sources that can affect these measurements such as pointing errors, calibration errors, non-linear sensor response, stray light from diffuse radiation that is detected with the direct beam (a problem affecting Brewer measurements as mentioned by several papers cited in our manuscript), uncertainty of the filter functions and their center wavelengths, etc. Considering that we do not have the raw data that other authors have collected to calculate TOCs during historical ellipses we do not want to speculate on the uncertainty of those measurements and their magnitude relative to our results.

As explained above, adding TOC data derived from different wavelength pairs would not change our conclusions about the effect of the solar eclipse on short-term TOC fluctuations. However, results obtained with the 305 / 313 and 313 / 340 pairs are valuable because they confirm the results from the 305 / 340 pair discussed in the manuscript. In order to keep the length of the manuscript within reasonable limits while also respecting the referee's suggestion, we added these new results to the Supplement with references provided in the manuscript.

**Change to manuscript**

The following was added to Sect. 5.2:

> "The following three wavelength pairs were used: (1) $\lambda_L = 340$ nm, $\lambda_S = 305$ nm; (2) $\lambda_L = 340$ nm, $\lambda_S = 313$ nm; (3) $\lambda_L = 313$ nm, $\lambda_S = 305$ nm.
>
> […]

"While TOCs could also be derived from direct irradiances, these measurements are not discussed here due to the relatively large uncertainty to calculate direct irradiance from shadowband data at wavelengths in the UV-B (280–315 nm), in particular at large SZAs."

The following was added to Sect: 7.4:

"Results obtained from the 305 / 340 wavelength pair are discussed below. Similar results calculated with the 305 / 313 and 313 / 340 wavelength pairs are presented in Sect. S2 of the Supplement."

[…]

Results obtained with the 305 / 313 and 313 / 340 wavelength pairs (Sect. S2 of the Supplement) corroborate these findings."

**Change to Supplement:**

Calculations of TOC from the 305 / 313 and 313 / 340 wavelength pairs have been added to the Supplement. Please see Section S2 of the new version of the Supplement for details.
* * *
**Comment by Referee**

Figure 9 and relevant text: Kazantzidis et al (2007), when using the 305/320 wavelength ratio (in order to reduce the effect of spectral effect of the eclipse on direct and diffuse irradiances) reported very similar results to those derived from yours when using the 305/340 wavelength ratio AND applying the Neckel or Pierce parameterization. Surprisingly, this paper is referenced only for the comparison of measured and model irradiances. However, the results of this paper for ozone, irradiances and irradiance ratios vs eclipse percentage are not referenced, although it is based on results from 8 narrowband multi-channel NILU-UV6 radiometers.

**Authors' response and change to manuscript**

As mentioned already earlier, we have now added the following to the manuscript to compare our ozone measurements with those reported by Kazantzidis et al (2007):

"Finally, Kazantzidis et al (2007) discuss TOC measurements performed with NILU-UV filter radiometers at several locations in Greece during the total solar eclipse of 29 March 2006. They did not observe any periodic fluctuations in TOC and only report a small increase in TOC of about 5 DU as the visible fraction of the Sun decreases from ~60% to ~20%. This small change could be caused by incomplete correction of the LD effect."

Furthermore, we have added the following to Sect. 8.2 (now Sect. 8.1) where we compare the magnitude of solar LD used in our paper with that used by other authors:

"Kazantzidis et al.(2007) have analyzed ratios of global spectral irradiance (305 nm / 380 nm, 312 nm / 380 nm, 340 nm / 380 nm, and PAR / 380 nm) that were measured with NILU-UV filter radiometers at three locations in Greece during the total solar eclipse of 29 March 2006. These measured ratios were compared with theoretical predictions based on the algorithm by Koepke et al. (2001) and the LD parameterization by Waldmeier. As the eclipse progressed, the model underestimated the measured spectral effect, capturing only half of the observed change. For example, measured ratios of spectral irradiances at 340 and 380 nm were 10 % lower close to totality compared to similar ratios calculated for the 1$^{st}$ and 4$^{th}$ contact. The theoretical calculation only predicted a decrease of 5%. Our results suggests that discrepancies between the measured and modeled ratios reported by Kazantzidis et al. (2007) can partly be attributed to limitations of the LD parameterization by Waldmeier used in their model."
* * *
**Comment by Referee**

Paragraph 8.1: the authors seem to have a point here. In order to better understand the similarities/differences with previous studies, a more detailed information is needed apart from direct/diffuse ratios: the measured wavelengths and the eclipse percentages should be provided. Moreover, the theoretical calculations from Emde and Mayer are quite capable to estimate the global irradiance (when normalized 5 minutes before totality) at 380 nm but maybe irradiance is significantly underestimated at 312 nm (Kazantzidis et al., 2007, figure 7 and relevant text). This affects directly the diffuse component. Of course, also this result is sensitive to factors like surface albedo, ozone profile and the dynamic range of the measuring system. All these factors should be mentioned.

**Authors' response**

We added the following sentence to Section 8.1. (now Sect. 8.2), which now specifies the measured wavelengths (i.e. erythemal irradiance), eclipse percentage (88%), location (Thessaloniki), and time (11 August 1999):

"However, these results disagree with the results by Zerefos et al. (2000; 2001), who suggest that the erythemal (sunburning) diffuse irradiance was declining at a slower rate than the erythemal direct irradiance during the solar eclipse observed in Thessaloniki, Greece, on 11 August 1999. The largest difference was observed at the time of the eclipse maximum when 88% of the solar disk was obscured and the diffuse irradiance was reduced 30 % less than the direct irradiance."

With regards to factors such as "surface albedo, ozone profile and the dynamic range of the measuring system" that affect irradiance close to totality, we note that we concluded our original manuscript with the sentence:

"During totality, the irradiance at the surface will become also more sensitive to the topography (e.g., the mountains surrounding the measurement sites), surface albedo (and its spectral dependence), and the distribution of ozone in the atmosphere (the ozone profile). These aspects will be discussed in a follow-on publication."

The "follow-on publication" mentioned here is close to completion and will be submitted soon. The factors enumerated by the referee will be discussed in detail therein.
* * *
**Comment by Referee**

Page 22-23: This comment refers to the ozone issue, described in detail by the authors. From my point of view, some (or much?) of these differences could be attributed on the measuring methods and the selected pairs of wavelengths. As mentioned before, a decrease of (uncorrected for LD effect) ozone retrievals has been reported by Kazantzidis et al (2007) for the same eclipse when using 8 NILU-UVs. As it was stated earlier in this review, the authors could strongly defend their findings if they will come up with the same results when using other wavelength pairs and the direct GUV irradiances. Unfortunately, this measuring campaign is not accompanied by more instruments.

**Authors' response**

One important "take-home message" from our paper is that accurate LD corrections are necessary for accurate TOC retrievals. Basically all papers that have discussed TOC measurements for previous eclipses use a LD correction that is based on a parameterization that is too simple for wavelengths in the UV. In addition to TOC retrievals using the 305 / 340 wavelength pair, we have now also calculated TOCs from the 305 / 313 and 313 / 340 wavelength pairs (Fig. 3

above). These new calculation corroborate our initial findings that the spurious peak in TOC measurements caused by LD effect can be greatly reduced by using either the LD parameterization by Neckel or Pierce. We agree with the referee's assessment that the presence of additional instrument during "our" eclipse could have strengthened our conclusions further. We hope that a multi-instrument campaign can be organized during an upcoming eclipse to get closure on some of the remaining questions raised in our paper.

**References**

Hooker, S. B., Bernhard, G., Morrow, J. H., Booth, C. R., Comer, T., Lind, R. N., and Quang, V.: Optical Sensors for Planetary Radiant Energy (OSPREy): calibration and Validation of Current and Next-Generation NASA Missions., NASA Goddard Space Flight Center, NASA/TM–2011–215872, 2012.

Kazantzidis, A., Bais, A. F., Emde, C., Kazadzis, S., and Zerefos, C. S.: Attenuation of global ultraviolet and visible irradiance over Greece during the total solar eclipse of 29 March 2006, Atmos. Chem. Phys., 7(23), 5959–5969, https://doi.org/10.5194/acp-7-5959-2007, 2007.

Koepke, P., Reuder, J., and Schween, J.: Spectral variation of the solar radiation during an eclipse, Meteorol. Z., 10(3), 179–186, https://doi.org/10.1127/0941-2948/2001/0010-0179, 2001.

Krotkov, Nickolay A., Pawan K. Bhartia, Jay R. Herman, James R. Slusser, Gordon J. Labow, Gwendolyn R. Scott, George T. Janson, Tom Eck, and Brent N. Holben. "Aerosol ultraviolet absorption experiment (2002 to 2004), part 1: ultraviolet multifilter rotating shadowband radiometer calibration and intercomparison with CIMEL sunphotometers." *Optical Engineering* 44, no. 4 (2005): 041004.

Morrow, J. H., Hooker, S. B., Booth, C. R., Bernhard, G., Lind, R. N., and Brown, J. W.: Advances in measuring the apparent optical properties (AOPs) of optically complex waters, NASA/TM–2010–215856, National Aeronautics and Space Administration, Goddard Space Flight Center, 2010.

Witthuhn, J., Deneke, H., Macke, A., and Bernhard, G.: Algorithms and uncertainties for the determination of multispectral irradiance components and aerosol optical depth from a shipborne rotating shadowband radiometer, Atmos. Chem. Phys., 10(2), 709–730, https://doi.org/10.5194/amt-10-709-2017, 2017.

Zerefos, C. S., Balis, D. S., Meleti, C., Bais, A. F., Tourpali, K., Kourtidis, K., K. Vanicek, K., Cappellani, F., Kaminski, U., Colombo, T., and Stübi, R.: Changes in surface solar UV

irradiances and total ozone during the solar eclipse of August 11, 1999, J. Geophys. Res., 105(D21), 26463–26473, https://doi.org/10.1029/2000JD900412, 2000.

Zerefos, C. S., Balis, D. S., Zanis, P., Meleti, C., Bais, A. F., Tourpali, K., D.Melas, D., Ziomas I., Galani E., Kourtidis, K., Papayannis A., and Gogosheva Z.:. Changes in surface UV solar irradiance and ozone over the Balkans during the eclipse of August 11, 1999, Adv. Space Res., 27(12), 1955–1963, https://doi.org/10.1016/S0273-1177(01)00279-4, 2001.

---

## Author Comment (AC4) · 12 Mar 2019

**Response to 2nd comments by Forrest M. Mims III, posted on 21 Dec 2018**

**General remarks to comment by reviewer**

We thank Mr. Mims for his additional comments but feel that many remarks are beyond the scope of the paper. Our paper is about the importance of applying corrections for solar limb darkening when observing a solar eclipse; the question of whether or not bow waves from the Moon's shadow may result in fluctuations in total ozone column (TOC); and the question of whether or not the ratio of direct-to-diffuse irradiance changes appreciably during the period of a solar eclipse (excluding the period near totality). The paper is NOT about the best, most accurate, or most precise method to measure TOC.

In our response to the first comments by the reviewer (posted on 24 November 2018), we provided new evidence that our method of measuring TOC during the eclipse is precise enough for detecting potential changes in TOC from bow waves. In brief, we concluded that the noise in our measurements is low enough for detecting relative changes in TOC of larger than 0.05 %. In addition, we calculated that changing aerosol concentrations during the time of our observations result in an additional uncertainty in TOC of 1.5 DU or 0.5 %. Even if our uncertainty estimates were too optimistic, for example due to an unknown systematic error in our TOC retrieval method, it would be highly unlikely that variations in our calculated TOC values would anti-correlate with real variations in TOC triggered by bow waves such that the resulting TOC measurements after the 3rd contact become basically flat, with a variation of only ±1 DU or ±0.33 % (see Fig. 9 of original manuscript). For comparison, the peak-to-peak amplitude attributed to bow waves reported by Zerefos et al. (2007) was 2.0–3.5 %, and the peak-to-peak amplitude reported by Mims and Mims (1993) was 1.7 %.

In conclusion, we cannot rule out that that the Moon's shadow led to variations of TOC in the order of ±0.3 % during the eclipse observed by us, but note that this upper limit is considerably lower than the fluctuations reported by Zerefos et al. (2007) and Mims and Mims (1993).

**Changes to manuscript**

In the abstract, the following sentence:

"In contrast to results of observations from earlier solar eclipses, no fluctuations in TOC were observed that could be attributed to gravity waves."

will be replaced with:

"In contrast to results of observations from earlier solar eclipses, no fluctuations in TOC

were observed that could be unambiguously attributed to gravity waves."

In Sect. 8.3., the following sentence:

"Our data do not support the observation by Zerefos (2000; 2007) and Mims and Mims (1993) that bow waves from the Moon's shadow lead to oscillations in TOC."

will be replaced with:

"Our data do not support the observation by Zerefos (2007) and Mims and Mims (1993)

that bow waves from the Moon's shadow may lead to oscillations in TOC with a peak-to- peak amplitude exceeding 1.5 %."

**Comment by reviewer:**

TWO KEY POINTS: The authors have made several important revisions to their paper, but they simply must remove their erroneous assertions to the effect that: (1) the signal at 300 nm is noisy, which suggests poor measurements by TOPS (which used 300 nm and 305 nm) and (2) full-sky measurements of the ozone layer are "similar" to the direct sun measurements employed by

Dobsons (and TOPS, Microtops, Brewers and Pandoras). These inappropriate claims and their 2- minute measurement time support their general assertions that raises doubts about the papers by me and others.

**Authors' Response**

Regarding (1): We do not assert anywhere in the paper that signals of the TOPS instruments are noisy or that measurements of this instrument are of poor quality. In fact, we do not even mention

"TOPS" in the manuscript. The discussion of the noise characteristics of the TOPS instrument is only part of the reviewer's first post and our response. It is and will not be part of the paper.

Regarding (2): We do not state in the manuscript that "full-sky measurements of the ozone layer are 'similar' to the direct sun measurements employed by Dobsons (and TOPS, Microtops,

Brewers and Pandoras)". We do not compare the accuracy of our TOC measurements with that of other methods. Like in the case of (1), the discussion on the quality of the different methods of measuring ozone is only part of the reviewer's first post and our response.

**Changes to manuscript**

None.
* * *
**Comment by reviewer:**

1. OPTIMUM WAVELENGTH SELECTION FOR MEASURING TOTAL COLUMN OZONE DURING A SOLAR ECLIPSE: The authors claim the 300-nm minimum employed by TOPS provides a noisy signal. They erroneously observe that: "Of note, the shortest wavelength of a Dobson is 305.5 nm and if measurements at 300 nm would be of great advantage, these instruments would likely use a shorter wavelength." The 305.5-nm Dobson minimum wavelength was selected due to the use of the instrument across a wide band of latitudes. However, there is ample signal at 300 nm at my site (29.9 N), as demonstrated by the instrument's detection of an error in NASA's Nimbus-7 TOMS ozone instrument (Mims, Nature, 1993). The 300 nm signal was especially strong at 22 N during the 1991 solar eclipse reported in my paper in GRL. The authors might be right about the 300-nm signal at their northerly location. But they cannot compare what might have been a noisy 300 nm signal at their northerly 44.36 N site to the much higher amplitude 300-nm signal at the 22 N site for the eclipse I measured. Furthermore, DeLuisi and others have demonstrated that wavelengths below 305 nm provide more accurate ozone measurements than higher wavelengths. (This explains why TOPS found the satellite error.)

**Authors' Response**

Similar to the last comment, we do not discuss the TOPS instrument in our paper. The quote in the reviewer's comment above is again from our response to the reviewer's first post. Also, we do not discuss the best wavelengths to be used to calculate TOC for the location of "our" eclipse. We simply state in the manuscript that TOCs were calculated from the measurements of the GUVis-3511's channels at 305 and 340 nm.

**Changes to manuscript**

None.
* * *
**Comment by reviewer:**

Of special concern is this from the author's abstract: "The total ozone column (TOC) was derived from measurements of global irradiance at 306 and 340nm." While the 306 nm minimum is appropriate for 44.36 N, the 340 nm upper wavelength is far too high, for it allows for significant aerosol errors. The closely-spaced 300 nm and 305 nm wavelengths of TOPS nearly eliminated the aerosol error, which explains this instrument's excellent accuracy when compared with hundreds of satellite measurements and an EPA Brewer for 60 days at my site. Consider this abstract by Saunders et al in High-Precision Atmospheric Ozone Measurements Using … trial spectral irradiances between 290 and 305 nm (JGR Atmospheres 1984 https://doi.org/10.1029/JD089iD04p05215 ) "Abstract "It is shown theoretically that many errors are significantly less when determining atmospheric ozone thicknesses from measurements of solar terrestrial spectral irradiance in the wavelength region between 290 and 305 nm as compared to the to the 305- to 340-nm region employed by the Dobson spectrophotometer. In order to test this conclusion experimentally, an elaborate set of state-of-the-art measurements have been made in the shorter wavelength region in Gainesville, Florida, between June 13 and June 18, 1980. Details of these measurements, including an extensive error analysis, are presented and indicate that such short-wavelength measurements, particularly between 295 and 305 nm, can be used to detect long-term changes of atmospheric ozone with an uncertainty not exceeding 1%. Observing conditions restricted the Gainesville measurements to zenith angles of less than 35°. Further investigations are required to determine the shortest wavelength that can be used at significantly greater zenith angles."

**Authors' Response**

We agree with the reviewer that aerosols will lead to systematic errors in TOC measurements if the wavelengths used for the retrieval are far apart. However, we have estimated the uncertainty in our TOC retrievals for the period of interest and have concluded that variations in aerosols during the period of the eclipse cause an uncertainty in TOC of only 1.5 DU or 0.5 %. We note that this uncertainty refers to the precision of TOC measurements (the metric of relevance to the paper), not absolute accuracy, which could be worse. However, our TOC retrievals agree to within 3 DU (or 1%) with OMI, suggesting that also the accuracy of our data is within acceptable limits.

The discussion of whether or not TOC measurements should be based on wavelengths in the 290 to 305 nm range is of little relevance to the paper because the GUVis-3511 radiometer has no channels with wavelengths below 305 nm. In addition, while TOC measurements using wavelengths shorter than 305 nm could indeed be more accurate, as the JGR quoted by the reviewer suggests, we like to point out that a higher accuracy can only be achieved if the filters of the instrument in question are well characterized. No filter instrument measures at exactly a nominal wavelength and all real instruments (including the TOPS) use filters with a finite bandpass. If a hypothetical channel that is supposed to measure at 300 nm measures in fact at 300.5 nm, TOC errors will result. We are not implying that this is the case for the TOPS instrument, but just like to point out that performing ozone measurements with channels at 300 and 305 nm can potentially lead to errors that could be comparable in magnitude to those affecting measurements using channels that are farther apart, and as a result are more sensitive to aerosols. Again, we cannot, and do not want to, assess the quality of TOPS measurements because we do not use data of this instrument in the paper and are not familiar with its characteristics.

The sentence "The total ozone column (TOC) was derived from measurements of global irradiance at 306 and 340nm." in the abstract does not judge whether this wavelength selection is the most appropriate. It simply states what was done and we see no reason to change it.

**Changes to manuscript**

None.
* * *
**Comment by reviewer:**

2. FULL-SKY VS. DIRECT SUN TOTAL OZONE MEASUREMENTS: The authors state in their paper: "The method of calculating TOC from measurements of global irradiance (instead of direct irradiance as it is typically done for Dobson, Brewer, TOPS, and Microtops instruments) was first proposed by Stamnes et al. (1991). We found that the accuracy of TOCs derived from global irradiance is similar to that of data from Dobson instruments or satellite (TOMS, OMI) observations if the look-up table takes local conditions into account (ozone profile, albedo, elevation, etc.) (Bernhard et al., 2005b)." The authors suggest that global irradiance provides TOC data ". . . similar to that of data from Dobson instruments or satellite. . . ." But the authors provide no data or citations to support this assertion, while leaving open the counter suggestion that Dobsons (and TOPS, Brewers and Pandoras) could be replaced by much simpler instruments that measure global irradiance and require no tracking and pointing.

**Authors' Response**

We do not state in our paper:

"The method of calculating TOC from measurements of global irradiance (instead of
direct irradiance as it is typically done for Dobson, Brewer, TOPS, and Microtops
instruments) was first proposed by Stamnes et al. (1991). We found that the accuracy of
TOCs derived from global irradiance is similar to that of data from Dobson instruments
or satellite (TOMS, OMI) observations if the look-up table takes local conditions into
account (ozone profile, albedo, elevation, etc.) (Bernhard et al., 2005b)."
as asserted in the comment by the reviewer above. This quote is taken from our comment to the
first post of the reviewer. Instead, we simply state in the paper:
"This method was first proposed by Stamnes et al. (1991) and was validated for GUV
instruments by Bernhard et al. (2005a)."
We are puzzled by the reviewer's comment:
"The authors suggest that global irradiance provides TOC data ". . . similar to that of data
from Dobson instruments or satellite. . . ." But the authors provide no data or citations to
support this assertion, while leaving open the counter suggestion that Dobsons (and
TOPS, Brewers and Pandoras) could be replaced by much simpler instruments that
measure global irradiance and require no tracking and pointing."
because the paper and the response to the post of the reviewer does include two citations (i.e.,
Bernhard et al., 2005a, and Bernhard et al., 2005b) where TOC measurements from global
irradiance are compared with Dobson direct measurements and satellite (TOMS, OMI)
observations.
In Bernhard et al., 2005b, we conclude that:
"When Dobson measurements are corrected for the temperature dependence of the ozone
absorption cross section and accurate air mass calculations are implemented, data from
the three instruments agree with each other to within ±2% on average and show no
significant dependence on SZA or total ozone."
The "three instruments" quoted above refer to the Dobson; our SUV-100 spectroradiometer,
which measures global irradiance; and TOMS. The reviewer may not agree with this conclusion,
but it is false to assert that "the authors provide no data or citations to support this assertion".
We further like to point out that the method of retrieving TOC from spectra of global irradiance
measured by SUV-100 radiometers has been published by Bernhard et al. (2003). This paper also
includes an uncertainty estimate of the method. The abstract ends with:

"On average, the new algorithm generates ozone values in spring 2.2 % lower than TOMS observations and 1.8 % higher than Dobson measurements. From the uncertainty budget and the comparison with TOMS and Dobson it can be concluded that ozone values retrieved from global UV spectra have a similar accuracy as observations with standard instrumentation used for ozone monitoring."

**Changes to manuscript**

In Sect. 5.2., we will replace

"This method was first proposed by Stamnes et al. (1991) and was validated for GUV instruments by Bernhard et al. (2005a)"

with

"The method of calculating TOC from measurements of global irradiance was first proposed by Stamnes et al. (1991) and was further validated by Bernhard et al. (2003; 2005b). The application of the method to GUV instruments was described by Bernhard et al. (2005a)."
* * *
**Comment by reviewer:**

Of course, this is not the case. The authors have gone much, much too far in suggesting their global, full-sky data is "similar" to direct sun measurements made by the recognized standard for nearly a century and all other ozone instruments. I have always been intrigued by the prospect of accurate ozone retrievals by pairs of global UV measurements at closely-space wavelengths. I urge the authors to prepare a detailed paper about their claim that global TOC measurements are "similar" to those by traditional direct sun instruments. An ideal comparison would be with Brewers, which measure both direct sun and global TOC. Meanwhile, it is inappropriate to make an unsupported claim about "similar" results.

**Authors' Response**

The reviewer suggests to prepare a detailed paper about our claim that global TOC measurements are "similar" to those by traditional direct sun instruments. This paper has already been written. In fact, there are two: Bernhard et al. (2003; 2005b) and we therefore don't agree with the reviewer's conclusion that it is inappropriate to make an unsupported claim about "similar" results.

We do not advocate in any of our papers that the established network of Dobson and Brewer instrument should be replaced with instruments measuring global irradiance. The "global irradiance method" is only sufficiently accurate if the ozone profile is known with sufficient accuracy, in particular for large solar zenith angles. So the method is not as independent as direct measurements that rely on Beer-Lambert's law and hence do not require knowledge of the profile. However, when there is cloud cover, the direct method cannot be used and Stamnes et al. (1991) showed that the "global" method is equally accurate as the zenith sky method used by Dobsons when the Sun is obstructed by clouds.

As a side note, because TOC retrievals from global irradiance depend on the ozone profile, the profile can in fact be determined from such measurements using a variant of the Umkehr method. We have recently published this "Global-Umkehr method" (Bernhard et al., 2017). Again, we not propose that this method is superior to the standard Umkehr method, which relies on zenith sky observations. The paper is just a prove of concept and makes the Umkehr method available to locations with global irradiance measurements. The abstract of Bernhard et al. (2017) concludes with "Total ozone columns (TOCs) calculated from the retrieved profiles agree to within 0:7±2:0 % (±1σ) with TOCs measured by the Ozone Monitoring Instrument on board the Aura satellite." This demonstrates again the good accuracy of the TOC retrievals from global measurements if its done right.

**Changes to manuscript**

None.
* * *
Finally, we like to conclude that most of the discussion above is of little relevance to our paper. The only important question is whether or not our TOC retrievals from global irradiance measurements of the GUVis-3511 are of sufficient precision to detect fluctuations in TOC that could be triggered by bow waves. Using a sensitivity analysis, we concluded that this is the case. Based on our results, we cannot rule out that there is an effect with an amplitude of ±1 DU or ±0.33 %. We simply conclude that we did not observe fluctuations in TOC that could be unambiguously attributed to gravity waves. It is possible that the magnitude of gravity wave effects varies from eclipse to eclipse, explaining the discrepancy in the data by us, Zerefos et al. (2007) and Mims and Mims (1993), but this is speculation.

**Changes to manuscript**

None.

**References**

Bernhard, G., Booth, C. R. and McPeters R. D., Calculation of total column ozone from global

UV spectra at high latitudes, J. Geophys. Res., 108(D17), 4532, doi:10.1029/2003JD003450,

2003.

Bernhard, G., Booth, C. R., and Ehramjian, J. C.:. Real-time ultraviolet and column ozone from multichannel ultraviolet radiometers deployed in the National Science Foundation's ultraviolet monitoring network, Opt. Eng., 44(4), 041011-1, https://doi.org/10.1117/1.1887195, 2005a.

Bernhard, G., Evans, R. D., Labow, G. J., and Oltmans, S. J.: Bias in Dobson total ozone measurements at high latitudes due to approximations in calculations of ozone absorption coefficients and air mass, J. of Geophys. Res., 110, D10305, https://doi.org/10.1029/2004JD005559, 2005b.

Bernhard, G., Petropavlovskikh, I., and Mayer, B.: Retrieving vertical ozone profiles from measurements of global spectral irradiance, Atmos. Meas. Tech., 10(12), 4979-4994, https://doi.org/10.5194/amt-10-4979-2017, 2017.

Mims III, F. M. and Mims, E. R.: Fluctuations in column ozone during the total solar eclipse of

July 11, 1991, Geophys. Res. Lett., 20(5), 367–370, https://doi.org/10.1029/93GL00493,

1993.

Stamnes, K., Slusser, J., and Bowen, M.: Derivation of total ozone abundance and cloud effects from spectral irradiance measurements, Appl. Opt., 30(30), 4418–4426, https://doi.org/10.1364/AO.30.004418, 1991.

Zerefos, C. S., Balis, D. S., Meleti, C., Bais, A. F., Tourpali, K., Kourtidis, K., K. Vanicek, K.,

Cappellani, F., Kaminski, U., Colombo, T., and Stübi, R.: Changes in surface solar UV

irradiances and total ozone during the solar eclipse of August 11, 1999, J. Geophys. Res.,

105(D21), 26463–26473, https://doi.org/10.1029/2000JD900412, 2000.

Zerefos C. S., Gerasopoulos E., Tsagouri I., Psiloglou B. E., Belehaki A., Herekakis T., Bais A.,

Kazadzis S., Eleftheratos C., Kalivitis N., and Mihalopoulos N.: Evidence of gravity waves into the atmosphere during the March 2006 total solar eclipse, Atmos. Chem. Phys., 7(18),

4943–4951, https://doi.org/10.5194/acp-7-4943-2007, 2007.

---

## Author Response (AR2)

**Response to Comments of Reviewers of Manuscript:**

**Measurements of spectral irradiance during the solar**

**eclipse of 21 August 2017: reassessment of the effect of**

**solar limb darkening and of changes in total ozone**

**Table of Contents**

A marked-up manuscript version of the manuscript is provided from page 48 onward.

**Response to comments by Editor, on 13 March 2019**

We thank the editor for his encouraging comments regarding the new version of our manuscript and are pleased to hear that we have satisfactorily responded to the comments by the referees.

All references to figures and other material (e.g., page and line numbers) provided in the following response to the editor's comments refer to the version of our manuscript that we have uploaded on 12 March 2019.

**Comment by Editor**

Concerning the statements about the diffuse to direct ratio. Reading the paper it is not clear to me from the measurements if this is increasing or not during the eclipse. Looking at figure 10, what it looks like is that direct to global ratio is slightly decreasing at all wavelengths close to the eclipse time. This means that the diffuse to global ratios has to slightly increase. Is the above statement more or less correct ?

**Authors' Response**

We presume that the question is with respect to Fig. 11, not Fig. 10, of the latest version of the manuscript. Yes, Figure 11 (and in particular Panel b of that figure) indicates that the direct-to-global ratio is decreasing during the eclipse period. However, the minimum ratios are not observed at the time near totality but about 30 minutes before totality (see also P18, L1).  Since global irradiance = direct irradiance + diffuse irradiance, this implies that the diffuse-to-direct and diffuse-to-global ratios increase during the eclipse period. As described in the manuscript (P18, L6-10 and P24, 8-11), the available evidence suggests that the decrease in the direct-to-global ratio and the concomitant increase in the diffuse-to-direct ratio are caused by aerosols, not by eclipse effects. However, we acknowledge (P24, L10-11) that "aerosol effects are difficult to decouple from processes initiated by the occlusion of the Sun or artifacts of the algorithm used to calculate the direct irradiance from shadowband data" and our assessment is therefore subject to uncertainties. On the other hand, according to 3D radiative transfer (3DRT) calculations by Emde and Mayer (2007), eclipse effects that affect direct and diffuse radiation differently are smaller than 1 % for times 10 minutes away from totality and smaller than 4 % for times 105 seconds away from totality (i.e., the shortest time to totality available from our shadowband measurements) (P24, L13-15). These calculations support our assessment that changes in the direct-to-global ratio were caused by aerosols.

In Sect. 8.2. (P24, L17-22), we stated that theoretical calculations [using 3D radiative
transfer models]

"disagree with the results by Zerefos et al. (2000; 2001), who suggest that the erythemal
(sunburning) diffuse irradiance was declining at a slower rate than the erythemal direct
irradiance during the solar eclipse observed in Thessaloniki, Greece, on 11 August 1999.
The largest difference was observed at the time of the eclipse maximum when 88% of the
solar disk was obscured and the diffuse irradiance was reduced 30 % less than the direct
irradiance. The available evidence from our observations and model calculations suggests
that the large change of the diffuse-to-direct ratio reported by Zerefos et al. (2000; 2001)
may have been a measurement artifact."

Considering that "our observations" refer to the direct-to-global ratio while the results by Zerefos
et al. (2000) refer to the diffuse-to-direct ratio, we performed additional calculations so that our
results can be directly compared with those by Zerefos et al. (2000). Specifically, we calculated
the diffuse-to-direct ratio at 315 nm (the wavelength most representative of the erythemal range)
from our measurements and compared this ratio with the corresponding ratio calculated with our
radiative transfer model. Model results were corrected for the bias between measurement and
model in the same way as the direct-to-global ratios shown in Fig. 11 of the manuscript.

Panel a of Fig. E1 below compares the measured and modeled diffuse-to-direct ratio while Panel
b shows the ratio of measurement and model, analogously to Fig. 11. Panel a can be directly
compared with Fig. 3a of Zerefos et al. (2000), which is also reproduced on the next page. The
following can be learned from the comparison of our results and those by Zerefos et al. (2000):

▪   While the measured diffuse-to-direct ratio increases during both eclipses, the increase in
our data is more broad and peaks about 30 minutes before the time of totality. In contrast,
data shown by Zerefos et al. (2000) are more "pointy" and peak at the time of the eclipse
maximum, which is 11:05 according to Zerefos et al. (2000).

▪   According to our data, the maximum diffuse-to-direct ratio is 31 %, while the same value
estimated from Fig. 3a of Zerefos et al. (2000) is 50 %.

[Figure]

Figure E1: Panel (a): ratio of diffuse-to-direct spectral irradiance. Measurements are shown as line. The bias-corrected model is indicated with a broken line. Panel (b): Ratio of the measured and modeled direct- to-global ratios, corrected for the bias between measurement and model.

[Figure]

Figure 3a of Zerefos et al. (2000), showing the ratio of diffuse-to-direct erythemal solar irradiance for

August 11, 1999, measured at Thessaloniki (solid line) and estimated from model calculations (dashed line).

**Change to manuscript**

Considering that aerosol introduce uncertainty in our assessment of whether or not the eclipse had a significant impact on the direct-to-global and diffuse-to-direct ratios, we weakened the statements where we compare our results with those by Zerefos et al. (2000, 2001).

In Sect. 8.2., we replaced the paragraph quoted above (P24, L17-22) with:

"These theoretical results disagree with the results by Zerefos et al. (2000; 2001), who suggest that the erythemal (sunburning) diffuse irradiance was declining at a slower rate than the erythemal direct irradiance during s solar eclipse observed in Thessaloniki, Greece, on 11 August 1999. The largest difference was observed at the time of the eclipse maximum when 88 % of the solar disk was obscured and the ratio of diffuse to direct irradiance was increased by approximately 50 % relative to the change expected from calculations with a one-dimensional radiative transfer model. For comparison, the increase in the diffuse-to-direct ratio calculated at 315 nm from our measurements was only 31 % and did not peak at the time of totality. The available evidence from our observations and 3DRT model calculations suggests that measurement artifact may have contributed to the large change of the diffuse-to-direct ratio reported by Zerefos et al. (2000; 2001)."

**Comment by Editor**

There was a lot of arguing against the non use of direct sun measurements and the choice of the 305nm and 340nm ( well, it was not actually a choice but what was available..). In addition, there is a number of other processes like the channel width (10nm FWHM) procedure to retrieve the 1nm spectrum and finally the FOV. FOV of the shadowband is difficult to simulate. And for sure it is larger than an "ideal" FOV of 0.5 degrees or even a 1.5-2.5 degree FOV that most of direct sun pointing AOT measuring instruments use. Correction techniques and calibration principles can improve a lot the AOT results but in the case of the eclipse things are much more complicated, especially at (low signal) lower wavelengths.

Could you comment on the above ?

**Authors' Response**

We agree that it would have been great if our (unfunded) measurement campaign would have also include Sun-pointing instruments with narrow FOV. Such measurements would have afforded to calculate AODs with lower uncertainty and would have also helped to corroborate our global and diffuse shadowband measurements. While we did not have the luxury of deploying additional instrument, we tried our best to keep the uncertainties of our measurements at low as possible, and documented these uncertainties. For example,

- we developed a new method (Section S1 of Supplement) to convert our 10 nm FWHM measurements to 1 nm FWHM measurements and calculated the uncertainty of spectral irradiances calibrated with this method (Section S.1.3 of Supplement),
- used a relatively new data analysis method to minimize uncertainties from the relatively large FOV of the shadowband (see P5, L26 – P6, L9),
- and used an instrument that has low noise levels (and high signal-to-noise ratios), which allowed measurements with sufficient precision at all times, with the exception of measurements of the two 305 nm channels during totality (Sect. 7.5 and 8.4).

All these elements have been discussed in the manuscript and we believe that no changes to the manuscript are necessary to further emphasize these aspects. Furthermore, we do not hide that there are limitations to our measurements and suggests (P26, L10-13) that remaining questions could be settled "by performing LD-corrected measurements of TOC with different instrument types during one of the upcoming solar eclipses."

**Change to manuscript**

None.
* * *
**Comment by Editor**

Finally, the ozone issue is a big puzzle for me. Kazadzis, Kazantzidis, 2 papers of Zerefos and Groebner studies all pointed an artificial drop of total column ozone. On the contrary you find an increase. the limb darkening corrections and explanations of all the previous papers can be improved as you say. But the fact remains that a number (about 10) of instruments at different locations (Brewer and NILUs) performing automatic measurements are providing the same artificial decline of ozone.

Would it be possible to explain why this is not the case here ?

Is it the wavelength choice ?

**3    Authors' Response**

First, we like to point out that the LD effect increases with decreasing wavelength below the

"Balmer break" at about 364.6 nm. Hence, if only a small sliver of the Sun is visible, the extraterrestrial spectrum is more reduced at shorter wavelengths than at longer wavelengths. This is evident both from Fig. 7 of our paper and from Fig. 6 of Neckel (2005). If no correction for LD

is applied, measurements at a short wavelength are therefore more suppressed during an eclipse than during the no-eclipse case, which should result in *larger* TOCs retrievals. So our results are consistent with theory. If some instruments report *smaller* TOCs during an eclipse, this would either mean that TOCs have indeed decreased during an eclipse (which is very unlikely, see Sect.

8.3) or measurements are affected by a systematic error (either in the raw data or in the data processing routine), which would be larger than the LD effect and goes in the opposite direction.

Let us discuss results from previous eclipses:

▪  **Bojkov (1968)**

As mentioned in the manuscript (P25, L3-5), the LD correction applied to Dobson spectrophotometer data from the solar eclipse of 20 May 1966 discussed of Bojkov (1968) reduces the TOC by up to 6 %, in good agreement with our calculations. Fig. 3 of the paper by Bojkov (1968) is reproduced below.

[Figure]

*Fig. 3.* Changes in total ozone calculated by $C_z$ wavelength measurement (-------), and the same after dark-limb correction ( × × × ).

It shows that the TOC (derived from the Dobson C wavelength pair; 311.2 nm / 332.3 nm) was increased close to the eclipse maximum (as in our observations) and that the LD correction reduces the effect. Unfortunately, the paper is very short and there are no useful details on the LD correction. Still, this is a valuable example from Dobson measurements that are by and large consistent with our data.

- **Gröbner et al., 2017**

Gröbner et al. (2017) retrieved TOC from Brewer spectrophotometers using a weighted mean of the direct solar irradiance measurements at 310.1 nm, 313.5 nm, 316.8 nm, and 320.0 nm. They observed a systematic decrease of up to 15 DU during a partial eclipse of 20 March 2015. According to Gröbner et al. (2017), this decrease can be explained to a large extent by LD. Their results "demonstrate that the largest contribution to the observed apparent total column ozone changes of the Brewer spectrophotometer are due to the CLV [aka LD] variations and the relative spectral changes between the specific wavelengths used by the Brewer spectrophotometer."

In contrast to our measurements, which are based on pairs of wavelengths (e.g., 305 nm / 340 nm), Brewer retrievals depend on four wavelengths (or sometimes five wavelengths as described by Kazadzis et al. (2007)) and the calculation of a weighted mean. This method could conceivably result in the spurious decrease of 15 DU reported by Gröbner et al. (2017). Since we have not operated Brewers and do not know the details of the algorithm, it is beyond the scope of our paper to calculate LD corrections for Brewer TOC retrievals.

- **Kazadzis et al. (2007), Zerefos et al. (2000; 2001)**

Kazadzis et al. (2007) and  Zerefos et al. (2000; 2001)  attribute this apparent reduction in TOC to contamination of the Brewer's direct measurements by diffuse radiation in the instrument's field of view (P24, L25-28). Since we are not experts in Brewer measurements we cannot determine whether this assertion is correct. However, we suspect that the combination of different wavelengths, as described by Gröbner et al. (2017), may have been the dominant factor in causing the apparent reduction in TOC. It would be interesting to test whether this apparent reduction would turn into a increase if the Brewer measurements reported by Kazadzis et al. (2007) were reprocessed using  a pair of wavelength (as done by us and Bojkov (1968)) instead of using the standard

Brewer algorithm. Again, such a undertaking   is beyond the scope of our paper.

- **Kazantzidis et al (2007)**

Kazantzidis et al (2007) report on TOC measurements performed during the solar eclipse of 29 March 2006 using NILU-UV filter radiometers, which measure global irradiance. TOCs retrieved by Kazantzidis et al. (2007) using the 305 nm / 320 nm wavelength pair increase as the visible fraction of the sun decreases from ~60%  to ~20% up to about 5 DU (Fig. 6 of Kazantzidis et al. (2007)). These increases are by and large consistent with our results. However, as the visible fractions of the sun decreases further, the derived TOCs decrease. The decline is small (<5 DU) for two sites, but quite large (~45 DU) at Kastelorizo. Kazantzidis et al. (2007) attribute this decrease to an artifact in their irradiance measurements and cite Zerefos et al. (2001) and Kazadzis et al. (2007). Considering that both authors use Brewer direct  measurements, we don't think that these publications are applicable to NILU-UV measurements, so the drop in TOC (although very small for 2 out of 3 sites) is indeed a mystery.

In summary, while we cannot explain in detail the patterns of TOC measurements performed by other groups in the past, we conclude that the increase in TOC seen in our measurements is consistent with theory. Furthermore, the results by Gröbner et al. (2017) suggest that the decrease in Brewer TOC measurements during an eclipse could partially be caused by the fact that ozone is calculated from four wavelengths rather than two wavelengths as in the case of our results. For example, by combining two wavelength pairs, the effect of aerosols is reduced in standard Dobson retrievals using the AD "double-pair" (Basher, 1982, Bernhard et al., 2004). However, in the case of an eclipse, this method could conceivably lead to incorrect results because the wavelength dependence from solar LD is incorrectly attributed to aerosols. At this point, this explanation is just an untested hypothesis, but the fact that the LD effect calculated by Gröbner et al. (2017) has a similar pattern than the TOC variations measured by two Brewer spectrophotometers during the solar eclipse of 20 March 2015 suggests that this hypothesis warrants further examination. However, this is beyond the scope of this paper.

**Change to manuscript**

We do not know with confidence why most studies in the past report artificial drops in TOC during solar eclipses and do not want to go too far into speculation. However, the hypothesis is intriguing that ozone retrievals with four wavelengths instead of two could potentially lead to systematic errors because LD and aerosol effects become mixed. We therefore added the
following sentence to Sect. 8.3:
 "We speculate that the observed drop in Brewer TOC measurements during an eclipse is
 partly caused by the fact that Brewer data are typically calculated from four instead of
 two wavelengths, as it is the case for our results. Using more than two wavelengths has
 the advantage that aerosol effects can be reduced (Basher, 1982). However, this method
 could potentially lead to errors if wavelength-dependent changes in irradiance
 caused by the LD effect are incorrectly attributed to aerosols."

**Response to comments by Anonymous Referee #1**

We thank the referee for his or her comments, which we have addressed as follows:

**Comment by Referee**

Paragraph 7.4: I am referring firstly to this paragraph because the paper is becoming quite confusing at this point. If the authors conclude that the LD parameterization by Waldmeier is too simple, particularly in UV range, why this is being discussed previously in the estimation of total O3? It is recommended to move this paragraph to 7.1 and then discuss the results based on those findings on LD parameterizations. In this concept, paragraph 7.3 should follow the new 7.1. In figure 10, the effect of the eclipse is masked by the dominant effect of changing solar zenith angle. The authors are asked to subtract the solar zenith angle effect by using the AOD and O3 measurements just at the end (or start) of the eclipse (as inputs in model calculations) and, then, discuss the effect of the eclipse and the changes on aerosols and ozone (these paragraph should follow now).

**Authors' Response**

We have changed the sequence of Sect. 7 as suggest by the referee. The sequence is now:

7.1 Effect of solar limb darkening

7.2 Direct-to-global ratio

7.3 Aerosol optical depth

7.4 Total ozone column

7.5 Measurements near and during totality

We have also changed the sequence of Sect. 8 (Discussion) to be consistent with Sect. 7. The new sequence of Sect. 8 is now:

8.1 Magnitude of solar limb darkening

8.2 Variations of direct-to-global ratio

8.3 Total ozone column variations during an eclipse

8.4 Validation of GUVis-3511 measurements

Following the suggestion by the referee, we have compared the measured direct-to-global ratio with the corresponding modeled ratio and have added a new panel to Figure 10 (now Figure 11) , which shows the ratio of measurement and model. The text was modified to describe the new panel, see below.

**Change to manuscript**

The following was added to Sect. "Direct-to-global ratio" (now Section 7.2)

"The measured direct-to-global ratio was compared with the modeled direct-to-global ratio using the same method as that employed in Sect. 7.1. In brief, the measured direct-to-global ratio was divided by the modeled direct-to-global ratio. To correct for the difference between measurement and model, linear functions were again constructed to match the ratios of measurement and model at the times of the 1$^{st}$ and 4$^{th}$ contact  Finally, the ratios were normalized by dividing with these linear functions. The resulting bias-corrected ratios of the measured and modeled direct-to-global ratio are shown in Fig. 11b and are denoted $R_{DG}(\lambda_i)$.

Before 15:00, the measured direct-to-global ratio is smaller than 0.03 at wavelengths in the UV-B

and values of $R_{DG}(\lambda_i)$ are subject to large uncertainties at these wavelengths. Before 15:20,

$R_{DG}(\lambda_i)$ varies between 0.55 and 0.87 for wavelengths in the visible range, indicating that the measured direct-to-global ratio is 13 to 45 % below the modeled one.  These low values can be attributed to increased aerosol loading prior to the start of the eclipse. After the start of the eclipse (1$^{st}$ contact), values of  $R_{DG}(\lambda_i)$ start to drop to reach a local minimum at 16:52, approximately

30 minutes before the eclipse maximum. The decline has a clear wavelength dependence with wavelengths in the UV-B decreasing the most (up to 20 %), followed by wavelengths in the visible (up to 15%) and IR (up to 3.5%). After this local minimum, $R_{DG}(\lambda_i)$ slowly increases during the remainder of the eclipse.

The change of $R_{DG}(\lambda_i)$ over the period of the eclipse could either be caused by processes initiated by the occlusion of the Sun or by variability from aerosols. Aerosols effects are more likely because any eclipse effects should conspicuously peak at the time of totality, not 30

minutes earlier. The wavelength dependence of $R_{DG}(\lambda_i)$ is also characteristic for aerosol effects (Sect. 7.3). In addition, the  minimum values of $R_{DG}(\lambda_i)$ during the eclipse are well within the range of values observed prior to the eclipse, which could be unambiguously attributed to
aerosols."

[Figure]

Fig. 1. Panel (a): ratio of direct-to-global spectral irradiance. Measurements are shown as heavy lines. Thin
lines connect measurements at the start of end of the eclipse and are drawn to guide the eye. Panel (b):
Ratio of the measured and modeled direct-to-global ratios, corrected for the bias between measurement and
model. Numbers "(1)" and "(2)" in the legend indicate the channel number for channels equipped with
identical wavelengths.
In Section 8.1., the following paragraph:
"The direct-to-global ratios shown in Fig. 10 increase with time as expected from the
decrease in SZA. Deviations from a straight line between the 1st and 4th contact are less
than 0.04. It is difficult to determine whether this deviation is caused by processes
initiated by the occlusion of the Sun, by variability from aerosols, or artifacts of the algorithm to calculate the direct irradiance from shadowband data. We therefore consider the deviation of 0.04 as an upper limit for the eclipse effect."

was removed and replaced with:

"Relative to model calculations for the unoccluded Sun, direct-to-global ratios during the eclipse decreased by up to 20 % in the UV-B, 15 % in the visible and 3.5 % in the IR (Fig. 1b). The largest decrease was observed about 30 minutes before totality. The timing of this decrease plus its spectral dependence suggest that changes in aerosol amounts are the main driver for the observed drop in the ratio. However, aerosol effects are difficult to decouple from processes initiated by the occlusion of the Sun or artifacts of the algorithm used to calculate the direct irradiance from shadowband data."

Note that this paragraph is now part of Section 8.2.
* * *
**Comment by Referee**

Page 2, lines 15-20: Kazantzidis et al (2007) were measuring with NILU-UVs and calculated total ozone, too. They reported a slight increase when the visible part of the sun was more than 20% and decreased significantly as the eclipse progressed.

**Authors' Response and change to manuscript**

The reference to Kazantzidis et al. (2007) has been added to the introduction in support of the following statements:

"study wavelength-dependent changes in spectral irradiance"

"short-term and longer-lasting fluctuations in the total ozone column (TOC)"

Furthermore, we have added the following to Sect. 8.3. of the Discussion:

"Finally, Kazantzidis et al (2007) discuss TOC measurements performed with NILU-UV filter radiometers at several locations in Greece during the total solar eclipse of 29 March 2006. They did not observe any periodic fluctuations in TOC and only report a small increase in TOC of about 5 DU as the visible fraction of the Sun decreases from ~60% to ~20%. This small change could be caused by incomplete correction of the LD effect."
* * *
**Comment by Referee**

Figure 3 and relevant text: please provide a figure only with the UV wavelengths, no logarithmic axes. From the literature it seems that the bandwidth of 305nm channel is quite wide: it is more than 10nm even at full width at half maximum. In this case, please discuss the possible implications when measuring with this high bandwidth.

**Authors' Response and change to manuscript**

Figure 3 was plotted separately for the UV, visible and IR wavelength range on a non-logarithmic x-axes. The new figure plus raw data in text format will be made available as Supplements.

The following was added to Section 3:

"The spectral bandwidth of all channels is approximately 10 nm full width at half maximum (FWHM) with the exception of the two channels at 305 nm, which have a bandwidth of 18.5 nm."

The following was added to the Caption of Fig. 3:

"A version of the figure, plotted separately for UV, visible, and IR wavelengths, is part of the Supplement."

The following was added to Sect. 5.1:

"Aerosol optical depth was not calculated for the two 305 nm channels because of the large bandwidth of these channels and the strong interference with ozone absorption at this wavelength. Both factors lead to large uncertainties."

As described in Section 5.2., TOCs are calculated using look-up tables that are based on response-weighted global irradiance, i.e., the spectral irradiance weighted with the spectral response functions shown in Fig. 3. TOC calculations therefore take the large bandwidth of the

305 nm channels into account.

The impact of the relatively large bandwidth of the 305 nm channels on spectral irradiance calibrations is implicitly addressed by the vicarious calibration method described in Sect. S1.2 of the Supplement. Uncertainties to derive spectral irradiance at 1 nm resolution from response- weighted irradiance are discussed in Section S1.3 of the Supplement.

| | |
|---|---|
| 1 | **Comment by Referee** |
| 2 | Page 5, lines 15-24: please provide/add some sentences about the performance of this method on |
| 3 | estimating the direct and the shadowband corrected spectral irradiances. |
| 4 | |
| 5 | **Authors' response and change to manuscript** |
| 6 | The following was added to the paragraph in question: |
| 7 | "The uncertainty of our method was estimated by Witthuhn et al. (2017). AOD can be |
| 8 | retrieved with an uncertainty of 0.02 for all channels within a 95 % confidence interval." |
| 9 | |
| 10 | **Comment by Referee** |
| 11 | Figure 4 and relevant text: What is the expected cosine error in UV wavelengths? Are there any |
| 12 | measurements? Are the measurements in UV channels (used for this study) cosine corrected? |
| 13 | How significant this effect is expected to be during the solar eclipse (when the direct component |
| 14 | of solar irradiance is minimized)? |
| 15 | |
| 16 | **Authors' Response** |
| 17 | The cosine error in the UV has also been measured but results were omitted from Fig. 4 because |
| 18 | these measurements are very noisy due to the low output of the FEL used for the characterization |
| 19 | in the UV. In general, the cosine error of wavelengths between 305 and 380 nm is similar to that |
| 20 | at 395 nm because scattering properties of PTFE (Teflon) deteriorate only towards longer |
| 21 | wavelengths but not towards shorter wavelengths. We have now generated a composite figure |
| 22 | showing cosine errors in the UV in the upper panel and those for the visible and IR ranges in the |
| 23 | lower panel: |

[Figure]

Fig. 2: Cosine error of the GUVis-3511 radiometer in one azimuthal plane at wavelengths between 305 and 395 nm (Panel a) and 395 and 1020 nm (Panel b). Measurements performed at the orthogonal plane are similar. Measurements at wavelengths below 395 nm are affected by noise due the low UV output of the incandescent lamp used for the characterization.

The new figure will be included in the Supplement. Measurements of UV channels used in our study are cosine error corrected. Corrections of channels between 305 and 380 nm are based on the cosine error of the 395 nm channel to avoid spurious corrections caused by the noisy cosine response data. The cosine error correction takes the measured direct-to-global ratio into account. The accuracy of the cosine error correction method is therefore not affected by the solar eclipse.

**Change to manuscript**

The following was added to Sect. 3:

> "Measurements of the cosine error at wavelengths below 395 nm are affected by noise due the low UV output of the incandescent lamp used for the characterization.

Corrections for the cosine error of UV channels described further below are based on the measured cosine error at 395 nm."

The following was added to the caption of Fig. 4:

"(A figure with cosine errors in the UV is provided in the Supplement.)"

The following was added to Sect. 4:

"Both direct and global irradiance measurements were corrected for the cosine error of the instrument's collector as described by Morrow et al. (2010), based on the measured cosine error (Fig. 4) and the ratio of direct and global irradiance extracted from the shadowband measurements."

**Comment by Referee**

Figure 8: Aerosol Optical Depth (AOD) around 305nm is not depicted in figures 8a and 8b. Why?

Do you think that the LD correction (8b) by Pierce is valid also for the lower UV wavelengths? If yes, how this decrease of AOD in these wavelengths can be explained? There is an assumption that this is not a measurement artifact but an unknown absorber. Which type of absorber could change the expected AOD values only at 314.2, 319.4nm (and 442.4nm!)? Moreover, what would be the AOD estimations if you apply the Neckel LD parameterization?

**Authors' Response and change to manuscript:**

The uncertainty in calculating AOD at 305 nm is far too high to be useful. As already mentioned earlier, we have added the following to Sect. 5.1 to make this clear:

"Aerosol optical depth was not calculated for the two 305 nm channels because of the large bandwidth of these channels and the strong interference with ozone absorption at this wavelength. Both factors lead to large uncertainties."

Yes, we believe that the LD correction by Pierce is valid also for the lower UV wavelengths because the difference in global irradiance between measurement and theory is also reasonable at nm (Fig. 11d of original manuscript). Furthermore, the LD corrections by Pierce and Neckel (which were derived from independent datasets) agree to within 1.8% at 305 nm.

We do not know with certainty why the AODs at 314.3, 319.4, and 442.4 nm are below the AODs expected from the Ångström parameterization, which was derived using channels between 340 and 1020 nm, excluding the channel at 940 nm, as stated in the manuscript. These low values could be caused by absorbing aerosols released by the near-by fires or measurement errors, but this is speculation. If the low values were due to measurement errors, one would expect that the discrepancies for the three wavelengths observed at 16:02:58 and 19:00:30 are of similar magnitude because measurements at both times were processed in the same way. The fact that these discrepancies are larger at 19:00:30 suggests that they are caused by real changes in aerosol properties. While these drops could be caused by an unknown absorber, we removed this suggestion from the text because we don't have evidence that such an absorbing aerosol or gas was indeed present.

Considering the uncertainty with respect to the cause of the low AODs at the three wavelengths, we changed the last sentence of the paragraph to:

> "As measurements at the two time were processed in the same way it seems unlikely that
> the discrepancies are caused by a measurement artifact."

With respect to the question "Moreover, what would be the AOD estimations if you apply the Neckel LD parameterization?" we note that AODs shown in the figure were calculated for times before the start (16:02:58) and after the end (19:00:30) of the eclipse. Hence, no LD parameterization was applied or is necessary.
* * *
**Comment by Referee**

Paragraph 7.2: It is quite surprising that although AOD can be estimated by direct GUV measurements, this is not happening for the total O3 amount as well, despite that direct measurements (divided by global ones) are presented in figure 10. Moreover, the 305/340 wavelength ratio methodology to derive total O3 is based on model calculations as a function of O3 and solar zenith angles but under cloud-free skies e.g. specifically defined direct and diffuse components of solar irradiance. This is not the solar eclipse case. And this is accounted by the authors. However, how valid are the LD parameterizations on CHANNEL irradiances when it is known that the direct/diffuse ratio has significant spectral sensitivity? And how repeatable will be their results if the they use: a) direct irradiances, b) different channel ratios, e.g. 305/320,

312/340? The authors here should acknowledge a couple of very crucial facts: 1) the 305/340

wavelength ratio of channel irradiances is a well-known method that can be used for total O3

estimations but it is accompanied by significant uncertainties (aerosol optical depth and scattering properties, cloudiness, ozone profile, direct/diffuse irradiance etc), 2) the comparison with ozone values or previous studies derived from instruments measuring the direct irradiance should be done under the acknowledgement that these measurements are correct not only because they are the standard ones but also because they are correct in terms of physics and the best in terms of overall uncertainty.

**Authors' Response**

First, we like to emphasize that this paper is not about the most accurate or appropriate method to calculate TOC. TOC measurements were mainly included in the manuscript (i) to illustrate that

TOC measurements during an eclipse are affected by the solar limb darkening (LD) effect and that errors arising from this effect can be greatly reduced with appropriate LD corrections, and (ii)

to determine whether gravity waves generated by the Moon's shadow can cause oscillations in

TOC as suggest by several authors in the past. In our opinion, both questions were sufficiently addressed with the results presented in the original manuscript. Please see also our response to the comments by Forrest M. Mims.

Motivated by the referee's comments, we have calculated total ozone now also from global irradiance of the wavelength pairs 305 / 313 and 313 / 340. Results are presented in Fig. 3. Fig. 3a shows results without LD correction. TOCs for the 305 / 340 pair (blue) are identical with the

"uncorrected" data shown in Fig. 9 of the original manuscript. Ozone values for the 313 / 340 are

*lower* than those of the 305 / 340 pair and exhibit a *negative* slope as a function of time. In contrast, ozone values for the 305 / 313 pair are *larger* than those of the 305 / 340 pair and exhibit a *positive* slope. These biases are likely caused by a small systematic error in measurements of the 313 nm channel. When measurements of this channels are scaled with a factor of 0.95 and ozone retrievals repeated, TOCs derived from the three wavelengths pairs are consistent to within

±2.5 DU outside the period of the eclipse (Fig. 3b). A systematic error of 5% is well within the expanded uncertainty of 7.3% of the 313 nm channel (see table S1 of Supplement.). Using this adjustment, measurements of all pairs agree with OMI data to within 5 DU (1.7%) (red line in

Fig. 3).

[Figure]

Fig. 3. Total ozone column derived from global irradiance measurements of the GUVis-3511 radiometer using different wavelength pairs. Panel (a): comparison of ozone retrievals with the 305 / 340 pair (blue), the 313 / 340 pair (orange), and the 305 / 313 pair (green). Panel (b): same as Panel (a) but measurements of the 313 nm channel were scaled with a factor of 0.95 before calculating ozone. Panel (c): TOC retrievals for the 313 / 340 pair using either no correction (blue), or the LD-corrections by Waldmeier (grey), Pierce (cyan), and Neckel (green). Panel (d): same as Panel (c) but for the 305 / 313 pair. Measurements of the 313 nm channel were scaled by 0.95 for all retrievals shown in Panels (c) and (d). Long-dashed lines indicate the start and end times of the eclipse and the short-dashed line indicates the time at totality. Note that y-scales of the various panels differ.

TOC data shown in Fig. 3c and 3d were corrected for the LD-effect similar to the TOC data derived from the 305 / 340 pair shown in Fig. 9 of the original manuscript. As in that figure, TOCs calculated for the 313 / 340 and 305 / 313 pairs (Fig. 1c and 1d, respectively) spike at the time of totality when no LD correction is applied. The LD correction by Waldmeier has only a marginal effect on this result. In contrast, LD corrections using the parameterizations by Pierce and Neckel greatly decrease this spurious spike.

Calculations of TOCs from global irradiance using the pairs 305 / 313 and 313 / 340 therefore confirm the conclusions reached from results of the 305 / 340 nm pair that were presented in the original manuscript:

- There are no periodic oscillations in the TOC, either before or after totality, that could be attributed to the effects of bow waves from the Moon's shadow.
- The LD corrections by Pierce and Neckel greatly reduce the spurious spike in TOCs during totality resulting from the LD effect. In contrast, the correction that is based on the parameterization by Waldmeier is too small.
- The spurious TOC spike is larger for the 313 / 340 pair than for the 305 / 340 pair even though the relative difference in the LD effect is smaller for the two wavelengths used in the former pair. This observation is due to the fact that small errors in measurements at 313 nm have a larger effect on TOC than at 305 nm because the absorption coefficient of ozone is much smaller at 313 than 305 nm. The effect of ozone absorption on the 305 / 340 ratio therefore outweighs the LD effect.
- There is no evidence that aerosol effects have an important effect on our TOC measurements. While we agree with the referee's assessment that the method of calculating TOCs from ratios of global irradiance at 305 and 340 nm "is accompanied by significant uncertainties (aerosol optical depth and scattering properties, cloudiness, ozone profile, direct/diffuse irradiance etc)," we believe that these uncertainties are of little relevance for the conclusions reached in our paper. Specifically:
  - AODs were characterized and used for the lookup tables for ozone retrievals. The good consistency of TOC retrievals for different wavelength pairs (after adjusting for the bias of measurements at 313 nm) confirms that aerosol effects were addressed adequately.
  - There were no clouds during the period of the measurements that could have skewed our TOC retrievals.
  - The ozone profile (through the Umkehr effect) becomes only important for SZAs > 75° (i.e., outside the range of SZAs occurring during the eclipse).
  - Changes in the direct-to-global irradiance agree to within 2.5% for the wavelength range 305 - 340 nm, except for a short period centered about totality (Fig. 1 above). So our look-up tables, which were calculated for no-eclipse conditions, are also suitable for calculating TOCs during the eclipse.

We did not calculate TOCs from measurements of direct irradiance because such retrievals have a comparatively large uncertain when direct irradiance in the UV-B (in particular 305 nm) is calculated from shadowband data. In contrast to wavelengths in the UV-A and visible, where direct irradiance under clear skies with moderate aerosol loading makes up a large fraction of the global irradiance, this is not true for wavelengths in the UV-B, in particular at large SZAs (see

Fig. 1a). When using a shadowband, direct irradiance in the UV-B is calculated as the difference of two large numbers (global and diffuse irradiance), and the uncertainty of the result is therefore higher than those of TOCs derived from global irradiance.

Regarding the referee's comment:

"the comparison with ozone values or previous studies derived from instruments measuring the direct irradiance should be done under the acknowledgement that these measurements are correct not only because they are the standard ones but also because they are correct in terms of physics and the best in terms of overall uncertainty."

While we agree that measuring TOC from direct irradiance is the more straightforward method (as only Beer-Lambert's law is involved), the referee's assertion that "these measurements are correct" is not appropriate. There are many error sources that can affect these measurements such as pointing errors, calibration errors, non-linear sensor response, stray light from diffuse radiation that is detected with the direct beam (a problem affecting Brewer measurements as mentioned by several papers cited in our manuscript), uncertainty of the filter functions and their center wavelengths, etc. Considering that we do not have the raw data that other authors have collected to calculate TOCs during historical ellipses we do not want to speculate on the uncertainty of those measurements and their magnitude relative to our results.

As explained above, adding TOC data derived from different wavelength pairs would not change our conclusions about the effect of the solar eclipse on short-term TOC fluctuations. However, results obtained with the 305 / 313 and 313 / 340 pairs are valuable because they confirm the results from the 305 / 340 pair discussed in the manuscript. In order to keep the length of the manuscript within reasonable limits while also respecting the referee's suggestion, we added these new results to the Supplement with references provided in the manuscript.

**Change to manuscript**

The following was added to Sect. 5.2:

"The following three wavelength pairs were used: (1) $\lambda_L = 340$ nm, $\lambda_S = 305$ nm; (2)

$\lambda_L = 340$ nm, $\lambda_S = 313$ nm; (3) $\lambda_L = 313$ nm, $\lambda_S = 305$ nm.

[…]

"While TOCs could also be derived from direct irradiances, these measurements are not discussed here due to the relatively large uncertainty to calculate direct irradiance from shadowband data at wavelengths in the UV-B (280–315 nm), in particular at large

SZAs."

The following was added to Sect: 7.4:

"Results obtained from the 305 / 340 wavelength pair are discussed below. Similar results calculated with the 305 / 313 and 313 / 340 wavelength pairs are presented in Sect. S2 of the Supplement."

[…]

Results obtained  with the 305 / 313 and 313 / 340 wavelength pairs (Sect. S2 of the

Supplement) corroborate these findings."

**Change to Supplement:**

Calculations of TOC from the 305 / 313 and 313 / 340 wavelength pairs have been added to the

Supplement. Please see Section S2 of the new version of the Supplement for details.

**Comment by Referee**

Figure 9 and relevant text: Kazantzidis et al (2007), when using the 305/320 wavelength ratio (in order to reduce the effect of spectral effect of the eclipse on direct and diffuse irradiances)

reported very similar results to those derived from yours when using the 305/340 wavelength ratio AND applying the Neckel or Pierce parameterization. Surprisingly, this paper is referenced only for the comparison of measured and model irradiances. However, the results of this paper for ozone, irradiances and irradiance ratios vs eclipse percentage are not referenced, although it is based on results from 8 narrowband multi-channel NILU-UV6 radiometers.

**Authors' response and change to manuscript**

As mentioned already earlier, we have now added the following to the manuscript to compare our ozone measurements with those reported by  Kazantzidis et al (2007):

"Finally, Kazantzidis et al (2007) discuss TOC measurements performed with NILU-UV

filter radiometers at several locations in Greece during the total solar eclipse of 29 March

2006. They did not observe any periodic fluctuations in TOC and only report a small increase in TOC of about 5 DU as the visible fraction of the Sun decreases from ~60% to ~20%. This small change could be caused by incomplete correction of the LD effect."

Furthermore, we have added the following to Sect. 8.2 (now Sect. 8.1) where we compare the magnitude of solar LD used in our paper with that used by other authors:

"Kazantzidis et al.(2007) have analyzed ratios of global spectral irradiance (305 nm / 380 nm, 312 nm / 380 nm, 340 nm / 380 nm, and PAR / 380 nm) that were measured with NILU-UV filter radiometers at three locations in Greece during the total solar eclipse of 29 March 2006. These measured ratios were compared with theoretical predictions based on the algorithm by Koepke et al. (2001) and the LD parameterization by Waldmeier. As the eclipse progressed, the model underestimated the measured spectral effect, capturing only half of the observed change. For example, measured ratios of spectral irradiances at 340 and 380 nm were 10 % lower close to totality compared to similar ratios calculated for the 1$^{st}$ and 4$^{th}$ contact. The theoretical calculation only predicted a decrease of 5%. Our results suggests that discrepancies between the measured and modeled ratios reported by Kazantzidis et al. (2007) can partly be attributed to limitations of the LD parameterization by Waldmeier used in their model."

**Comment by Referee**

Paragraph 8.1: the authors seem to have a point here. In order to better understand the similarities/differences with previous studies, a more detailed information is needed apart from direct/diffuse ratios: the measured wavelengths and the eclipse percentages should be provided. Moreover, the theoretical calculations from Emde and Mayer are quite capable to estimate the global irradiance (when normalized 5 minutes before totality) at 380 nm but maybe irradiance is significantly underestimated at 312 nm (Kazantzidis et al., 2007, figure 7 and relevant text). This affects directly the diffuse component. Of course, also this result is sensitive to factors like surface albedo, ozone profile and the dynamic range of the measuring system. All these factors should be mentioned.

**Authors' response**

We added the following sentence to Section 8.1. (now Sect. 8.2), which now specifies the measured wavelengths (i.e. erythemal irradiance), eclipse percentage (88%), location (Thessaloniki), and time (11 August 1999):

"However, these results disagree with the results by Zerefos et al. (2000; 2001), who suggest that the erythemal (sunburning) diffuse irradiance was declining at a slower rate than the erythemal direct irradiance during the solar eclipse observed in Thessaloniki, Greece, on 11 August 1999. The largest difference was observed at the time of the eclipse maximum when 88% of the solar disk was obscured and the diffuse irradiance was reduced 30 % less than the direct irradiance."

With regards to factors such as "surface albedo, ozone profile and the dynamic range of the measuring system" that affect irradiance close to totality, we note that we concluded our original manuscript with the sentence:

"During totality, the irradiance at the surface will become also more sensitive to the topography (e.g., the mountains surrounding the measurement sites), surface albedo (and its spectral dependence), and the distribution of ozone in the atmosphere (the ozone profile). These aspects will be discussed in a follow-on publication."

The "follow-on publication" mentioned here is close to completion and will be submitted soon. The factors enumerated by the referee will be discussed in detail therein.

**Comment by Referee**

Page 22-23: This comment refers to the ozone issue, described in detail by the authors. From my point of view, some (or much?) of these differences could be attributed on the measuring methods and the selected pairs of wavelengths. As mentioned before, a decrease of (uncorrected for LD effect) ozone retrievals has been reported by Kazantzidis et al (2007) for the same eclipse when using 8 NILU-UVs. As it was stated earlier in this review, the authors could strongly defend their findings if they will come up with the same results when using other wavelength pairs and the direct GUV irradiances. Unfortunately, this measuring campaign is not accompanied by more instruments.

**Authors' response**

One important "take-home message" from our paper is that accurate LD corrections are necessary for accurate TOC retrievals. Basically all papers that have discussed TOC measurements for previous eclipses use a LD correction that is based on a parameterization that is too simple for wavelengths in the UV. In addition to TOC retrievals using the 305 / 340 wavelength pair, we have now also calculated TOCs from the 305 / 313  and 313 / 340 wavelength pairs (Fig. 3

above). These new calculation corroborate our initial findings that the spurious peak in TOC

measurements caused by LD effect can be greatly reduced by using either the LD

parameterization by Neckel or Pierce. We agree with the referee's assessment that the presence of additional instrument during "our" eclipse could have strengthened our conclusions further. We hope that a multi-instrument campaign can be organized during an upcoming eclipse to get closure on some of the remaining questions raised in our paper.

**Response to comments by Anonymous Referee #2**

We thank the referee for his or her comments, which we have addressed as follows:

**Comment by Referee**

A further examination on the effect of the above (mainly the FOV) issues have to be considered when comparing results of this and older papers using different principles of measurements. For example the definition of the diffuse (or direct) irradiance calculated using a shadow (band) that has spectral, solar zenith angle, (and in this case) also sun- dimensions dependent, apparent shadow dimensions compared with the instrument diffuser, could impact the presented results.

**Authors' Response**

First we like to point out that total ozone column (TOC) was calculated from measurements of global spectral irradiance, not direct solar irradiance as indicated by the referee's introductory statements. The FOV of the shadowband is therefore irrelevant for TOC retrievals. Please see our response to the comments by Forrest M. Mims III regarding TOC calculations.

Shadowband measurements were only used to determined aerosol optical depth and the direct-to-global ratio. The shadowband obstructs a wedge of the sky with a width of 15° when the band is vertical and 13° when it is horizontal. These relative large angles may suggest that the instrument is not able to adequately remove the contribution of circumsolar radiation when calculating the direct irradiance, which would also lead to systematic errors in aerosol optical depth (AOD) retrievals. However, this notion is not correct due to the unique way of shadowband operation and data analysis. Other shadowband radiometer typically use two measurements on either side of the Sun to correct for the signal lost from the portion of the sky that is obscured by the shadowband. For example, side-band measurements of Multifilter Rotating Shadowband Radiometer (MFRSR) are typically performed at 9° from the center of the Sun (Krotkov et al., 2005). Hence, the increase of diffuse radiation towards the Sun is often not corrected adequately. In contrast, direct measurements and AOD retrievals of GUVis-3511 measurements are based on measurements at high sampling rate (15 Hz) whereby the band moves slowly and continuously over the instrument. The processing algorithm determines the irradiance from direct Sun plus the unshaded portion of the sky by analyzing several seconds of measurements when the shade from the band is close to the diffuser but not in contact with it. The algorithm is described in detail by

Morrow et al., (2010) and Witthuhn et al. (2017). The algorithm compensates for the effect of the relatively large FOV, resulting in AOD data that have similar accuracy than that of traditional shadowband radiometers.

**Change to manuscript**

The following text will be added to Sect. 3 of the manuscript:

> "The band obstructs a wedge of the sky with a width of 15° when the band is vertical and 13° when it is horizontal. […] The algorithm compensates for the effect of the relatively large width of the shadowband, resulting in AOD data that have similar accuracy than those of traditional shadowband radiometers, which use measurements on either side of the Sun to correct for the portion of the sky that is obscured by the band. For example, side-band measurements of Multifilter Rotating Shadowband Radiometer (MFRSR) are typically performed at 9° from the center of the Sun (Krotkov et al., 2005) and therefore may not adequately measure circumsolar radiation. Our algorithm alleviates this problem."

**Comment by Referee**

What do you mean by: P6, L10 The calibration was further optimized for the conditions (solar zenith angle, TOC, AOD, etc.) at the measurement site.

**Authors' Response**

As described in the Supplement of the manuscript, the instrument measures "response-weighted irradiance". The conversion to spectral irradiance requires knowledge of the solar zenith angle, TOC, AOD, etc., and uncertainties in these parameters results in uncertainty in the conversion factor. These issues are discussed at length in the Supplement.

**Change to manuscript**

The following will be added after the sentence in question:

> "Details of this optimization are provided in the Supplement."

**Response to comments by Forrest M. Mims III, posted on 24 November 2018**

We thank Mr. Mims for his good comments, which have helped to improve our manuscript.

**Comment by reviewer**

While the results in this paper are certainly intriguing, there are significant differences between the instrument employed by the authors and the TOPS instrument employed by Mims and Mims. The author's instrument uses filters having a FWHM bandpass of 10 nm, while TOPS has filters with a 5-nm bandpass FWHM. TOPS also measured column ozone at 300nm and 305 nm, which is much more sensitive to ozone variations than the wavelengths used by the authors. TOPS is also a direct sun instrument that can provide measurements in a few seconds, while the author's instrument is a full-sky C1 device with an exceptionally long 2-minute scan time. As we have learned from comparisons with an EPA Brewer placed at our site, the much faster scan time provided by TOPS provides higher resolution results and avoids errors caused by aerosol changes that can occur during minute-duration scans. Moreover, TOPS often detects subtle changes in the ozone column missed by Dobson and Brewer instruments, which both require considerably more time for an ozone measurement. Before our findings are ruled out by this paper, we feel that the authors should point out the very significant instrumental differences, especially the filter wavelengths, the filter bandpasses and the time required per scan. In each of these cases, TOPS offers superior performance when compared with their instrument. Thus, the findings of subtle waves in the ozone layer by TOPS cannot be so quickly discounted by this paper. I close by observing that TOPS uncovered a calibration drift in the Nimbus-7 Total Ozone Mapping Spectrometer (TOMS) (Satellite Monitoring Error, Nature 361, 1993). TOPS evolved into Microtops and then Microtops II. All these instruments provide results in close agreement with Brewers and Dobsons at the Mauna Loa Observatory. Thank you for considering the points made herein. Forrest M. Mims III fmimsiii@yahoo.com Interactive comment on Atmos. Chem. Phys. Discuss., https://doi.org/10.5194/acp-2018-1048, 2018.

**Authors' Response**

As described in Section 5.2. of the manuscript, the total ozone column (TOC) was derived from measurements of response-weighted global irradiance measured by the GUVis-3511.

Specifically, ratios of measurement at 340 and 305 nm were compared with similar ratios in a look-up table that was calculated with a radiative transfer model as a function of SZA and TOC.

The look-up table was calculated by taking the response functions of the instrument (Fig. 3 of manuscript) and observing conditions (e.g., aerosol optical depth, (AOD)) into account. The method of calculating TOC from measurements of global irradiance (instead of direct irradiance as it is typically done for Dobson, Brewer, TOPS, and Microtops instruments) was first proposed by Stamnes et al. (1991). We found that the accuracy of TOCs derived from global irradiance is similar to that of data from Dobson instruments or satellite (TOMS, OMI) observations if the look-up table takes local conditions into account (ozone profile, albedo, elevation, etc.) (Bernhard et al., 2005b). For example, this study uncovered systematic errors in Dobson measurements associated with approximations in the standard Dobson retrieval method, which subsequently helped to better understand the limitation of Dobson measurements. We have further validated the method for GUV instruments (Bernhard et al., 2005a). Based on this work we believe that our

TOC measurements are not of inferior quality compared to TOPS measurements and provide further evidence below.

Ability to detect small changes in TOC

As described in the manuscript (P23, L26ff.), our data do not indicate oscillations in TOC that may have been triggered by bow waves from the Moon's shadow. In contrast, Zerefos et al.

(2000) reported peak-to-peak variation in TOC of about 1 %. while Zerefos et al. (2007) reported a peak-to-peak amplitude of 2.0–3.5 %. Finally Mims and Mims (1993) describe a peak-to-peak amplitude of up to 5 DU (1.7 %). We show in the following that our method is capable of detecting fluctuations of this magnitude.

By analyzing the values of our ozone look-up table, we determined that a 1 % change in the ratio of the response-weighted global irradiance at 340 and 305 nm results in a TOC change of 1.1 DU

at the start of the eclipse. At the end of the eclipse (when the SZA is smaller) a 1 % change in the ratio leads to a change of 1.6 DU. As described in the manuscript, ozone calculations are based on the average of 45 seconds (not 2 minutes as stated by the reviewer) of global spectral irradiance measurements that are sampled at 1 Hz. By calculating the standard deviation of these samples and using standard error propagation, we calculated an uncertainty (confidence interval of 95 %)

for the ratio of measurements at 340 and 305 nm of 0.13 % for the start and 0.06 % for the end of the eclipse. By combining these uncertainty estimates with the sensitivity of the TOC to changes in this ratio, we determined that our measurements are able to detect changes in ozone of 0.14 DU

at the start and 0.10 DU at the end of the eclipse. Since the average TOC on 21 August 2017 was about 298 DU, these absolute changes translate to relative changes of 0.05 % and 0.03 %, respectively. Our method is therefore well capable to detect changes of the magnitude of 1 to 3.5

% reported by Zerefos et al. (2000, 2007) and Mims and Mims (1993).

We agree with the reviewer that measurements at 300 nm are more sensitive to changes in ozone than measurements at 305 nm. However, measurements at 300 nm are also noisier than measurements at 305 nm because the irradiance at 300 nm is smaller than that at 305 nm by factors of 10 (end of eclipse) to 30 (start of eclipse). Moreover, small errors in the characterization (e.g., center wavelength) of the filters functions have a larger effect at shorter wavelength. Since we do not have access to a TOPS or Microtops, we cannot determine whether these instruments are really superior to the GUVis-3511 in determining the TOC as the reviewer asserts. (Of note, the shortest wavelength of a Dobson is 305.5 nm and if measurements at 300

nm would be of great advantage, these instruments would likely use a shorter wavelength.) In any case, our analysis above illustrates that our instrument is sensitive enough for detecting bow- waved induced TOC variations of the proposed magnitude.

Effect of aerosols on ozone retrieval

The look-up table for TOC retrievals was calculated with the same model parameters that were used to convert response weighted irradiance measurements to spectral irradiance as described in the Supplement to this publication. Specifically, aerosol extinction was parameterized with

Ångström's turbidity formula by setting the Ångström coefficients α and β to 2.10 and 0.0394, respectively. These values were derived from the instrument's measurement of direct solar irradiance at 19:00:30 after the end of the eclipse. At the start (16:02:58; see Fig. 8b of manuscript), the Ångström coefficients were 1.96 and 0.0570, respectively. To quantify the effect of changing aerosol conditions on TOC retrievals, we modeled the global spectral irradiance for

16:02:58 using either the Ångström coefficients used for the TOC look-up table or the coefficients applicable to this time. The difference in AOD changed the ratio of global spectral irradiance at 340 and 305 nm by only 0.12 %. Using the same sensitivity factor discussed above, we determined that the resulting bias in TOC is 0.13 DU or 0.04 %.

Limb-darkening corrected AODs (Fig. 8a), suggest that the largest AOD during the eclipse occurred at 16:52 when the Ångström coefficients were 1.95 and 0.0788, respectively. (This estimate is somewhat uncertain due to the uncertainty of the limb-darkening correction.) We calculated the aerosol effect in a similar way as before and conclude that the elevated AOD at

16:52 increases the retrieved TOC values by 1.5 DU or 0.5 %. Again, this value is considerably smaller than the fluctuations reported by Zerefos et al. (2000, 2007). Yet, aerosols can explain 1.5

DU of the 5.0 DU increase in limb-darkening corrected TOC measurements that can be seen in

Fig 8a of the manuscript between 16:00 and 17:00.

We also like to point out that TOC retrievals from global irradiance is less sensitive to changes in

AOD than retrievals using the direct beam (such as those utilized by the TOPS instruments)

because photos that are removed from the direct beam by aerosols contribute to the global irradiance. So direct measurements are not necessarily better.

Sampling frequency

TOC measurements discussed in the manuscript are available at a rate of one value every 2

minutes. It is therefore not possible to determine whether the moon's shadows resulted in fluctuations in TOC on a shorter time scale. However, oscillations reported by Zerefos et al.

(2000) had a period of 20 minutes and those reported by Zerefos et al. (2007) had periods ranging between 30 and 40 minutes. Zhang et al. (2017) reported ionospheric bow waves, which manifested themselves as electron content disturbances, with a period of about 25 minutes during the solar eclipse of 21 August 2017. If these bow waves had also affected the ozone layer, these oscillations should have been detectable with our sampling frequency. Finally, Mims and Mims (1993) report that TOC measurements taken during the total solar eclipse of 11 July 1991 show three fluctuations, which began 700 seconds after the third contact with durations of 378, 270 and

432 seconds, respectively. It is curious that these oscillations had a much shorter duration than those reported by Zerefos et al., (2000, 2007) and Zhang et al. (2017). In any case, oscillations of this duration should still shown up in our 2-minute data considering the low noise in our data discussed above and illustrated in Fig. 9, in particular between the 3[rd] and 4[th] contact.

Concluding remarks

We did not "rule out" the findings of the paper by Mims and Mims (1993) as stated by the reviewer. Instead, we simply stated that our data do not support the observation by Zerefos (2000;

2007) and Mims and Mims (1993), and concluded that the question of whether or not bow waves from the Moon's shadow can lead to variations in TOC is still up for debate. We suggested that this debate could be settled by performing limb-darkening-corrected measurements of TOC with different instrument types during one of the upcoming solar eclipses. If such measurements were to show fluctuations in TOC with the same magnitude and timing, the effect of bow waves on

TOC could be convincingly demonstrated. We still believe that this is a reasonable path forward and might stimulate future research.

**Changes to manuscript**

The following text was added to Sect. 5.2 of the manuscript:

"By analyzing the sensitivity of TOC to changes in the irradiance ratio and by quantifying the noise in the measurements of the 305 and 340 nm channel we determined that our measurements are able to detect changes in ozone of 0.14 DU (0.05 %) at the start and 0.10 DU (0.03 %) at the end of the eclipse at 95 % confidence level for constant aerosol conditions. The additional uncertainty in TOC values due to changing AODs was determined to be 1.5 DU (0.5 %)."

In Sect. 7.1. (now Sect. 7.3), the following sentence:

"Results corrected for the LD effect (thick lines in Fig. 8a) indicate that the AOD was monotonically decreasing over the period of the eclipse (from 0.41 to 0.32 at 319 nm and from 0.05 to 0.04 at 1018 nm) without a spurious spike near the time of totality."

was replaced with:

"Results corrected for the LD effect (thick lines in Fig. 8a (now Fig. 12a)) indicate that the AOD was increasing between the 1$^{st}$ contact and 16:52 (from 0.41 to 0.58 at 319 nm and from 0.05 to 0.07 at 1018 nm) and then monotonically decreasing to 0.34 at 319 nm and 0.04 at 1018 nm at the end of the eclipse. Corrected results do not have  a spurious spike near the time of totality."

In Sect. 7.2 (now Sect. 7.4), the following sentence:

"(We have no explanation for the increase in TOC of about 5 DU between 16:00 and

17:00 other than natural variability)"

was replaced with:

> "The increase in TOC of about 5 DU between 16:00 and 17:00 can partly be explained by increasing AODs over this period, which cause a high-bias in the ozone retrievals."

In Sect. 8.3, the following sentence:

> "Bow waves with wavelengths between of 300 and 400 km and a period of about 25 minutes have indeed been observed during the solar eclipse of 21 August 2017 (Zhang et al., 2017)."

was replaced with:

> "Ionospheric bow waves with wavelengths between of 300 and 400 km and a period of about 25 minutes, which manifested themselves as electron content disturbances, have indeed been observed during the solar eclipse of 21 August 2017 (Zhang et al., 2017). If these bow waves had affected the ozone layer, these oscillations should have been detectable with our sampling frequency of one TOC value every two minutes."

Also in Sect. 8.3, the following sentence:

> "These small variations are well within the natural variability of the TOC."

was replaced with:

> "These small variations are well within the natural variability of the TOC and the uncertainty of our TOC retrieval of 1.5 DU (0.5 %), which is mainly caused by the effect of changing aerosols on ozone calculations."

**Response to 2nd comments by Forrest M. Mims III, posted on 21 December 2018**

**General remarks to comment by reviewer**

We thank Mr. Mims for his additional comments but feel that many remarks are beyond the scope of the paper. Our paper is about the importance of applying corrections for solar limb darkening when observing a solar eclipse; the question of whether or not bow waves from the Moon's shadow may result in fluctuations in total ozone column (TOC); and the question of whether or not the ratio of direct-to-diffuse irradiance changes appreciably during the period of a solar eclipse (excluding the period near totality). The paper is NOT about the best, most accurate, or most precise method to measure TOC.

In our response to the first comments by the reviewer (posted on 24 November 2018), we provided new evidence that our method of measuring TOC during the eclipse is precise enough for detecting potential changes in TOC from bow waves. In brief, we concluded that the noise in our measurements is low enough for detecting relative changes in TOC of larger than 0.05 %. In addition, we calculated that changing aerosol concentrations during the time of our observations result in an additional uncertainty in TOC of 1.5 DU or 0.5 %. Even if our uncertainty estimates were too optimistic, for example due to an unknown systematic error in our TOC retrieval method, it would be highly unlikely that variations in our calculated TOC values would anti-correlate with real variations in TOC triggered by bow waves such that the resulting TOC measurements after the 3[rd] contact become basically flat, with a variation of only ±1 DU or ±0.33 % (see Fig. 9 of original manuscript). For comparison, the peak-to-peak amplitude attributed to bow waves reported by Zerefos et al. (2007) was 2.0–3.5 %, and the peak-to-peak amplitude reported by Mims and Mims (1993) was 1.7 %.

In conclusion, we cannot rule out that that the Moon's shadow led to variations of TOC in the order of ±0.3 % during the eclipse observed by us, but note that this upper limit is considerably lower than the fluctuations reported by Zerefos et al. (2007) and Mims and Mims (1993).

**Changes to manuscript**

In the abstract, the following sentence:

"In contrast to results of observations from earlier solar eclipses, no fluctuations in TOC
were observed that could be attributed to gravity waves."

will be replaced with:

"In contrast to results of observations from earlier solar eclipses, no fluctuations in TOC
were observed that could be unambiguously attributed to gravity waves."

In Sect. 8.3., the following sentence:

"Our data do not support the observation by Zerefos (2000; 2007) and Mims and Mims
(1993) that bow waves from the Moon's shadow lead to oscillations in TOC."

will be replaced with:

"Our data do not support the observation by Zerefos (2007) and Mims and Mims (1993)
that bow waves from the Moon's shadow may lead to oscillations in TOC with a peak-to-
peak amplitude exceeding 1.5 %."
* * *
**Comment by reviewer:**

TWO KEY POINTS: The authors have made several important revisions to their paper, but they
simply must remove their erroneous assertions to the effect that: (1) the signal at 300 nm is noisy,
which suggests poor measurements by TOPS (which used 300 nm and 305 nm) and (2) full-sky
measurements of the ozone layer are "similar" to the direct sun measurements employed by
Dobsons (and TOPS, Microtops, Brewers and Pandoras). These inappropriate claims and their 2-
minute measurement time support their general assertions that raises doubts about the papers by
me and others.

**Authors' Response**

Regarding (1): We do not assert anywhere in the paper that signals of the TOPS instruments are
noisy or that measurements of this instrument are of poor quality. In fact, we do not even mention
"TOPS" in the manuscript. The discussion of the noise characteristics of the TOPS instrument is
only part of the reviewer's first post and our response. It is and will not be part of the paper.

Regarding (2): We do not state in the manuscript that "full-sky measurements of the ozone layer
are 'similar' to the direct sun measurements employed by Dobsons (and TOPS, Microtops,
Brewers and Pandoras)". We do not compare the accuracy of our TOC measurements with that of
other methods. Like in the case of (1), the discussion on the quality of the different methods of
measuring ozone is only part of the reviewer's first post and our response.

**Changes to manuscript**

None.
* * *
**Comment by reviewer:**

1. OPTIMUM WAVELENGTH SELECTION FOR MEASURING TOTAL COLUMN OZONE DURING A SOLAR ECLIPSE: The authors claim the 300-nm minimum employed by TOPS provides a noisy signal. They erroneously observe that: "Of note, the shortest wavelength of a Dobson is 305.5 nm and if measurements at 300 nm would be of great advantage, these instruments would likely use a shorter wavelength." The 305.5-nm Dobson minimum wavelength was selected due to the use of the instrument across a wide band of latitudes. However, there is ample signal at 300 nm at my site (29.9 N), as demonstrated by the instrument's detection of an error in NASA's Nimbus-7 TOMS ozone instrument (Mims, Nature, 1993). The 300 nm signal was especially strong at 22 N during the 1991 solar eclipse reported in my paper in GRL. The authors might be right about the 300-nm signal at their northerly location. But they cannot compare what might have been a noisy 300 nm signal at their northerly 44.36 N site to the much higher amplitude 300-nm signal at the 22 N site for the eclipse I measured. Furthermore, DeLuisi and others have demonstrated that wavelengths below 305 nm provide more accurate ozone measurements than higher wavelengths. (This explains why TOPS found the satellite error.)

**Authors' Response**

Similar to the last comment, we do not discuss the TOPS instrument in our paper. The quote in the reviewer's comment above is again from our response to the reviewer's first post. Also, we do not discuss the best wavelengths to be used to calculate TOC for the location of "our" eclipse. We simply state in the manuscript that TOCs were calculated from the measurements of the GUVis-3511's channels at 305 and 340 nm.

**Changes to manuscript**

None.
* * *
**Comment by reviewer:**

Of special concern is this from the author's abstract: "The total ozone column (TOC) was derived from measurements of global irradiance at 306 and 340nm." While the 306 nm minimum is appropriate for 44.36 N, the 340 nm upper wavelength is far too high, for it allows for significant aerosol errors. The closely-spaced 300 nm and 305 nm wavelengths of TOPS nearly eliminated the aerosol error, which explains this instrument's excellent accuracy when compared with hundreds of satellite measurements and an EPA Brewer for 60 days at my site. Consider this abstract by Saunders et al in High-Precision Atmospheric Ozone Measurements Using ... trial spectral irradiances between 290 and 305 nm (JGR Atmospheres 1984 https://doi.org/10.1029/JD089iD04p05215 ) "Abstract "It is shown theoretically that many errors are significantly less when determining atmospheric ozone thicknesses from measurements of solar terrestrial spectral irradiance in the wavelength region between 290 and 305 nm as compared to the to the 305- to 340-nm region employed by the Dobson spectrophotometer. In order to test this conclusion experimentally, an elaborate set of state-of-the-art measurements have been made in the shorter wavelength region in Gainesville, Florida, between June 13 and June 18, 1980. Details of these measurements, including an extensive error analysis, are presented and indicate that such short-wavelength measurements, particularly between 295 and 305 nm, can be used to detect long-term changes of atmospheric ozone with an uncertainty not exceeding 1%. Observing conditions restricted the Gainesville measurements to zenith angles of less than 35°. Further investigations are required to determine the shortest wavelength that can be used at significantly greater zenith angles."

**Authors' Response**

We agree with the reviewer that aerosols will lead to systematic errors in TOC measurements if the wavelengths used for the retrieval are far apart. However, we have estimated the uncertainty in our TOC retrievals for the period of interest and have concluded that variations in aerosols during the period of the eclipse cause an uncertainty in TOC of only 1.5 DU or 0.5 %. We note that this uncertainty refers to the precision of TOC measurements (the metric of relevance to the paper), not absolute accuracy, which could be worse. However, our TOC retrievals agree to within 3 DU (or 1%) with OMI, suggesting that also the accuracy of our data is within acceptable limits.

The discussion of whether or not TOC measurements should be based on wavelengths in the 290 to 305 nm range is of little relevance to the paper because the GUVis-3511 radiometer has no channels with wavelengths below 305 nm. In addition, while TOC measurements using wavelengths shorter than 305 nm could indeed be more accurate, as the JGR quoted by the reviewer suggests, we like to point out that a higher accuracy can only be achieved if the filters of the instrument in question are well characterized. No filter instrument measures at exactly a nominal wavelength and all real instruments (including the TOPS) use filters with a finite bandpass. If a hypothetical channel that is supposed to measure at 300 nm measures in fact at 300.5 nm, TOC errors will result. We are not implying that this is the case for the TOPS instrument, but just like to point out that performing ozone measurements with channels at 300 and 305 nm can potentially lead to errors that could be comparable in magnitude to those affecting measurements using channels that are farther apart, and as a result are more sensitive to aerosols. Again, we cannot, and do not want to, assess the quality of TOPS measurements because we do not use data of this instrument in the paper and are not familiar with its characteristics.

The sentence "The total ozone column (TOC) was derived from measurements of global irradiance at 306 and 340nm." in the abstract does not judge whether this wavelength selection is the most appropriate. It simply states what was done and we see no reason to change it.

**Changes to manuscript**

None.
* * *
**Comment by reviewer:**

2. FULL-SKY VS. DIRECT SUN TOTAL OZONE MEASUREMENTS: The authors state in their paper: "The method of calculating TOC from measurements of global irradiance (instead of direct irradiance as it is typically done for Dobson, Brewer, TOPS, and Microtops instruments) was first proposed by Stamnes et al. (1991). We found that the accuracy of TOCs derived from global irradiance is similar to that of data from Dobson instruments or satellite (TOMS, OMI) observations if the look-up table takes local conditions into account (ozone profile, albedo, elevation, etc.) (Bernhard et al., 2005b)." The authors suggest that global irradiance provides TOC data ". . . similar to that of data from Dobson instruments or satellite. . . ." But the authors provide no data or citations to support this assertion, while leaving open the counter suggestion that Dobsons (and TOPS, Brewers and Pandoras) could be replaced by much simpler instruments that measure global irradiance and require no tracking and pointing.

**Authors' Response**

We do not state in our paper:

"The method of calculating TOC from measurements of global irradiance (instead of
direct irradiance as it is typically done for Dobson, Brewer, TOPS, and Microtops instruments) was first proposed by Stamnes et al. (1991). We found that the accuracy of TOCs derived from global irradiance is similar to that of data from Dobson instruments or satellite (TOMS, OMI) observations if the look-up table takes local conditions into account (ozone profile, albedo, elevation, etc.) (Bernhard et al., 2005b)."

as asserted in the comment by the reviewer above. This quote is taken from our comment to the first post of the reviewer. Instead, we simply state in the paper:

"This method was first proposed by Stamnes et al. (1991) and was validated for GUV instruments by Bernhard et al. (2005a)."

We are puzzled by the reviewer's comment:

"The authors suggest that global irradiance provides TOC data ". . . similar to that of data from Dobson instruments or satellite. . . ." But the authors provide no data or citations to support this assertion, while leaving open the counter suggestion that Dobsons (and TOPS, Brewers and Pandoras) could be replaced by much simpler instruments that measure global irradiance and require no tracking and pointing."

because the paper and the response to the post of the reviewer does include two citations (i.e., Bernhard et al., 2005a, and Bernhard et al., 2005b) where TOC measurements from global irradiance are compared with Dobson direct measurements and satellite (TOMS, OMI) observations.

In Bernhard et al., 2005b, we conclude that:

"When Dobson measurements are corrected for the temperature dependence of the ozone absorption cross section and accurate air mass calculations are implemented, data from the three instruments agree with each other to within ±2% on average and show no significant dependence on SZA or total ozone."

The "three instruments" quoted above refer to the Dobson; our SUV-100 spectroradiometer, which measures global irradiance; and TOMS. The reviewer may not agree with this conclusion, but it is false to assert that "the authors provide no data or citations to support this assertion".

We further like to point out that the method of retrieving TOC from spectra of global irradiance measured by SUV-100 radiometers has been published by Bernhard et al. (2003). This paper also includes an uncertainty estimate of the method. The abstract ends with:

"On average, the new algorithm generates ozone values in spring 2.2 % lower than TOMS observations and 1.8 % higher than Dobson measurements. From the uncertainty budget and the comparison with TOMS and Dobson it can be concluded that ozone values retrieved from global UV spectra have a similar accuracy as observations with standard instrumentation used for ozone monitoring."

**Changes to manuscript**

In Sect. 5.2., we will replace

"This method was first proposed by Stamnes et al. (1991) and was validated for GUV instruments by Bernhard et al. (2005a)"

with

"The method of calculating TOC from measurements of global irradiance was first proposed by Stamnes et al. (1991) and was further validated by Bernhard et al. (2003; 2005b). The application of the method to GUV instruments was described by Bernhard et al. (2005a)."
* * *
**Comment by reviewer:**

Of course, this is not the case. The authors have gone much, much too far in suggesting their global, full-sky data is "similar" to direct sun measurements made by the recognized standard for nearly a century and all other ozone instruments. I have always been intrigued by the prospect of accurate ozone retrievals by pairs of global UV measurements at closely-space wavelengths. I urge the authors to prepare a detailed paper about their claim that global TOC measurements are "similar" to those by traditional direct sun instruments. An ideal comparison would be with Brewers, which measure both direct sun and global TOC. Meanwhile, it is inappropriate to make an unsupported claim about "similar" results.

**Authors' Response**

The reviewer suggests to prepare a detailed paper about our claim that global TOC measurements are "similar" to those by traditional direct sun instruments. This paper has already been written. In fact, there are two: Bernhard et al. (2003; 2005b) and we therefore don't agree with the reviewer's conclusion that it is inappropriate to make an unsupported claim about "similar" results.

We do not advocate in any of our papers that the established network of Dobson and Brewer instrument should be replaced with instruments measuring global irradiance. The "global irradiance method" is only sufficiently accurate if the ozone profile is known with sufficient accuracy, in particular for large solar zenith angles. So the method is not as independent as direct measurements that rely on Beer-Lambert's law and hence do not require knowledge of the profile. However, when there is cloud cover, the direct method cannot be used and Stamnes et al. (1991) showed that the "global" method is equally accurate as the zenith sky method used by Dobsons when the Sun is obstructed by clouds.

As a side note, because TOC retrievals from global irradiance depend on the ozone profile, the profile can in fact be determined from such measurements using a variant of the Umkehr method. We have recently published this "Global-Umkehr method" (Bernhard et al., 2017). Again, we not propose that this method is superior to the standard Umkehr method, which relies on zenith sky observations. The paper is just a prove of concept and makes the Umkehr method available to locations with global irradiance measurements. The abstract of Bernhard et al. (2017) concludes with "Total ozone columns (TOCs) calculated from the retrieved profiles agree to within $0:7 \pm 2:0$ % ($\pm 1\sigma$) with TOCs measured by the Ozone Monitoring Instrument on board the Aura satellite." This demonstrates again the good accuracy of the TOC retrievals from global measurements if its done right.

**Changes to manuscript**

None.
* * *
Finally, we like to conclude that most of the discussion above is of little relevance to our paper. The only important question is whether or not our TOC retrievals from global irradiance measurements of the GUVis-3511 are of sufficient precision to detect fluctuations in TOC that could be triggered by bow waves. Using a sensitivity analysis, we concluded that this is the case. Based on our results, we cannot rule out that there is an effect with an amplitude of $\pm 1$ DU or $\pm 0.33$ %. We simply conclude that we did not observe fluctuations in TOC that could be unambiguously attributed to gravity waves. It is possible that the magnitude of gravity wave effects varies from eclipse to eclipse, explaining the discrepancy in the data by us, Zerefos et al. (2007) and Mims and Mims (1993), but this is speculation.

**Changes to manuscript**

None.

[revised manuscript text omitted]